# Data-Source Adaptive Online Learning under Heteroscedastic Noise

**Amith Bhat Hosadurga Anand** [1]   **Haipeng Luo** [2]   **Aadirupa Saha** [1]

## Abstract

In this paper, we address the standard $K$-armed multi-armed bandit (MAB) with $M$ heterogeneous data sources, each exhibiting unknown and distinct noise variances, $\{\sigma_j^2\}_{j=1}^M$. The learner performs standard regret minimization, with the added challenge of choosing which data source to query at each round. We propose SOAR (*Source-Optimistic Adaptive Regret minimization*), a novel algorithm that adaptively balances exploration and exploitation by jointly constructing upper confidence bounds for arm rewards and lower confidence bounds for data source variances. Our theoretical analysis establishes that SOAR achieves a regret bound of $\tilde{O}\left(\sigma^{*2}\sum_{i=2}^K \frac{1}{\Delta_i}\right)$, along with a preprocessing cost that depends only on the problem parameters $\{\sigma_j\}_{j=1}^M$, $K$, and grows at most logarithmically with the horizon $T$; where $\sigma^{*2}$ is the minimum source variance, and $\Delta_i$ denotes the suboptimality-gap of the $i$-th arm reward. The $\tilde{O}(.)$ notation hides the polylogarithmic factors in these problem parameters. Notably, despite not knowing the minimum-variance source, SOAR matches the instance-dependent regret of a standard MAB run on a single source of variance $\sigma^*$. This near-optimal instance-dependent regret analysis of SOAR underscores its effectiveness in dynamically managing heteroscedastic noise without incurring significant overhead. Experiments on synthetic problem instances as well as a real dataset (MovieLens 32M) demonstrate that our method significantly outperforms baseline bandit algorithms in terms of regret performance. Our work opens a new direction for adaptively leveraging multiple heterogeneous data sources, extending beyond traditional bandit frameworks.

[1]Department of Computer Science, University of Illinois Chicago , Illinois, USA. [2]Thomas Lord Department of Computer Science, University of Southern California, California, USA. Correspondence to: Amith Bhat Hosadurga Anand <abhat69@uic.edu>.

*Proceedings of the 43$^{rd}$ International Conference on Machine Learning*, Seoul, South Korea. PMLR 306, 2026. Copyright 2026 by the author(s).

## 1. Introduction

Online learning and multi-armed bandits (MABs) constitute a central framework for sequential decision-making under uncertainty (Auer et al., 2002; Lattimore & Szepesvári, 2020). In the classical stochastic MAB model, the learner repeatedly selects an arm and observes a noisy reward drawn from a fixed distribution associated with that arm. A key implicit assumption in this literature is that feedback is generated from a single homogeneous data source.

However, in many real-world systems, observations may originate from *multiple heterogeneous sources*, each exhibiting different and unknown levels of noise. Such scenarios naturally arise in clinical trials conducted across hospitals with varying measurement reliability, recommender systems aggregating feedback from diverse user populations—for instance, movie recommendation platforms where user reviews and ratings vary widely in reliability—and online systems collecting results from multiple sensing or logging pipelines. In these settings, noisy feedback can significantly delay learning, making adaptive source selection a critical component of regret minimization.

**Problem Statement (Informal).** We consider a stochastic $K$-armed bandit problem with $M$ independent data sources. Each arm $i \in [K]$ is associated with an unknown mean reward $\mu_i$, and each source $j \in [M]$ is associated with an unknown noise variance $\sigma_j^2$. At each round, the learner selects an *arm–source pair* $(i, j)$ and observes a noisy reward whose variance depends only on the source. The learner has no prior knowledge of either the arm means or the source variances. The objective is to minimize cumulative regret relative to the optimal arm while adaptively selecting sources to mitigate the effect of heteroscedastic noise on learning efficiency.

**Related Work.** Variance-adaptive bandits exploit empirical variance estimates to refine confidence bounds and improve the exploration–exploitation tradeoff (Audibert et al., 2009; Cowan et al., 2018; Howard et al., 2018). However, these approaches assume a single data-generating process per arm and are not designed to reason about multiple parallel sources with heterogeneous noise. Multi-fidelity bandits study settings in which learners can query information at different levels of fidelity, typically trading off evaluation cost against accuracy (Kandasamy et al., 2016; Song et al.,

2019; Poloczek et al., 2017). While related, our setting differs fundamentally: we focus on adaptive exploration across multiple sources with unknown and heterogeneous noise variances, rather than explicit cost–accuracy trade-offs.

Consequently, existing bandit frameworks do not directly address the challenge of jointly adapting to arm rewards and source selection in an online setting. We summarize how our algorithm relates to these lines of work in Table 1. A more detailed discussion of our problem motivation and related work is deferred to Appendix A.

**Our Contributions:** In this paper, we address these questions by designing an algorithm that jointly explores arms and sources via confidence-based reasoning. We propose SOAR (Source-Optimistic Adaptive Regret minimization), which combines Upper Confidence Bounds (UCBs) on arm rewards with Lower Confidence Bounds (LCBs) on source variances. Our analysis shows that this joint LCB–UCB strategy achieves instance-dependent regret that matches, up to logarithmic factors, that of an oracle MAB with prior knowledge of the optimal source variance.

Our main contributions can be summarized as follows:

- **Problem Formulation.** We introduce a stochastic MAB with multiple heterogeneous data sources and unknown noise variances (Section 2).
- **Baseline Strategies and Limitations.** We analyze natural baseline strategies for source selection and identify instances of suboptimal regret (Section 3).
- **Algorithm Design.** We develop the variance and reward estimation machinery: UCB/LCB constructions, algorithm parameters, and concentration guarantees. Building on these, we propose SOAR, a variance-aware algorithm that jointly selects arms and sources via an LCB–UCB mechanism, with a preprocessing phase that eliminates noisy sources (Section 4).
- **Theoretical Analysis.** We establish high-probability, instance-dependent regret guarantees for SOAR (Section 5), showing oracle-optimality up to logarithmic factors (Remark 5.2) and improvement over the considered baselines (Remark 5.3) .
- **Empirical Evaluation.** We validate our theory on synthetic benchmarks and the MovieLens (Harper & Konstan, 2015) dataset, demonstrating improvements over baselines (Section 6).

## 2. Problem Setting

In this section, we formalize the heterogeneous multi-source MAB problem by specifying the arms, sources, reward model, and regret formulation.

**Useful Notation.** Let $\mathbb{R}_+$ and $\mathbb{N}$ denote the set of positive reals and positive integers, respectively. Let $[n] :=$ $\{1, 2, \ldots n\}$, for any $n \in \mathbb{N}$.

Consider a stochastic bandit problem with $K$ arms $[K] :=$ $\{1, \ldots, K\}$ and $M$ data sources $[M] := \{1, \ldots, M\}$. Each arm $i \in [K]$ has an unknown mean reward $\mu_i \in \mathbb{R}$, while each data source $j \in [M]$ has an unknown noise standard deviation $\sigma_j \in \mathbb{R}_+$ and a (possibly unknown) fourth central moment $\kappa_j \in \mathbb{R}_+$.

At each round $t \in [T]$, the learner selects an arm $i_t \in [K]$ and a source $j_t \in [M]$, then observes a reward $X_t = \mu_{i_t} + \varepsilon_t$, where $\varepsilon_t \sim \mathcal{D}(j_t)$ and $\mathcal{D}(j_t)$ is a mean-zero distribution with variance $\sigma_j^2$ and fourth central moment $\kappa_j$. Equivalently, for any arm $i$ at round $t$, $\mathbf{E}[(X_t - \mu_{i_t})^4 \mid j_t] = \kappa_{j_t}$ since $X_t - \mu_{i_t} = \varepsilon_t$.

**Assumption 2.1.** For all arms $i \in [K]$, $\mu_i \in [0, \bar{\mu}]$ and for all sources $t$, support$(\mathcal{D}_{j_t}) \subseteq [\bar{\eta}, \bar{\eta}]$.

Let $i^* := \arg\max_{i \in [K]} \mu_i$ denote the optimal arm and $j^* := \arg\min_{j \in [M]} \sigma_j$ denote the optimal source (without loss of generality, we assume $i^* = 1$ and $j^* = 1$). We let $\mu^* := \mu_{i^*}$ and $\sigma^* := \sigma_{j^*}$ denote, the largest mean reward across arms and the smallest standard deviation across sources respectively.

The learner's objective is to minimize the expected regret, defined as: $\text{Reg}_T := \mathbb{E}\left[\sum_{t=1}^{T}\left(\mu^* - \mu_{i_t}\right)\right]$.

**Impact of Source Selection on Regret.** Although the expected regret does not explicitly depend on the selected sources $j_t$, source selection has a critical indirect effect. Failure to identify and prioritize low-variance sources leads to noisier observations, slowing reward estimation, and inflating regret. Moreover, a simple two-step strategy that first identifies the lowest-variance source and then runs a standard MAB algorithm on it is generally suboptimal, as detailed in Section 3.

## 3. Baselines and Their Limitations

Before presenting SOAR, we motivate its design by examining two natural baseline algorithms for the heterogeneous multi-source bandit problem and exposing where they break down. As discussed in our related work (Section 1 and Appendix A), to the best of our knowledge, no prior work addresses our exact setting. Hence we construct two UCB-based baselines and analyze their limitations; these limitations directly motivate the simultaneous source-arm exploration strategy that SOAR adopts.

We start with a description of UUCB (Algorithm 1), analyze its regret and show how it degrades on adverse instances.

### 3.1. Baseline-1: Uniform UCB (UUCB)

**Algorithm Description: Uniform-UCB (UUCB).** This baseline (Algorithm 1) selects a data source uniformly at random at each round $t$ and uses the corresponding observations to update reward estimates and apply standard multi-armed bandit (MAB) routines. Under this uniform se-

*Table 1.* Summary of related work in sequential decision-making. SOAR is the only method that adaptively handles *multiple* sources with *unknown, heterogeneous* variances.

| Algorithm | Setting | Multiple sources | Unknown / hetero. variance | Real-data eval. |
|---|---|---|---|---|
| MF-UCB (Kandasamy et al., 2016) | Multi-fidelity $K$-armed bandit. | ✓[†] | ✗ | ✗ |
| MF-MI-Greedy (Song et al., 2019) | Multi-fidelity Bayesian optimization. | ✓[†] | ✗ | ✓ |
| UCB-V / PAC-UCB (Audibert et al., 2009) | Stochastic MAB, empirical variance. | ✗ | ✓ | ✗ |
| UGapE (Gabillon et al., 2012) | Best-arm identification, stochastic MAB. | ✗[‡] | ✓ | ✗ |
| Normal Bandits (Cowan et al., 2018) | Normal MAB, unknown mean & variance. | ✗ | ✓ | ✗ |
| **SOAR (ours)** | Multi-source MAB, heterogeneous noise. | ✓ | ✓ | ✓ |

[†] Fidelities, not heterogeneous-variance sources in our sense.     [‡] Extends to "multiple bandits," different from heterogeneous sources.

lection strategy, each arm $i \in [K]$ experiences "*an effective variance*" equal to the average variance across all sources: $\tilde{\sigma}^2 := \frac{1}{M} \sum_{j=1}^{M} \sigma_j^2$. Arm selection uses UCB-V (Audibert et al., 2009).

---

**Algorithm 1** Uniform-UCB (UUCB)
---
1: **Input:** Arms $[K]$, Sources $[M]$, conf. $\delta \in (0, 1)$.
2: **for** $t = 1, 2, \ldots, T$ **do**
3:     Sample source $j_t \sim \text{Unif}([M])$
4:     Select arm $i_t$ via UCB-V over arms
5:     Pull $(i_t, j_t)$, observe $X_t$, update estimates
6: **end for**
---

**Regret Analysis of UUCB:** Following standard upper confidence bound (UCB)-based analyses for MAB algorithms (Lattimore & Szepesvári, 2020) the regret of UUCB can be bounded as $O\left(\tilde{\sigma}\sqrt{KT}\right)$, or, more specifically, using instance-dependent regret bounds as $O\left(\tilde{\sigma}^2 \sum_{i \neq i^*} \frac{\log(MKT)}{\Delta_i}\right)$. However, this uniform averaging approach can be gravely suboptimal in certain instances.

**Worst Case Instance for UUCB (WC-1).** Consider now the worst-case scenario for UUCB (*worst-case instance 1 or WC-1*) where all sources except the optimal one share a similarly high variance $\sigma_{\max}^2$. The effective variance is then dominated by these noisy sources, $\tilde{\sigma}^2 = \frac{(M-1)\sigma_{\max}^2}{M} \approx \sigma_{\max}^2$, so the instance-dependent bound deteriorates to $O\left(\sigma_{\max}^2 \sum_{i \neq i^*} \frac{\log(MKT)}{\Delta_i}\right)$ and the instance-independent bound to $O(\sigma_{\max}\sqrt{KT})$, where $\Delta_i := \mu^* - \mu_i$ is the suboptimality gap of arm $i$. Thus UUCB accuracy is effectively governed by the noisiest sources.

### 3.2. Baseline-2: Explore-then-Commit UCB (ETC-UCB)

**Algorithm Description: ETC-UCB.** In this baseline (Algorithm 2), the learner first attempts to identify the data source with the smallest variance by fixing a single arm and running a best-arm identification (BAI) algorithm across the sources. After this initial phase, the learner commits exclusively to the identified "good" variance source and runs a standard MAB algorithm on the $K$ arms, querying arm rewards solely from this selected source.

---

**Algorithm 2** Explore-then-Commit UCB (ETC-UCB )
---
1: **Input:** Arm set $[K]$, Sources $[M]$, confidence $\delta \in (0, 1)$, tolerance $\epsilon$.
2: **Phase-1 (Source identification):** Fix an arm $i_0 \in [K]$; run best-arm identification over sources $[M]$ to find the lowest-variance source $\hat{j}$.
3: **Phase-2 (Arm exploitation):** Run a standard UCB over arms $[K]$, querying rewards only from source $\hat{j}$.
---

**Regret Analysis of ETC-UCB :** To analyze the regret incurred by this two-phase strategy, we note that the first phase (source identification) incurs a regret of approximately $O\left(\sum_{j \neq j^*} \frac{\bar{\mu} \log(MKT)}{\max\{\epsilon^2, (\Delta_j^{\sigma^2})^2\}}\right)$, where $\epsilon \in \mathbb{R}_+$ is a user-specified error tolerance parameter determining the desired precision (PAC-optimality) in identifying the minimal-variance source. Importantly, during each round of the first phase, the algorithm potentially incurs $O(\bar{\mu})$ since it continues to query a fixed arm and does not make progress in identifying the optimal reward arm $i^* \in [K]$.

In the second phase (arm exploitation), the algorithm's regret can be bounded as $O\left((\sigma^* + \epsilon)^2 \sum_{i \neq i^*} \frac{\log(MKT)}{(\Delta_i)}\right)$, leading to an overall regret (summing two phases) of:

$$O\left(\sum_{j \neq j^*} \frac{\bar{\mu} \log(MKT)}{\max\{\epsilon^2, (\Delta_j^{\sigma^2})^2\}} + (\sigma^* + \epsilon)^2 \sum_{i \neq i^*} \frac{\log(MKT)}{\Delta_i}\right)$$

We remark that $\epsilon$ is a tunable parameter whose choice significantly impacts the performance of this approach. However, it is hard for the learner to optimize the value of $\epsilon$ without the knowledge of the variance gaps $\{\Delta_j^{\sigma^2}\}_{j \in [M]}$.

**Worst Case Instance for ETC-UCB (WC-2).** Consider now the worst-case scenario for ETC-UCB (*worst-case in-*

*stance 2 or WC-2*), where all sources have roughly identical variances, $\sigma_1^2 \approx \sigma_2^2 \approx \cdots \approx \sigma_M^2$, so every gap $\Delta_j^{\sigma^2} \approx 0$. Here, distinguishing sources is pointless as the learner could commit to any one at random, yet ETC-UCB still wastes its source-identification phase trying to separate equally good sources, incurring approximately $O\left(\sum_{j \neq j^*} \frac{\bar{\mu} \log(MKT)}{\max\{\epsilon^2, (\Delta_j^{\sigma^2})^2\}}\right)$ regret. Worse, setting $\epsilon \to 0$ is disastrous: the factor $1/\max\{\epsilon^2, (\Delta_j^{\sigma^2})^2\}$ blows up, forcing the algorithm to keep sampling all sources to detect non-existent differences.

**No fixed $\epsilon$ is robust.** Since the learner does not know the minimal source variance $\sigma^*$ in advance, choosing an appropriate $\epsilon$ becomes challenging, and no instance-independent choice is safe.

Suppose the learner naively fixes $\epsilon = c_1 > 0$. Under WC-2, where $\Delta_j^{\sigma^2} \approx 0$ for all $j \neq j^*$, each Phase-1 denominator becomes $\max\{\epsilon^2, (\Delta_j^{\sigma^2})^2\} \approx \epsilon^2 = c_1^2$, so the regret contribution satisfies
$$\sum_{j \neq j^*} \frac{\bar{\mu} \log(MKT)}{\max\{\epsilon^2, (\Delta_j^{\sigma^2})^2\}} \approx \sum_{j \neq j^*} \frac{\bar{\mu} \log(MKT)}{c_1^2},$$
which grows unboundedly as $c_1 \to 0$. Conversely, an overly large $\epsilon$ inflates the Phase-2 term $(\sigma^* + \epsilon)^2 \sum_{i \neq i^*} \log(MKT)/\Delta_i$, through its $\epsilon^2$ factor, and the regret again scales poorly. Thus any fixed $\epsilon = c_1$ can be far from optimal across instances.

**Takeaway.** Both baselines fail because they decouple source and arm exploration. An effective algorithm must *simultaneously* and adaptively favor low-variance sources while identifying the best arm which is precisely the design SOAR adopts.

# 4. Main Algorithm: Source-Optimistic Adaptive Regret minimization (SOAR)

In this section, we present our proposed algorithm, SOAR, developing its estimation machinery and confidence bounds alongside the algorithmic procedure that uses them. At a high level, SOAR consists of two key components: (i) a preprocessing phase that eliminates "bad" or high variance sources, and (ii) an adaptive LCB–UCB procedure for joint source–arm selection. We first introduce some additional notation, then describe the two stages in turn, and finally establish SOAR's regret guarantees in comparison to the baselines of Section 3.

**Notation.** For any $t \in [T]$, define
$$n_i(t) := \sum_{s=1}^{t} \mathbb{1}\{i_s = i\}, \qquad m_j(t) := \sum_{s=1}^{t} \mathbb{1}\{j_s = j\},$$
$$n_{ij}(t) := \sum_{s=1}^{t} \mathbb{1}\{i_s = i\}\mathbb{1}\{j_s = j\}.$$
(1)

where $n_i(t)$ denotes the number of times arm $i$ is selected up to time $t$, $m_j(t)$ denotes the number of times source $j$ is selected up to time $t$, and $n_{ij}(t)$ denotes the number of times the pair $(i, j)$ is selected up to time $t$.

We also define three additional quantities used in the algorithm's analysis. For each arm $i \in [K]$, let $\Delta_i := \mu^* - \mu_i$ denote the suboptimality gap. For each source $j \in [M]$, let $\Delta_j^{\sigma} := \sigma_j - \sigma^*$ and $\Delta_j^{\sigma^2} := \sigma_j^2 - \sigma^{*2}$ denote its excess standard deviation and excess variance relative to the most reliable source, respectively.

**A Brief Description of SOAR.** SOAR proceeds in two stages, summarized in Figure 1: (1) the preprocessing subroutine PREPROCESS, which constructs a high-probability upper bound on the variances using its own variance estimator and confidence bounds, and (2) the adaptive UCB–LCB selection phase, which begins with an initial exploration step and relies on a separate estimation-and-confidence scheme. We now develop each stage with its estimators and bounds.

## 4.1. Stage 1: Preprocessing to Remove "Bad" Sources

PREPROCESS is a preprocessing subroutine that eliminates high-variance sources. Each source is queried a fixed number of times (the runtime budget $\tau_p$) on the same arm, variance confidence intervals (LCB/UCB) are constructed, and a source is removed if its LCB exceeds the minimum UCB across all sources. The surviving sources form the pruned set $S_G \subseteq [M]$, with cardinality $\tilde{M} := |S_G|$, which is passed to Stage (ii); the procedure is outlined in Algorithm 3.

Intuitively, sources with variance much larger than the minimum only add noise without reducing regret. Concretely, PREPROCESS removes any source with $\sigma_j^2 > \sigma^{*2} + c^{*2}$ and ensures all surviving sources satisfy $\sigma_j^2 - \sigma_*^2 < c^{*2}$, where $c^*$ is a user-chosen tolerance parameter of the order of $\sigma_j$. *Remark* 4.1. Intuitively, $c^{*2}$ serves as a variance floor: sources with $\sigma_j^2 < c^{*2}$ are treated as having effective variance $c^{*2}$ (handled as a separate case in our analysis in Theorem 5.4), while our main regret analysis focuses on the regime $\exists j \in S_G$ such that $\sigma_j^2 \geq c^{*2}$. This design avoids issues arising from extremely low-variance sources. The role of $c^*$ is further discussed in Remark 5.2.

This preprocessing simplifies the variance landscape, leading to tighter regret bounds and more efficient exploration. We next introduce the estimators and confidence bounds it relies on, before stating the algorithm and its key guarantee.

**Parameter Estimation.** The empirical mean reward of arm $i$, and the empirical variance of source $j$ *computed during* PREPROCESS, are given by
$$\hat{\mu}_i(t) := \frac{1}{n_i(t)} \sum_{s \leq t: i_s = i} X_s,$$
$$\hat{\sigma}_{j,\text{pre}}^2(t) := \frac{1}{m_j(t)} \sum_{s \leq t: j_s = j} \left(X_s - \hat{\mu}_{i_s}(t)\right)^2.$$
(2)

**Stage 1: Source pruning**

**Stage 2: Adaptive selection**

*Figure 1.* Overview of SOAR. **Stage 1** (PREPROCESS) prunes high-variance sources, keeping a surviving set of sources, $\mathcal{S}_G = \{j : \sigma_j^2 - \sigma_*^2 < c^{*2}\}$. **Stage 2** first samples each arm and surviving source enough times for the reward UCBs and variance LCBs to hold with high probability, then adaptively pulls the arm with the highest reward UCB and the source with the lowest variance LCB, repeating to horizon $T$.

**Confidence Bounds.** For each arm $i \in [K]$ and each source $j \in [M]$, we define the UCB and the LCB on the variance estimate $\sigma_{t,\text{pre}}^2(j)$. Our bounds rely on the following concentration lemma.

**Lemma 4.2** (PREPROCESS Variance Concentration). *Fix $\delta \in (0,1)$ and a sampling budget $\tau_p \in \mathbb{N}$. Assume $\epsilon'' < \min\left\{6\sigma_j^2, \frac{18\sigma_j^4}{\bar{\eta}^2}\right\}$ for each source $j \in [M]$, where $\epsilon''$ is the parameter appearing in Bernstein's Inequality (Boucheron et al., 2013; Zhou, 2020). Then, with probability at least $(1 - \delta/3)$,*

$$\left|\hat{\sigma}_{j,\text{pre}}^2(\tau_p) - \sigma_j^2\right| \leq 8\bar{\eta}\,\sigma_j\sqrt{\frac{\log(4M/\delta)}{\tau_p}}.$$

*Proof.* The proof is deferred to Appendix B.1. □

Using Lemma 4.2 we can derive the following equations:

$$\text{UCB}_{\tau_p}^{\sigma^2,\text{pre}}(j) \coloneqq \hat{\sigma}_{j,\text{pre}}^2(\tau_p) + 8\bar{\eta}^2\sqrt{\frac{\log(12M/\delta)}{\tau_p}}, \quad (3)$$

$$\text{LCB}_{\tau_p}^{\sigma^2,\text{pre}}(j) \coloneqq \max\left\{\hat{\sigma}_{j,\text{pre}}^2(\tau_p) - 8\bar{\eta}^2\sqrt{\frac{\log(12M/\delta)}{\tau_p}}, 0\right\}. \quad (4)$$

It remains to specify how large the runtime budget $\tau_p$ must be for PREPROCESS to succeed, which the following result (Theorem 4.3) establishes.

**Theorem 4.3** (Stopping Condition of PREPROCESS). *Consider any $\delta \in (0,1)$. If PREPROCESS is run with runtime budget of at least $\tau_p$, where $\tau_p \geq \frac{1024\bar{\eta}^4 \log(12M/\delta)}{c^{*4}}$, where $c^* \in \mathbb{R}_+$ is a user defined algorithm parameter of the order of $\sigma_j$, then any source $j \in [M]$ with variance $\sigma_j^2 > \sigma^{*2} + c^{*2}$, will be eliminated with probability $(1 - \delta/3)$.*

*Proof.* The proof is provided in Appendix B.2. □

### 4.2. Stage 2: The Adaptive UCB-LCB Selection Phase

Having pruned the high-variance sources, we now turn to how SOAR makes decisions over the surviving set $S_G$. This phase reasons jointly about arm rewards and source variances, which calls for a different set of estimators and confidence bounds, developed below.

**Parameter Estimators.** PREPROCESS (Algorithm 3) queries sources using a single fixed arm, rather than multiple arms as in the adaptive UCB–LCB phase. This motivates a separate set of **unbiased pooled** estimators of the arm means and source variances, defined below.

$$\hat{\mu}_i(t) \coloneqq \frac{1}{n_i(t)} \sum_{s \leq t : i_s = i} X_s,$$

$$\hat{\mu}_{ij}(t) \equiv \hat{\mu}_i(\tau_{i,j}(t)) \coloneqq \frac{1}{n_{ij}(t)} \sum_{\ell \in \tau_{i,j}(t)} X_\ell,$$

$$\hat{\sigma}_{ij}^2(t) \coloneqq \frac{1}{n_{ij}(t) - 1} \sum_{\ell \in \tau_{i,j}(t)} \left(X_\ell - \hat{\mu}_{ij}(t)\right)^2, \quad (5)$$

$$\hat{\sigma}_j^2(t) \coloneqq \frac{\sum_{i \in [K]} (n_{ij}(t) - 1)\hat{\sigma}_{ij}^2(t)}{\sum_{i=1}^K n_{ij}(t) - K}$$

$$= \frac{\sum_{i \in [K]} \sum_{\ell \in \tau_{i,j}(t)} \left(X_\ell - \hat{\mu}_{ij}(t)\right)^2}{m_j(t) - K}.$$

Here, $\tau_{i,j}(t) \coloneqq \{\ell \leq t \mid j_\ell = j, \ i_\ell = i\}$ for $j \in S_G$ and $i \in [K]$ denotes the index set of time steps up to $t$ in which arm $i$ is pulled in combination with source $j$, and its cardinality is $|\tau_{i,j}(t)| = n_{ij}(t)$.

### A Brief Description of the Estimators.

- $\hat{\mu}_{ij}(t)$ is the empirical mean reward of arm $i$ when queried specifically through source $j$.
- $\hat{\sigma}_{ij}^2(t)$ is the corresponding unbiased sample variance estimate for the pair $(i,j)$ computed from the same restricted observations.
- $\hat{\sigma}_j^2(t)$ is the pooled (unbiased) variance estimate of source $j$, obtained by aggregating the sample variances $\hat{\sigma}_{ij}^2(t)$ across all arms and normalizing by the total degrees of freedom.

**Algorithm 3** PREPROCESS: Source Pruning

---

1: **Input:** Arm set: $[K]$, Feedback Sources: $[M]$, Confidence parameter: $\delta \in (0,1)$, Runtime budget: $\tau_p \in \mathbb{N}_+$.

2: **Init:** $S_{\mathcal{G}} \leftarrow [M]$. Fix any arm $i_0 \in [K]$
3: **for** $j \in [M]$ **do**
4:      Query source $j$ for Arm-$i_0$, $\tau_p$ times
5: **end for**
6: Compute $\text{UCB}_{\tau_p}^{\sigma^2,\text{pre}}(j)$ and $\text{LCB}_{\tau_p}^{\sigma^2,\text{pre}}(j)$ for all $j \in [M]$ using Equation (3) and Equation (4)
7: $m \leftarrow \min_{j \in [M]} \text{UCB}_{\tau_p}^{\sigma^2,\text{pre}}(j)$
8: **for** $j \in [M]$ such that $\text{LCB}_{\tau_p}^{\sigma^2,\text{pre}}(j) > m$ **do**
9:      $S_{\mathcal{G}} \leftarrow S_{\mathcal{G}} \setminus \{j\}$      *// Eliminate variance source*
10: **end for**
11: Return $S_{\mathcal{G}}$      *// Pruned set of sources*

---

**Confidence Bound on the Variance Estimate.** Recall from Section 2 that for each source $j \in [M]$, the fourth central moment is denoted by $\kappa_j$. We now define

$$Q_j(t) = \begin{cases} \max\{\kappa_j, \nu\} & \text{if } \kappa_j \text{ is known} \\ \max\{\bar{\eta}^2 \hat{\sigma}_j^2(t), \nu\} & \text{if } \kappa_j \text{ is unknown} \end{cases} \quad (6)$$

where $\nu \in \mathbb{R}_+$ is a user-defined algorithm parameter chosen to be on the same scale as the fourth central moments $\{\kappa_j\}_{j \in [M]}$, and serves a role similar to $c^{*2}$ (see Remark 4.1). Armed with this knowledge we now define the LCB for each source $j \in S_{\mathcal{G}}$ on the variance estimate $\hat{\sigma}_j^2(t)$. Our bound relies on the following concentration lemma.

**Lemma 4.4** (Source Variance Concentration). *Assume there exists a source $j \in S_{\mathcal{G}} \subseteq [M]$ such that $\sigma_j^2 \geq c^{*2}$ where $c^* \in \mathbb{R}_+$ is a user-defined parameter of the order of $\sigma_j$. Recall $Q_j(t)$ as defined in Equation (6) where $\nu \in \mathbb{R}_+$ is a user-defined parameter chosen on the scale of fourth central moment $\kappa_j$. If the source $j$, is queried for atleast $m_j(t)$ iterations where $m_j(t) \geq K + \frac{4\bar{\eta}^4 \log(3MT/\delta)}{\nu}$, then for any $t \in [T]$, with probability at least $1 - \delta/3$,*

$$\left| \hat{\sigma}_j^2(t) - \sigma_j^2 \right| \leq \sqrt{\frac{Q_j(t) \log(3MT/\delta)}{m_j(t) - K}}. \quad (7)$$

*Proof.* The proof is deferred to Appendix C.1      □

Using Lemma 4.4 we can derive the following equation:

$$\text{LCB}_t^{\sigma^2}(j) \coloneqq \hat{\sigma}_j^2(t) - 2\sqrt{\frac{Q_j(t) \log(3MT/\delta)}{m_j(t) - K}} \quad (8)$$

**Confidence Bound on the Mean Reward Estimate.** For each arm $i \in [K]$ and each source $j \in S_{\mathcal{G}} \subseteq [M]$, we can derive an UCB on the empirical mean reward $\hat{\mu}_t(t)$.

Before stating the UCB, we note that the bound relies on the following concentration corollary and concentration lemma.

**Corollary 4.5** (Variance Sandwiching). *Consider any $\delta \in (0,1)$. Assume there exists a source $j \in S_{\mathcal{G}} \subseteq [M]$ such that $\sigma_j^2 \geq c^{*2}$ where $c^* \in \mathbb{R}_+$ is a user-defined parameter of the order of $\sigma_j$. If such a source $j$ is queried at least $m_j(t)$ times where $m_j(t) \geq K + \frac{16\bar{\eta}^4 \log(3MT/\delta)}{c^{*4}}$, then we have with probability $(1 - \delta/3)$*

$$\sigma_j^2 \leq 2\hat{\sigma}_j^2(t) \leq 3\sigma_j^2.$$

*Proof.* The proof is deferred to Appendix C.2      □

**Lemma 4.6** (Mean Reward Concentration). *Consider any $\delta \in (0,1)$. Assume there exists a source $j \in S_{\mathcal{G}} \subseteq [M]$ such that $\sigma_j^2 \geq c^{*2}$ and $\text{LCB}_{\tau_p}^{\sigma^2,\text{pre}}(j) \geq \frac{c^{*2}}{2}$, where $c^* \in \mathbb{R}_+$ is a user-defined parameter of the order of $\sigma_j$. If an arm $i \in K$, is queried for at least $\alpha$ iterations with this source $j$ where $\alpha \geq \frac{\bar{\eta}^2 \log(3KT/\delta)}{c^{*2}}$ then for any $t \in [T]$, we have, with probability at least $1 - \delta/3$,*

$$|\mu_i - \hat{\mu}_i(t)| \leq \frac{2\sqrt{\log(3KT/\delta) \sum_{j=1}^{M} n_{ij}(t) \sigma_j^2}}{n_i(t)}. \quad (9)$$

*Proof.* The proof is deferred to Appendix C.3      □

Using Lemma 4.6 and Corollary 4.5 we obtain:

$$\text{UCB}_t^{\mu}(i) \coloneqq \hat{\mu}_i(t) + \frac{2\sqrt{2\log(3KT/\delta) \sum_{j=1}^{M} n_{ij}(t) \hat{\sigma}_j^2(t)}}{n_i(t)} \quad (10)$$

The UCB and LCB quantities defined above enjoy standard high-probability coverage guarantees. Although these results are direct consequences of existing concentration inequalities, we record them as Corollary C.6 and Corollary C.7 in Appendix C.4 for completeness.

### 4.3. Putting it Together: SOAR

We now assemble the two stages into the full procedure, SOAR (Algorithm 4). After preprocessing prunes the source set, an initial exploration phase guarantees that each arm and surviving source is sampled often enough for the confidence bounds developed above to hold; the algorithm then enters its adaptive LCB/UCB loop for joint source–arm selection.

**Initial Exploration.** After PREPROCESS concludes, we identify a source $\bar{j}$ such that

$$\text{LCB}_{\tau_p}^{\sigma^2,\text{pre}}(\bar{j}) \geq \frac{c^{*2}}{2}.$$

The existence of such a source is guaranteed under the regime assumption on $c^*$ stated in Remark 4.1 and formally

---

**Algorithm 4** SOAR: Source-Optimistic Adaptive Regret

---

1: **Input:** Arm set $[K]$, Sources $[M]$, confidence $\delta \in (0,1)$, exploration parameters $\alpha, \beta \in \mathbb{N}_+$, preprocessing budget $\tau_p \in \mathbb{N}_+$, algorithm parameters $c^*, \nu \in \mathbb{R}_+$.

2: Calculate $\tau_p$ from Theorem 4.3.

3: $S_{\mathcal{G}} \leftarrow \text{PREPROCESS}([M], [K], \delta/3, \tau_p); \quad \tilde{M} \leftarrow |S_{\mathcal{G}}|$

4: **Initial Exploration:**

5: Identify $\bar{j} \in S_{\mathcal{G}}$ with $\text{LCB}_{\tau_p}^{\sigma^2, \text{pre}}(\bar{j}) \geq \frac{c^{*2}}{2}$.

6: Query each arm $i \in [K]$ via source $\bar{j}$ for $\alpha$ rounds (Equation (11))

7: Fix $i_0 \in [K]$; query each $j \in S_{\mathcal{G}}$ via arm $i_0$ for $\beta$ rounds (Equation (12))

8: **Adaptive Selection:**

9: **for** $t = M\tau_p + \tilde{M}\beta + K\alpha + 1, \ldots, T$ **do**

10: $\quad i_t \leftarrow \arg\max_{i \in [K]} \text{UCB}_{t-1}^{\mu}(i);$

11: $\quad j_t \leftarrow \arg\min_{j \in S_{\mathcal{G}}} \text{LCB}_{t-1}^{\sigma^2}(j)$

12: $\quad$ Pull $(i_t, j_t)$, observe reward $X_t$.

13: $\quad$ Update counts (Equation (1)).

14: $\quad$ Update estimators $\hat{\mu}_i(t), \hat{\sigma}_j^2(t)$ (Equation (5))

15: $\quad$ Update bounds $\text{UCB}_t^{\mu}(i), \text{LCB}_t^{\sigma^2}(j)$ (Equation (10), Equation (8))

16: **end for**

---

proven in Appendix C.3. We then pull each arm $i \in [K]$ using source $\bar{j}$ at least $\alpha$ times, where

$$\alpha \geq \frac{\bar{\eta}^2 \log(3KT/\delta)}{c^{*2}}. \tag{11}$$

This ensures that the confidence bounds and estimates in Lemma 4.6 hold with high probability.

Similarly, we query every surviving source $j \in S_{\mathcal{G}} \subseteq [M]$ at least $\beta$ times using any fixed arm $i_0 \in [K]$, where

$$\beta \geq 2K + \frac{4\bar{\eta}^4 \log(3MT/\delta)}{\nu} + \frac{16\bar{\eta}^4 \log(3MT/\delta)}{c^{*4}}. \tag{12}$$

This ensures that the confidence bounds and estimates in Lemma 4.4 and Corollary 4.5 hold with high probability. In particular, sampling each surviving source $\beta$ times relates the empirical variance $\hat{\sigma}_j^2(t)$ to the true $\sigma_j^2$ via the sandwiching bound of Corollary 4.5, which lets the mean-reward UCB (Equation (10)) be formed from the empirical variance rather than the unknown true variance.

**Adaptive Selection.** From round $t = M\tau_p + \tilde{M}\beta + K\alpha + 1$ onward, the algorithm adaptively selects the arm $i_t$ with the largest upper confidence bound on mean reward and the source $j_t$ with the smallest lower confidence bound on variance, then queries the pair $(i_t, j_t)$, updates counts, and recomputes estimates and bounds. Intuitively, this UCB–LCB mechanism balances optimistic exploration of arms with

cautious, variance-aware selection of sources, guiding the learner toward rewarding arms and low-noise feedback. In Section 5, we show that this strategy yields regret nearly matching that of an oracle with access to the optimal low-variance source, formalizing the effectiveness of our simultaneous exploration–exploitation design.

# 5. Regret Analysis of SOAR:

In this section, we present the main theoretical result in Theorem 5.1, establishing a near-optimal regret guarantee for SOAR. The analysis introduces one additional parameter $\gamma \in \mathbb{R}_+$ (on the scale of $\sigma$), used only in the bound and not in the algorithm. As discussed in Remark 5.2, the bound matches, up to logarithmic factors, the instance-dependent regret of an Oracle MAB with prior knowledge of the optimal source variance. We further compare it with the baselines (Remark 5.3), present a result for the alternate regime (Theorem 5.4) and show how the dependence on $c^*$ can be removed via an adaptive choice (Remark 5.5).

**Theorem 5.1** (Main Result: Regret Analysis of SOAR). *Consider $\delta \in (0,1)$. Assume there exists a source $j \in S_{\mathcal{G}} \subseteq [M]$ such that $\sigma_j^2 \geq c^{*2}$ where $c^*, \gamma \in \mathbb{R}_+$ are user-defined parameters of the order of $\sigma_j$. Recall $Q_j(t)$ as defined in Equation (6) where $\nu \in \mathbb{R}_+$ is a user-defined parameter chosen on the scale of fourth central moment $\kappa_j$.*

*Then for any choice of preprocessing budget $\tau_p \geq \frac{1024 \cdot \bar{\eta}^4 \log(12M/\delta)}{c^{*4}}$, initial-exploration parameters $\alpha \geq \frac{\bar{\eta}^2 \log(3KT/\delta)}{c^{*2}}$, $\beta \geq 2K + \frac{4\bar{\eta}^4 \log(3MT/\delta)}{\nu} + \frac{16\bar{\eta}^4 \log(3MT/\delta)}{c^{*4}}$, the regret of SOAR (Algorithm 4) can be bounded by $\tilde{O}\Big( \frac{M\bar{\eta}^4\bar{\mu}}{c^{*4}} + \frac{K\bar{\eta}^2\bar{\mu}}{c^{*2}} + KM\bar{\mu} + \frac{M\bar{\eta}^4\bar{\mu}}{\nu} +$*

$$\sqrt{K}\sqrt{KMc^{*2} + \sum_{j: \sigma_j - \sigma^* > \gamma} \frac{Q_j(T)}{(\gamma + 2\sigma^*)^2} + \sum_{i=2}^{K} \frac{(\sigma^* + \gamma)^2}{\Delta_i}}\Big).[1]$$

*with high probability $(1 - \delta)$. SOAR can also be shown to yield an instance-independent (worst-case) regret bound of*

$$\tilde{O}\Big( \frac{M\bar{\eta}^4\bar{\mu}}{c^{*4}} + \frac{K\bar{\eta}^2\bar{\mu}}{c^{*2}} + KM\bar{\mu} + \frac{M\bar{\eta}^4\bar{\mu}}{\nu} + (\sigma^* + c^*)\sqrt{KT} \Big)$$

*Proof.* The proof is deferred to Appendix D.1. □

Notably, our regret analysis shows only a negligible dependence on source variances, a consequence of the LCB–UCB selection mechanism that swiftly prioritizes lower-variance sources while maintaining aggressive reward exploration.

*Remark* 5.2 (Implication of Theorem 5.1).

1. **Dominant Term and Optimality:** From the instance-dependent regret bound in Theorem 5.1, observe that when $\bar{\eta}, \bar{\mu} = O(1)$—a standard and reasonable

---

[1]Throughout the paper, the $\tilde{O}(\cdot)$ notation suppresses polylogarithmic factors in the problem parameters.

assumption under bounded-moment noise models—preprocessing, exploration and estimation contribute only $O(\log T)$ (or $T$-independent) regret, while the final term $\sum_{i=2}^{K} \frac{(\sigma^{*}+\gamma)^2}{\Delta_i}$ captures the instance difficulty. Consequently, as $T$ increases, the regret is asymptotically dominated by the final term.

Additionally since $\gamma$ can be chosen arbitrarily small ($O(1)$ as per discussion below), this simplifies to $\tilde{O}\left(\sum_{i=2}^{K} \frac{\sigma^{*2}}{\Delta_i}\right)$, This is in fact near-optimal: the standard $K$-armed MAB problem admits a lower bound of $\Omega\left(\sum_{i \neq i^*} \frac{1}{\Delta_i}\right)$ (Auer et al., 2002), which, scaled by variance, becomes $\Omega\left(\sum_{i \neq i^*} \frac{\sigma^2}{\Delta_i}\right)$ for a single source. In the multi-source setting, any algorithm cannot do better than $\Omega\left(\sum_{i \neq i^*} \frac{\sigma^{*2}}{\Delta_i}\right)$ by a standard reduction to the single-source case with access to the best source. SOAR's dominant term matches this lower bound up to logarithmic factors, achieving oracle-optimal regret despite not knowing $\sigma^{*}$.

2. **Role of the Parameter $\gamma$.** The parameter $\gamma > 0$ stabilizes the variance-dependent confidence bounds in degenerate regimes where $\sigma^{*} \to 0$. In typical settings, $\sigma^{*}$ is bounded away from zero, and $\gamma$ can be chosen as a small constant (e.g., $\gamma \approx O(10^{-p})$ for $p \in \{2, 3\}$) which does not affect the dominant regret term.

3. **Known Fourth Central Moments $\kappa_j$:** When the fourth central moments $\kappa_j$ are known, the quantity $Q_j(t)$ simplifies to $Q_j(t) = \max\{\kappa_j, \nu\}$ Substituting this into Theorem 5.1, the variance-separation term becomes $\sum_{j:\, \sigma_j - \sigma^* > \gamma} \frac{\max\{\kappa_j, \nu\}}{(\gamma + 2\sigma^*)^2}$.

   In this case, choosing $\nu$ smaller than the smallest $\kappa_j$ does not improve the regret bound while increasing the preprocessing cost (as $\beta \propto \nu^{-1}$). On the other hand increasing $\nu$ inflates the variance separation term. Thus, when $\kappa_j$ are known, it is natural to choose $\nu \approx \min_j \kappa_j$, yielding the tightest bound.

4. **Gaussian Noise:** In particular, for Gaussian rewards where $\kappa_j = 3\sigma_j^4$, the variance-separation contributes $\sqrt{K}\sqrt{KMc^{*2} + \sum_{j:\, \sigma_j - \sigma^* > \gamma} \frac{3\sigma_j^4}{(\gamma + 2\sigma^*)^2}}$ to the regret. Note that this contribution is independent of $T$ (for fixed problem parameters), and is therefore asymptotically dominated by the leading instance-dependent term $\sum_{i=2}^{K} \frac{(\sigma^{*}+\gamma)^2}{\Delta_i}$.

5. **Unknown Fourth Moments $\kappa_j$:** When $\kappa_j$ are unknown, $Q_j(t)$ is defined as $Q_j(t) = \max\{\bar{\eta}^2 \hat{\sigma}_j^2(t), \nu\}$ where $\nu > 0$ acts as a floor ensuring stability of variance-dependent confidence bounds.

   In this setting, the choice of $\nu$ induces an explicit tradeoff. On one hand, a larger $\nu$ increases the numerator of the regret bound through the term $\sum_{j:\, \sigma_j - \sigma^* > \gamma} \frac{Q_j(T)}{(\gamma + 2\sigma^*)^2}$, potentially inflating the regret when empirical variance estimates are small. On the other hand, $\nu$ appears in the denominator of the exploration parameter $\beta$ so decreasing $\nu$ increases the required exploration budget .

Notably, this tradeoff mirrors that of the parameter $c^*$, which controls the balance between reliable variance separation and exploration cost; in practice, both $\nu$ and $c^*$ can be chosen as small constants ($\sim O(1)$) without affecting the leading-order regret.

Together, these observations show that SOAR attains oracle-optimal instance-dependent regret up to logarithmic factors, despite unknown variances and higher-order moments.

*Remark* 5.3 (Improved Regret Bound and Comparison to Baselines). Comparing the regret guarantee in Theorem 5.1 with the baselines analyzed in Section 3, we find that SOAR improves over both UUCB (Baseline-1) and ETC-UCB (Baseline-2) under the worst-case instances **WC-1** and **WC-2**.

Consequently, SOAR incurs an instance-dependent regret of $\tilde{O}\Big( \frac{M\bar{\eta}^4\bar{\mu}}{c^{*4}} + \frac{K\bar{\eta}^2\bar{\mu}}{c^{*2}} + KM\bar{\mu} + \frac{M\bar{\eta}^4\bar{\mu}}{\nu} +$

$\sqrt{K}\sqrt{KMc^{*2} + \sum_{j:\, \sigma_j - \sigma^* > \gamma} \frac{Q_j(T)}{(\gamma + 2\sigma^*)^2} + \sum_{i=2}^{K} \frac{(\sigma^*+\gamma)^2}{\Delta_i}}\Big)$.

In contrast, Baseline-1:UUCB suffers, under **WC-1**, an instance-dependent regret of $\tilde{O}\left( \sigma_{\max}^2 \sum_{i \neq i^*} \frac{\log(MKT)}{\Delta_i} \right)$, and Baseline-2:ETC-UCB incurs $\tilde{O}\left( \sum_{j \neq j^*} \frac{\bar{\mu}}{(\Delta_j^{\sigma^2})^2} + \sigma^{*2} \sum_{i \neq i^*} \frac{1}{\Delta_i} \right)$ whose first term can blow up under **WC-2** when the variance gaps $\{\Delta_j^{\sigma^2}\}$ are very small.

Moreover, SOAR attains a worst-case (instance-independent) regret of $\tilde{O}\Big( \frac{M\bar{\eta}^4\bar{\mu}}{c^{*4}} + \frac{K\bar{\eta}^2\bar{\mu}}{c^{*2}} + KM\bar{\mu} + \frac{M\bar{\eta}^4\bar{\mu}}{\nu} + (\sigma^* + c^*)\sqrt{KT}\Big)$ compared to Baseline-1:UUCB which under **WC-1** has worst-case regret $\tilde{O}\left( \sigma_{\max}\sqrt{KT} \right)$, and Baseline-2:ETC-UCB which has a worst-case regret of $\tilde{O}\left( \sum_{j \neq j^*} \frac{\bar{\mu}}{(\Delta_j^{\sigma^2})^2} + \sigma^*\sqrt{KT} \right)$.

Unlike UUCB, whose regret scales with the average variance and can be dominated by multiple high-variance sources, SOAR adaptively limits their effect. Similarly, ETC-UCB incurs unnecessary regret when source variances are nearly identical, as it dedicates a full phase to

distinguishing them; by balancing arm and source exploration, SOAR avoids this inefficiency. Corresponding plots verifying these claims appear in Section 6.

We next analyze the complementary regime in which all surviving sources have variance below the threshold $c^{*2}$, and establish the corresponding regret guarantees for SOAR.

**Theorem 5.4** (Regret Analysis of SOAR in alternate regime). *Consider $\delta \in (0, 1)$. Assume that all sources $j \in S_{\mathcal{G}} \subseteq [M]$ are such that $\sigma_j^2 < c^{*2}$ where $c^* \in \mathbb{R}_+$ is a user-defined parameter of the order of $\sigma_j$.*
*Then for any choice of preprocessing budget $\tau_p \geq \frac{1024 \cdot \bar{\eta}^4 \log(12M/\delta)}{c^{*4}}$, inital exploration parameter $\alpha \geq \bar{\eta}^2 c^{*2} \log(3KT/\delta)$, then the regret of SOAR (Algorithm 4) can be bounded by $\tilde{O}\left(\frac{M\bar{\eta}^4 \bar{\mu}}{c^{*4}} + K\bar{\eta}^2 c^{*2}\bar{\mu} + \sum_{i=2}^{K} \frac{c^{*2}}{\Delta_i}\right)$ with high probability $(1 - \delta)$.*
*In this regime, SOAR can also be shown to yield an instance-independent (worst-case) regret bound of*

$$\tilde{O}\left(\frac{M\bar{\eta}^4\bar{\mu}}{c^{*4}} + K\bar{\eta}^2 c^{*2}\bar{\mu} + Kc^{*2}\sqrt{T}\right)$$

*Proof.* The proof is deferred to Appendix D.2. □

We now propose how the dependence on $c^*$ in Theorem 5.1 can be removed in Remark 5.5.

*Remark* 5.5 (Adaptive choice of $c^*$). The dependence on $c^*$ can be removed by replacing it with the adaptive proxy $\hat{\sigma}^{*2} := \min_{j \in [M]} \text{LCB}_{\tau_p}^{\sigma^2, \text{pre}}(j)$, formed from the PREPROCESS variance LCBs (Equation (4)). Substituting $\hat{\sigma}^{*2}$ for $c^*$ in Theorem 5.1 preserves the dominant $\sigma^*\sqrt{KT}$ rate, with a small floor $c^*_{\text{opt}} = O(T^{-1/10})$ handling the degenerate regime $\sigma^* \to 0$; see Appendix E for details.

## 6. Experiments

In this section, we evaluate SOAR on synthetic benchmarks and a real-world MovieLens panel. Arm rewards are Gaussian with fixed means and source-dependent variances, and shaded regions in the plots denote $95\%$ confidence intervals; in every setup, regret grows linearly at first, owing to the preprocessing and initial-exploration phases.

Our synthetic results are deferred to Appendix F: they cover scaling with the number of arms (Figure 4) and sources (Figure 5), as well as worst-case instances in which SOAR outperforms both baselines—decisively under WC-1 (against UUCB, Figure 6) and WC-2 (against ETC-UCB, Figure 7). We focus here on the real-world MovieLens panel.

**MovieLens Panel** To illustrate source adaptivity on real ratings, we build a small fixed panel from the Movie-Lens 32M dataset (Harper & Konstan, 2015): we select 15 reviewers who rated the same 500 movies (7,500 ratings in total). We model each movie as an arm $i$ with an unknown

mean reward $\mu_i \in [0, 5]$, corresponding to its mean rating in the panel. For each reviewer, we compute the variance of their ratings and take it as the source variance.

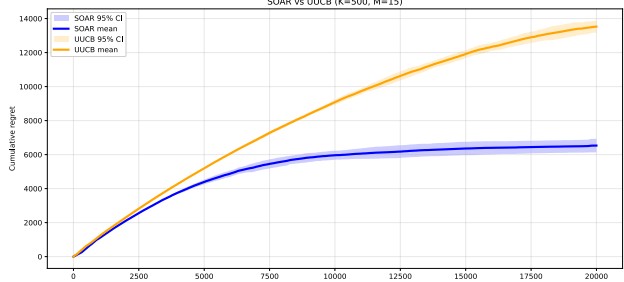

*Figure 2.* SOAR vs. Baseline-1: UUCB on MovieLens.

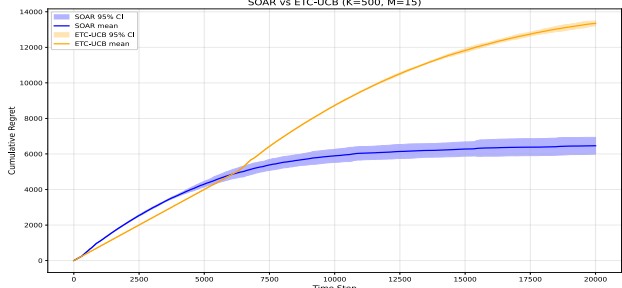

*Figure 3.* SOAR vs. Baseline-2:ETC-UCB on MovieLens.

**Findings.** On the MovieLens panel, SOAR rapidly concentrates on the lowest-variance reviewer while converging to the highest-rated movie, achieving substantially lower cumulative regret than both baselines (Figures 2–3). This aligns with our theory: SOAR preferentially routes pulls to low-variance sources.

## 7. Discussion

We studied the problem of online learning with multiple data sources under heteroscedastic noise, aiming to minimize regret through adaptive source and arm selection. Our proposed algorithm, SOAR, simultaneously explores and exploits both the data sources (with varying unknown variances) and the reward-generating arms, significantly improving over conventional baselines.

**Future Scope.** A compelling direction for future research is to extend our adaptive learning framework to more general contextual bandit or reinforcement learning scenarios. Additionally, developing algorithms that gracefully handle non-stationary environments, where the quality of data sources changes over time, remains an open challenge that would substantially broaden the applicability of our adaptive learning framework. Extending the framework to other feedback models, e.g. demonstration or relative feedback, might also be interesting future work.

## Acknowledgements

We thank Thomas Kleine Buening (ETH AI Center, Zurich, Switzerland) for early discussions that contributed to the LCB-UCB idea behind SOAR. We also thank the anonymous reviewers for their thoughtful feedback, which substantially improved the paper; in particular, the discussion on adaptively selecting $c^*$ (Remark 5.5, Appendix E) was prompted by reviewer suggestions.

## Impact Statement

This paper presents work whose goal is to advance the field of Machine Learning. There are a few potential societal consequences of our work, none of which we feel must be specifically highlighted here.

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

## Appendix:

## A. Problem Motivation, Baselines and Related Work

**Problem Motivation:**  Online learning, particularly multi-armed bandit (MAB) problems, has proven foundational for decision-making in uncertain and sequential environments. A crucial yet understudied scenario within this paradigm arises when learners can access multiple heterogeneous data sources, each characterized by unknown and distinct levels of noise. Such problems are prevalent in numerous real-world applications, including clinical trials (where different hospitals or laboratories provide data with varying degrees of reliability), recommender systems (with user segments differing in the quality of their feedback), and online advertising platforms (where different advertising channels generate outcomes with heterogeneous variance levels). Successfully managing these multiple data sources while simultaneously identifying the most rewarding actions (arms) is critical for minimizing cumulative regret and maximizing overall system performance. The motivation for studying this problem lies in its practical significance and inherent complexity. Decision-makers frequently face trade-offs between cheaper yet noisy data sources and more expensive yet reliable ones.

This scenario naturally aligns with the literature on multi-fidelity learning and variance-adaptive bandits. Multi-fidelity bandits (e.g., Kandasamy et al., 2016 (Kandasamy et al., 2016), Song et al., 2019 (Song et al., 2019)) typically involve leveraging low-cost, low-accuracy models to accelerate learning with expensive, high-accuracy sources. While related, our setting notably differs in that we focus explicitly on adaptive exploration across multiple sources characterized by differing noise variances rather than explicit cost-accuracy trade-offs. Variance-adaptive bandits (see, e.g., Audibert et al (Audibert et al., 2009), Cowan and Katehakis, (Cowan et al., 2018)) adapt the exploration rate based on observed variance within a single data-generating process. However, these existing approaches typically do not address scenarios involving multiple parallel sources with heterogeneous noise profiles, underscoring a clear gap in the literature.

**Baselines:**  To bridge this gap, two intuitive baseline strategies naturally emerge but, unfortunately, fail dramatically in key worst-case scenarios. The first baseline selects data sources uniformly at random, effectively averaging the variance across all sources. In practice, this leads to severe performance degradation when all the sources, barring the optimal source exhibit extremely high variance—referred to as Worst-Case Instance 1 (WC-1). Here, the average variance becomes dominated by the variance of the high-variance sources, which dramatically inflates the regret. The second baseline employs a two-phase approach, initially identifying the source with the lowest variance through a dedicated exploration phase before transitioning to standard MAB strategies. However, this method becomes problematic in instances where all sources exhibit nearly identical variances—Worst-Case Instance 2 (WC-2). In WC-2, the initial exploration phase becomes redundant and costly, incurring significant regret without practical gain since distinguishing among similar-variance sources is unnecessary.

**Motivating SOAR:**  To address these shortcomings, we propose SOAR (Source-Optimistic Adaptive Regret Minimization), a novel adaptive algorithm that jointly balances exploration and exploitation across both data sources and reward arms. SOAR begins with a preprocessing phase to eliminate high-variance sources, then employs an innovative combination of upper confidence bounds (UCB) on arm rewards and lower confidence bounds (LCB) on source variances. By dynamically adapting to observed feedback, SOAR gravitates toward reliable, low-noise sources while simultaneously identifying and exploiting the most rewarding arms. Our theoretical analysis demonstrates that SOAR attains a near-optimal regret bound

of $\tilde{O}\left(\frac{M\bar{\eta}^4\bar{\mu}}{c^{*4}} + \frac{K\bar{\eta}^2\bar{\mu}}{c^{*2}} + KM\bar{\mu} + \frac{M\bar{\eta}^4\bar{\mu}}{\nu} + \sqrt{K}\sqrt{KMc^{*2} + \sum_{j:\sigma_j-\sigma^*>\gamma}\frac{Q_j(T)}{(\gamma+2\sigma^*)^2}} + \sum_{i=2}^{K}\frac{(\sigma^*+\gamma)^2}{\Delta_i}\right)$ approaching the

performance of an oracle algorithm with privileged knowledge of the best variance source (Remark 5.2). This remarkable performance is attributed to our carefully crafted confidence-bound mechanism, which enables adaptive management of heterogeneous noise with negligible regret overhead. We anticipate that this work will open up exciting new avenues in adaptive online learning, inspiring future research directions such as handling arm-dependent variances, incorporating explicit costs associated with querying various data sources, and developing algorithms that are robust to non-stationary conditions where source qualities evolve over time.

**Related Work.**  Multi-fidelity bandits address scenarios where learners can query information at different levels of fidelity, often trading off cost and accuracy. For example, Kandasamy et al., 2016 (Kandasamy et al., 2016) introduced the idea of leveraging inexpensive, low-fidelity evaluations to accelerate optimization with expensive, high-fidelity sources, while Song et al., 2019 (Song et al., 2019) extended this framework to generalized black-box optimization. Further developments explored Gaussian process models (Poloczek et al., 2017) and batched multi-fidelity queries (Takeno et al., 2020).

Variance-adaptive bandits study settings where exploration policies explicitly account for the variance of observed rewards, with a rich literature focused on reward-dependent variance scaling. Audibert et al., 2009 (Audibert et al., 2009) proposed UCB-V, which adjusts exploration using empirical variance; Gabillon et al., 2012 (Gabillon et al., 2012), and Cowan and Katehakis, 2018 (Cowan et al., 2018) further analyzed the benefits of variance-based sampling in best-arm identification. Howard et al., 2018 (Howard et al., 2018) proposed empirical Bernstein inequalities to tighten exploration bounds. While these works significantly improve learning under variable noise, they assume a single stream of data per arm and are not designed to reason about multiple parallel, noisy sources with arm-independent structure.

Our setting integrates and generalizes both paradigms: it learns from multiple independent sources with unknown variances while simultaneously identifying optimal reward arms. This joint optimization across sources and arms calls for new algorithmic and analytical tools, which we develop in SOAR. Our results demonstrate that a carefully designed LCB-UCB framework can match the performance of idealized single-source MABs, while gracefully adapting to complex heteroscedastic environments.

# B. Proofs Related to PREPROCESS

## B.1. Proof of PREPROCESS Variance Concentration Lemma

**Lemma B.1** (PREPROCESS Variance Concentration). *Fix $\delta \in (0,1)$ and a sampling budget $\tau_p \in \mathbb{N}$. Assume $\epsilon'' < \min\left\{6\sigma_j^2, \frac{18\sigma_j^4}{\bar{\eta}^2}\right\}$ for each source $j \in [M]$, where $\epsilon''$ is the parameter appearing in Bernstein's Inequality ([Boucheron et al., 2013](); [Zhou, 2020]()). Then, with probability at least $(1 - \delta/3)$,*

$$\left|\hat{\sigma}_{j,\mathrm{pre}}^2(\tau_p) - \sigma_j^2\right| \leq 8\bar{\eta}\,\sigma_j\sqrt{\frac{\log(4M/\delta)}{\tau_p}}.$$

*Proof.* Recall that PREPROCESS queries every source on a single fixed arm $i_0 \in [K]$ (Algorithm 3); we therefore fix $i = i_0$ throughout this proof. We bound the deviation of the preprocessing variance estimate $\hat{\sigma}_{j,\mathrm{pre}}^2(\tau_p)$ from the true source variance $\sigma_j^2$. The argument proceeds in three steps. First, we decompose the deviation into two contributions: (I) the fluctuation of the empirical second moment $\frac{1}{\tau_p}\sum_k (X_k - \mu_i)^2$ around $\sigma_j^2$, and (II) the error from estimating the unknown mean $\mu_i$ by $\hat{\mu}_i(\tau_p)$. We then control each term separately via Bernstein's Inequality (also stated below). Finally, we combine the two bounds and take a union bound over all sources $j \in [M]$ to obtain the stated high-probability guarantee. We begin with the decomposition:

$$
\begin{aligned}
\left|\hat{\sigma}_{j,\mathrm{pre}}^2(\tau_p) - \sigma_j^2\right| &= \left|\frac{1}{\tau_p}\sum_{k=1}^{\tau_p}\left(X_k - \hat{\mu}_i(\tau_p)\right)^2 - \sigma_j^2\right|, \\
&= \left|\frac{1}{\tau_p}\sum_{k=1}^{\tau_p}\left(X_k - \mu_i + \mu_i - \hat{\mu}_i(\tau_p)\right)^2 - \sigma_j^2\right|, \\
&\leq \left|\frac{2}{\tau_p}\sum_{k=1}^{\tau_p}\left(X_k - \mu_i\right)^2 + \frac{2}{\tau_p}\sum_{k=1}^{\tau_p}(\mu_i - \hat{\mu}_i(\tau_p))^2 - \sigma_j^2\right|, \\
&\leq \underbrace{\left|\frac{2}{\tau_p}\sum_{k=1}^{\tau_p}\left(X_k - \mu_i\right)^2 - \sigma_j^2\right|}_{(\mathrm{I}) \leq \epsilon/2} + \underbrace{\left|\frac{2}{\tau_p}\sum_{k=1}^{\tau_p}(\mu_i - \hat{\mu}_i(\tau_p))^2\right|}_{(\mathrm{II}) \leq \epsilon/2}.
\end{aligned}
$$

**Bounding Term (I):** To bound Term I we will use Bernstein's Inequality, stated below.

**Theorem B.2** (Bernstein's Inequality). *([Boucheron et al., 2013](), Cor. 2.11); ([Zhou, 2020](), Theorem 1.2) Suppose $Z_1, \ldots, Z_n$ are independent random variables with finite variances, and suppose that*

$$\max_{1 \leq k \leq n}|Z_k| \leq b$$

*almost surely for some constant $b > 0$. Let*

$$V = \sum_{k=1}^{n}\mathbb{E}[Z_k^2].$$

*Then, for every $t \geq 0$,*

$$\Pr\left(\sum_{k=1}^{n}(Z_k - \mathbb{E}[Z_k]) \geq t\right) \leq \exp\left(-\frac{t^2}{2(V + \frac{1}{3}bt)}\right),$$

*and*

$$\Pr\left(\sum_{k=1}^{n}(Z_k - \mathbb{E}[Z_k]) \leq -t\right) \leq \exp\left(-\frac{t^2}{2(V + \frac{1}{3}bt)}\right).$$

Now, coming back to our proof:
Fix a source $j \in [M]$ and arm $i \in [K]$. Consider $Z_k = (X_k - \mu_i)^2$, where $X_k = \mu_i + \varepsilon$ and $\varepsilon \sim \mathcal{D}(j)$ is drawn from an

(unknown) underlying noise distribution $\mathcal{D}(j)$ of the selected data-source $j \in [M]$.

From our problem setting and Assumption 2.1, it is evident $\left|Z_k\right| \leq \bar{\eta}^2$. Now

$$V = \sum_{k=1}^{\tau_p} \mathbb{E}[Z_k^2] \leq \sum_{k=1}^{\tau_p} \bar{\eta}^2 E[Z_k] \leq \tau_p\, \bar{\eta}^2 \sigma_j^2.$$

Applying Bernstein's, we have with probability $\left(1 - \dfrac{\delta}{6}\right)$:

$$\Pr\left(\left|\sum_{k=1}^{\tau_p}\left((X_k - \mu_i)^2 - \sigma_j^2\right)\right| \geq \epsilon\right) \;\leq\; 2\exp\left(-\frac{\epsilon^2}{2\left(\tau_p\, \bar{\eta}^2\sigma_j^2 + \frac{1}{3}\bar{\eta}^2\epsilon\right)}\right),$$

$$\equiv \Pr\left(\left|\frac{1}{\tau_p}\sum_{k=1}^{\tau_p}(X_k - \mu_i)^2 - \sigma_j^2\right| \geq \frac{\epsilon}{\tau_p}\right) \;\leq\; 2\exp\left(-\frac{\epsilon^2}{2\left(\tau_p\, \bar{\eta}^2\sigma_j^2 + \frac{1}{3}\bar{\eta}^2\epsilon\right)}\right),$$

$$\implies \Pr\left(\left|\frac{1}{\tau_p}\sum_{k=1}^{\tau_p}(X_k - \mu_i)^2 - \sigma_j^2\right| \geq \epsilon'\right) \;\leq\; 2\exp\left(-\frac{\tau_p^2\epsilon'^2}{2\left(\tau_p\, \bar{\eta}^2\sigma_j^2 + \frac{1}{3}\bar{\eta}^2\, \tau_p\, \epsilon'\right)}\right), \quad \left[\epsilon' = \frac{\epsilon}{\tau_p}\right]$$

Replace $\epsilon'$ by $\epsilon''/2$ we get:

$$\Pr\left(\left|\frac{1}{\tau_p}\sum_{k=1}^{\tau_p}(X_k - \mu_i)^2 - \sigma_j^2\right| \geq \frac{\epsilon''}{2}\right) \;\leq\; 2\exp\left(-\frac{\tau_p\, \epsilon''^2}{8\left(\bar{\eta}^2\sigma_j^2 + \frac{1}{6}\bar{\eta}^2\epsilon\right)}\right).$$

Choose $\epsilon''$ such that $\dfrac{\sigma_j^2}{\epsilon''^2} > \dfrac{1}{6\epsilon}$ i.e $\epsilon'' < 6\sigma_j^2$, This is equivalent to $\dfrac{\delta}{6} \geq 2\exp\left(-\dfrac{\tau_p\, \epsilon''^2}{16\bar{\eta}^2\sigma_j^2}\right) \implies \epsilon'' \leq 4\bar{\eta}\sigma_j\sqrt{\dfrac{\log(12/\delta)}{\tau_p}}.$

Thus with $\epsilon'' < 6\sigma_j^2$, and a union bound over all sources $j \in [M]$ we have with probability $\left(1 - \dfrac{\delta}{6}\right)$,

$$\left|\frac{2}{\tau_p}\sum_{k=1}^{\tau_p}(X_k - \mu_i)^2 - \sigma_j^2\right| \leq 4\bar{\eta}\sigma_j\sqrt{\frac{\log(12M/\delta)}{\tau_p}}.$$

**Bounding Term (II):** Consider $D_k = (X_k - \mu_i)^2$, where $X_k = \mu_i + \varepsilon$ and $\varepsilon \sim \mathcal{D}(j)$ is drawn from an (unknown) underlying noise distribution $\mathcal{D}(j)$ of the selected data-source $j \in [M]$. Similar to how we bounded term(I), it is evident $\left|D_k\right| \leq \bar{\eta}$. Additionally conditional variance $V = \sum_{k=1}^{\tau_p} E[D_k^2] = \tau_p \cdot \sigma_j^2$

Applying Bernstein's, we have with probability $(\delta/6)$:

$$\Pr\left(\frac{1}{\tau_p}\left|\sum_{k=1}^{\tau_p}\left(X_k - \mu_i\right)\right| \geq \epsilon'/2\right) \;\leq\; 2\exp\left(-\frac{\tau_p\, \epsilon'^2}{8\left(\sigma_j^2 + \frac{1}{6}\bar{\eta}\epsilon'\right)}\right), \quad \left[\epsilon' = \frac{\epsilon}{\tau_p}\right]$$

$$\implies \Pr\left(\left|\hat{\mu}_i(\tau_p) - \mu_i\right| \geq \epsilon'/2\right) \;\leq\; 2\exp\left(-\frac{\tau_p\, \epsilon'^2}{8\left(\sigma_j^2 + \frac{1}{6}\bar{\eta}\epsilon'\right)}\right),$$

$$\implies \Pr\left(\left|\hat{\mu}_i(\tau_p) - \mu_i\right|^2 \geq \epsilon'^2/4\right) \;\leq\; 2\exp\left(-\frac{\tau_p\, \epsilon'^2}{8\left(\sigma_j^2 + \frac{1}{6}\bar{\eta}\epsilon'\right)}\right),$$

Setting $\epsilon'' = \epsilon'^2/2$, so that $\epsilon' = \sqrt{2\epsilon''}$, the bound becomes

$$\Pr\left(\left|\hat{\mu}_i(\tau_p) - \mu_i\right|^2 \geq \epsilon''/2\right) \;\leq\; 2\exp\left(-\frac{\tau_p\, \epsilon''}{4\left(\sigma_j^2 + \frac{\sqrt{2}}{6}\bar{\eta}\sqrt{\epsilon''}\right)}\right).$$

Choosing $\epsilon$ small enough that $\frac{\sigma_j^2}{\epsilon} > \frac{\sqrt{2}\,\bar\eta}{6\sqrt{\epsilon}}$, equivalently $\epsilon < \frac{18\sigma_j^4}{\bar\eta^2}$, the second term in the denominator is dominated by $\sigma_j^2$.

Imposing $\delta/6 \geq 2\exp\left(-\frac{\tau_p\,\epsilon''}{4\sigma_j^2}\right)$ then gives $\epsilon'' \leq \frac{8}{\tau_p}\sigma_j^2\log(12/\delta)$. Thus, for $\epsilon'' < \frac{18\sigma_j^4}{\bar\eta^2}$, a union bound over all sources $j \in [M]$ yields, with probability at least $1 - \frac{\delta}{6}$,

$$\left|\hat\mu_i(\tau_p) - \mu_i\right|^2 \leq \frac{4}{\tau_p}\sigma_j^2\log(12M/\delta).$$

**Putting it All Together:**

$$\left|\hat\sigma_{\tau_p}^2(j) - \sigma_j^2\right| \leq \left|\frac{2}{\tau_p}\sum_{k=1}^{\tau_p}\left(X_k - \mu_i\right)^2 - \sigma_j^2\right| + \left|\frac{2}{\tau_p}\sum_{k=1}^{\tau_p}(\mu_i - \hat\mu_i(\tau_p))^2\right|,$$

$$\leq \underbrace{4\bar\eta\sigma_j\sqrt{\frac{\log(12M/\delta)}{\tau_p}}}_{C} + \underbrace{\frac{8}{\tau_p}\sigma_j^2\log(12M/\delta)}_{D}.$$

Now we can choose the number of times each source is queried (i.e, preprocessing budget $\tau_p$) such that $C > D$ and then bound the expression $C + D$ by $2C$. This is achieved when $\tau_p > \frac{4\sigma_j^2\log(12M/\delta)}{\bar\eta^2}$.

Now, we know from our problem setup and Assumption 2.1 that $\sigma_j^2 \leq \bar\eta^2$ for any $j \in [M]$. Therefore the greatest lower bound for $\tau_p$ is $\tau_p > 4\log(12M/\delta)$ which is true for any valid $M$. Additionally this constraint on $\tau_p$ is subsumed by the stopping condition stated in Theorem 4.3. Hence we have with probability $1 - \frac{\delta}{3}$:

$$\left|\hat\sigma_{j,\text{pre}}^2(\tau_p) - \sigma_j^2\right| \leq 8\bar\eta\sigma_j\sqrt{\frac{\log(12M/\delta)}{\tau_p}}.$$

which concludes the proof. $\qquad\square$

## B.2. Proof of Stopping Condition of PREPROCESS

**Theorem 4.3** (Stopping Condition of PREPROCESS). *Consider any $\delta \in (0,1)$. If* PREPROCESS *is run with runtime budget of at least $\tau_p$, where $\tau_p \geq \frac{1024\bar\eta^4\log(12M/\delta)}{c^{*4}}$, where $c^* \in \mathbb{R}_+$ is a user defined algorithm parameter of the order of $\sigma_j$, then any source $j \in [M]$ with variance $\sigma_j^2 > \sigma^{*2} + c^{*2}$, will be eliminated with probability $(1 - \delta/3)$.*

*Proof.* For the purposes of this proof we will be using the variance concentration defined above in Lemma 4.2.
For the preprocessing algorithm to proceed without stopping, we need to find $\tau_p$ such that,

$$\begin{aligned}
&\text{LCB}_{\tau_p,\text{pre}}^{\sigma^2}(j) \leq \text{UCB}_{\tau_p,\text{pre}}^{\sigma^2}(j^*),\\
\iff\ & \hat\sigma_{j,\text{pre}}^2(\tau_p) - \text{conf}_j^{\sigma^2}(\tau_p) \leq \hat\sigma_{j^*,\text{pre}}^2(\tau_p) + \text{conf}_{j^*}^{\sigma^2}(\tau_p),\\
\implies\ & \sigma_j^2 - 2\,\text{conf}_j^{\sigma^2}(\tau_p) \leq \sigma^{*2} + 2\,\text{conf}_{j^*}^{\sigma^2}(\tau_p), &&\text{[Lemma 4.2]}\\
\implies\ & \sigma_j^2 - \sigma^{*2} \leq 2\,\text{conf}_j^{\sigma^2}(\tau_p) + 2\,\text{conf}_{j^*}^{\sigma^2}(\tau_p) \leq 4\,\text{conf}_j^{\sigma^2}(\tau_p),\\
\implies\ & \Delta_j^{\sigma^2} \leq 4\,\text{conf}_j^{\sigma^2}(\tau_p),\\
\implies\ & \Delta_j^{\sigma^2} \leq 32\,\bar\eta\,\sigma_j\sqrt{\frac{\log(12M/\delta)}{\tau_p}}.
\end{aligned}$$

For any $j \in [M]$, the source is eliminated when $\sigma_j^2 - \sigma_*^2 > c^{*2}$, i.e. $\Delta_j^{\sigma^2} > c^{*2}$. Requiring this separation to be detectable gives

$$c^{*2} \le 32\,\bar{\eta}\,\sigma_j \sqrt{\frac{\log(12M/\delta)}{\tau_p}},$$

$$\implies \quad c^{*4} \le \frac{1024\,\bar{\eta}^4 \log(12M/\delta)}{\tau_p}, \qquad\qquad [\sigma_j^2 \le \bar{\eta}^2,\ \text{Assumption 2.1}]$$

$$\implies \quad \tau_p \le \frac{1024\,\bar{\eta}^4 \log(12M/\delta)}{c^{*4}}.$$

Hence for PREPROCESS to proceed with the elimination of sources we need $\tau_p > \dfrac{1024\bar{\eta}^4 \log(12M/\delta)}{c^{*4}}$ which concludes the proof. $\qquad\square$

# C. Proofs Related to the adaptive phase of `SOAR`:

### C.1. Proof of the Reward Variance Concentration:

**Lemma C.1** (Source Variance Concentration)**.** *Assume there exists a source $j \in S_{\mathcal{G}}$ such that $\sigma_j^2 \geq c^{*2}$ where $c^* \in \mathbb{R}_+$ is a user-defined parameter chosen on the scale of $\sigma_j$. Define*

$$Q_j := \begin{cases} \max\{\kappa_j, \nu\}, & \text{if } \kappa_j \text{ is known,} \\ \max\{\bar{\eta}^2 \, \hat{\sigma}_j^2(t), \nu\}, & \text{if } \kappa_j \text{ is unknown,} \end{cases}$$

*Recall that $\nu \in \mathbb{R}_+$ is a user-defined parameter chosen on the scale of the fourth central moment $\kappa_j$. If the source $j$, is queried for atleast $m_j(t)$ iterations where $m_j(t) = K + \dfrac{4\bar{\eta}^4 \log(3MT/\delta)}{\nu}$. Then for any $t \in [T]$, with probability at least $1 - \delta/3$,*

$$\left| \hat{\sigma}_j^2(t) - \sigma_j^2 \right| \leq \sqrt{\frac{Q_j(t) \log(3MT/\delta)}{m_j(t) - K}}.$$

*Proof.* Suppose $T_k(j) = \{k \leq t : j_k = j\}$ denote the number of times a source $j \in S_{\mathcal{G}}$ was selected upto time $t$. Consider the sequence of random variables defined by:

$$D_k = (X_k - \hat{\mu}_{i_k j}(t))^2 - \frac{\sigma_j^2(n_{i_k j}(t) - 1)}{n_{i_k j}(t)} \quad \forall k \in T_t(j).$$

We observe that $\{D_k\}_{k \in T_t(j)}$ forms a Martingale Difference Sequence (MDS) with respect to the filtration sequence $\mathcal{F}_k = \sigma\left(\{i_s\}_{s=1}^k\right)$.

**Verifying the MDS property:** Note that $\mathbb{E}\left[D_k | \mathcal{F}_k\right] = 0$. This follows from the property that

$$\mathbb{E}\left[(X_i - \hat{\mu}_{i_k j}(t))^2 | \mathcal{F}_k\right] = \frac{\sigma_j^2(n_{i_k j}(t) - 1)}{(n_{i_k j}(t))}.$$

Specifically:

$$\mathbb{E}\left[D_k | \mathcal{F}_k\right] = \mathbb{E}\left[((X_k - \hat{\mu}_{i_k j}(t))^2 | \mathcal{F}_k)^2\right] - \frac{\sigma_j^2(n_{i_k j}(t) - 1)}{n_{i_k j}(t)},$$

$$= \frac{\sigma_j^2(n_{i_k j}(t) - 1)}{n_{i_k j}(t)} - \frac{\sigma_j^2(n_{i_k j}(t) - 1)}{n_{i_k j}(t)} = 0.$$

**Bounding $D_k$:** Additionally we note that $D_k$ is bounded i.e $D_k \leq 2\bar{\eta}^2$. This follows as:

$$D_k = (X_k - \hat{\mu}_{i_k j}(t))^2 - \frac{\sigma_j^2(n_{i_k j}(t) - 1)}{n_{i_k j}(t)}, \qquad k \in T_t(i)$$

$$\leq (X_k - \mu_{i_k})^2 + (\mu_{i_k} - \hat{\mu}_{i_k j}(t))^2,$$

$$\leq 2\bar{\eta}^2.$$

Considering these properties we can now leverage standard martingale inequalities:

**Theorem C.2** (Freedman's Inequality, (Raban))**.** *Let $\{(D_k, \mathcal{F}_k)\}$ be a martingale difference sequence such that*

1. $\mathbb{E}[D_k \mid \mathcal{F}_{k-1}] = 0$

2. $D_k \leq b$.

*Then for all $\lambda \in (0, 1/b)$ and $\delta \in (0, 1)$,*

$$\mathbb{P}\left(\sum_{k=1}^{T} D_k \leq \lambda \sum_{k=1}^{T} \mathbb{E}[D_k^2 \mid \mathcal{F}_{k-1}] + \frac{\log(1/\delta)}{\lambda}\right) \geq 1 - \delta.$$

Now, since the martingale difference sequence (MDS) $\{D_k\}_{k \in T_t(j)}$ satisfies the conditions of Theorem C.2, we can apply Freedman's Inequality. We begin by evaluating the partial sum $\sum_{k \in T_t(j)} D_k$, which corresponds to the left-hand side of Theorem C.2.

**Summing $D_k$:**

$$\sum_{k \in T_t(j)} D_k = \sum_{i=1}^{K} \sum_{\ell \in \tau_{ij}(t)} \left((X_\ell - \hat{\mu}_{i_\ell j}(t))^2 - \frac{\sigma_j^2(n_{i_\ell j}(t) - 1)}{n_{i_\ell j}(t)}\right),$$

$$= \sum_{i=1}^{K} \left(\sum_{\ell \in \tau_{ij}(t)} (X_\ell - \hat{\mu}_{i_\ell j}(t))^2 - \sigma_j^2(n_{i_\ell j}(t) - 1)\right),$$

$$= (m_j(t) - K)\left|(\hat{\sigma}_j^2(t) - \sigma_j^2)\right|.$$

Our next task is to bound the predictable quadratic variation term $\sum_{k \in T_t(j)} \mathbf{E}[D_k^2 \mid \mathcal{F}_k]$, which serves as the conditional variance proxy in Theorem C.2. Note that $\mathbf{E}[D_k \mid \mathcal{F}_k] = 0$ since $\{D_k\}$ is a martingale difference sequence.

To achieve this we will make use of *Result 3* in (O'Neill, 2014) (p. 284), which provides a moment expression for the (unbiased) sample variance under simple random sampling. Specifically,

$$\text{Var}(S_n^2) = \frac{1}{n}\left(\mu_4 - \frac{n-3}{n-1}\sigma^4\right). \tag{13}$$

Here, $S_n^2 := \frac{1}{n-1}\sum_{i=1}^{n}(X_i - \bar{X})^2$ denotes the unbiased sample variance computed from $n$ i.i.d. samples $X_1, \ldots, X_n$ with mean $\mu$ and population variance $\sigma^2$. Moreover, $\mu_4 := \mathbb{E}[(X - \mu)^4]$ denotes the fourth central moment of the underlying distribution.

Adapting the above (Equation (13)) to our setting we obtain the following expression:

$$\text{Var}(\hat{\sigma}_{ij}^2(t)) = \frac{1}{n_{ij}(t)}\left(\kappa_j - \frac{n_{ij}(t) - 3}{n_{ij}(t) - 1}\sigma_j^4\right) \leq \frac{\kappa_j}{n_{ij}(t)}. \tag{14}$$

**Bounding Variation Term:**

$$\sum_{k \in T_t(j)} \mathbb{E}[D_k^2 \mid \mathcal{F}_{k-1}] = \sum_{i=1}^{K} \sum_{\ell \in \mathcal{T}_{ij}(t)} \mathbb{E}\left[\left((X_\ell - \hat{\mu}_{i_\ell j}(t))^2 - \frac{\sigma_j^2(n_{i_\ell j}(t) - 1)}{n_{i_\ell j}(t)}\right)^2\right],$$

$$\leq \sum_{i=1}^{K} \sum_{\ell \in \mathcal{T}_{ij}(t)} \text{Var}\left[(X_k - \hat{\mu}_{i_k j}(t))^2\right],$$

$$\leq \sum_{i=1}^{K} \text{Var}(\hat{\sigma}_{ij}^2(t))(n_{ij}(t) - 1)^2,$$

$$\leq \sum_{i=1}^{K} \frac{(n_{ij}(t) - 1)^2 \kappa_j}{n_{ij}(t)}, \quad \text{[From Equation (14)]}$$

$$\leq (m_j(t) - K)\kappa_j,$$

$$\leq (m_j(t) - K)Q_j(t).$$

where

$$Q_j(t) = \begin{cases} \max\{\kappa_j, \nu\} & \text{if } \kappa_j \text{ is known} \\ \max\{\bar{\eta}^2 \hat{\sigma}_j^2(t), \nu\} & \text{if } \kappa_j \text{ is unknown} \end{cases}$$

**Putting it All Together:** Hence, we have with high probability $1 - \delta/3$

$$(m_j(t) - K) \left| (\hat{\sigma}_j^2(t) - \sigma_j^2) \right| \leq \lambda (m_j(t) - K) Q_j(t) \;+\; \frac{\log(3/\delta)}{\lambda}.$$

Optimizing over $\lambda$ (i.e., minimizing the resulting bound) yields

$$\lambda^* = \sqrt{\frac{\log(3/\delta)}{Q_j(t)(m_j(t) - K)}}.$$

Applying the Freedman boundary constraint over $\lambda^*$ yields:

$$\lambda^* = \sqrt{\frac{\log(3MT/\delta)}{Q_j(t)(m_j(t) - K)}} < \frac{1}{2\bar{\eta}^2},$$

$$m_j(t) > K + \frac{4\bar{\eta}^4 \log(3MT/\delta)}{Q_j(t)},$$

$$m_j(t) > K + \frac{4\bar{\eta}^4 \log(3MT/\delta)}{\nu}.$$

The last line follows from the definition of $Q_j(t)$. Applying a union bound over all sources $j \in S_{\mathcal{G}} \subseteq [M]$ and all $t \in [T]$ yields the following bound that holds with probability at least $1 - \delta/3$.

$$\left| \hat{\sigma}_j^2(t) - \sigma_j^2 \right| \leq 2\sqrt{\frac{Q_j(t)\log(3MT/\delta)}{m_j(t) - K}}.$$

For the above bound to hold, it suffices that

$$m_j(t) > K + \frac{4\bar{\eta}^4 \log(3MT/\delta)}{\nu}.$$

Which concludes the proof. $\qquad\square$

### C.2. Proof of the Sandwiching Corollary

**Corollary C.3** (Variance Sandwiching). *Consider any $\delta \in (0,1)$. Assume there exists a source $j \in S_{\mathcal{G}} \subseteq [M]$ such that $\sigma_j^2 \geq c^{*2}$ where $c^* \in \mathbb{R}_+$ is a user-defined parameter of the order of $\sigma_j$. If such a source $j$ is queried at least $m_j(t)$ times where $m_j(t) \geq K + \frac{16\bar{\eta}^4 \log(3MT/\delta)}{c^{*4}}$, then we have with probability $(1 - \delta/3)$*

$$\sigma_j^2 \;\leq\; 2\hat{\sigma}_j^2(t) \;\leq\; 3\sigma_j^2.$$

*Proof.* Recall from Lemma 4.4 that: $\left| \hat{\sigma}_j^2(t) - \sigma_j^2 \right| \leq 2\sqrt{\frac{Q_j(t)\log(3MT/\delta)}{m_j(t) - K}}$.

Choosing $m_j(j)$ such that $m_j(t) = K + \frac{16\,Q_j(t)\log(3MT/\delta)}{\sigma_j^4}$ we have

$$\left| \hat{\sigma}_j^2(t) - \sigma_j^2 \right| \leq \frac{\sigma_j^2}{2} \implies \frac{\sigma_j^2}{2} \leq \hat{\sigma}_j^2(t) \leq \frac{3\sigma_j^2}{2},$$
$$\implies \sigma_j^2 \leq 2\hat{\sigma}_j^2(t) \leq 3\sigma_j^2.$$

Now $m_j(t) = K + \frac{16 Q_j(t)\log(3MT/\delta)}{\sigma_j^4}$ can be upper bounded by $K + \frac{16\bar{\eta}^4 \log(3MT/\delta)}{c^{*4}}$ using our regime condition ($\exists\, j \in S_{\mathcal{G}} \subseteq [M]$ such that $\sigma_j^2 \geq c^{*2}$) and Assumption 2.1, which concludes the proof. $\qquad\square$

## C.3. Proof of the Mean Reward Concentration:

**Lemma C.4** (Mean Reward Concentration)**.** *Consider any $\delta \in (0,1)$. Assume there exists a source $j \in S_{\mathcal{G}} \subseteq [M]$ such that $\sigma_j^2 \geq c^{*2}$ and $\mathrm{LCB}_{\tau_p}^{\sigma^2, \mathrm{pre}}(j) \geq \frac{c^{*2}}{2}$, where $c^* \in \mathbb{R}_+$ is a user-defined parameter of the order of $\sigma_j$. If an arm $i \in K$, is queried for at least $\alpha$ iterations with this source $j$ where $\alpha \geq \frac{\bar{\eta}^2 \log(3KT/\delta)}{c^{*2}}$ then for any $t \in [T]$, we have, with probability at least $1 - \delta/3$,*

$$|\mu_i - \hat{\mu}_i(t)| \leq \frac{2\sqrt{\log(3KT/\delta) \sum_{j=1}^{M} n_{ij}(t)\sigma_j^2}}{n_i(t)}. \tag{9}$$

*Proof.* Suppose $T_t(i) = \{s \leq t : i_s = i\}$ denotes the set of time steps at which arm $i$ was selected up to time $t$. Consider the sequence of random variables defined by

$$D_{t,i} := X_{t,i} - \mu_i, \quad \text{for all } t \in T_t(i).$$

To establish a concentration bound, the key observation is that $\{D_{t,i}\}_{t \in T_t(i)}$ forms a martingale difference sequence with respect to the filtration $\mathcal{F}_{t-1} = \sigma\left(\{X_s, i_s, j_s\}_{s=1}^{t-1}, i_t = i\right)$, generated by all arm selections and observations prior to round $t$. Precisely note that each $D_{t,i}$ is integrable, adapted to $\mathcal{F}_t$, and satisfies $\mathbf{E}[D_{t,i} \mid \mathcal{F}_{t-1}] = \mathbf{E}[X_{t,i} - \mu_i \mid \mathcal{F}_{t-1}] = \mu_i - \mu_i = 0$. Thus, by construction, the sequence satisfies the martingale difference property and enables us to leverage standard martingale concentration inequalities.

**Theorem C.5** (Freedman's Inequality, (Raban))**.** *Let $\{(D_k, \mathcal{F}_k)\}$ be a martingale difference sequence such that*

1. $\mathbb{E}[D_k \mid \mathcal{F}_{k-1}] = 0$

2. $D_k \leq b$.

*Then for all $\lambda \in (0, 1/b)$ and $\delta \in (0,1)$,*

$$\mathbb{P}\left(\sum_{k=1}^{T} D_k \leq \lambda \sum_{k=1}^{T} \mathbb{E}[D_k^2 \mid \mathcal{F}_{k-1}] + \frac{\log(1/\delta)}{\lambda}\right) \geq 1 - \delta.$$

We have a bound $\bar{\eta}$ on $D_{t,i}$, the first requirement of the theorem has already been proven. Applying Freedman's, we get

$$\mathbb{P}\left(\sum_{t \in T_i(t)} D_{t,i} \leq \lambda \sum_{t \in T_i(t)} \mathbb{E}[D_{t,i}^2 \mid \mathcal{F}_{t-1}] + \frac{\log(3/\delta)}{\lambda}\right) \geq 1 - \frac{\delta}{3},$$

$$\mathbb{P}\left(\sum_{t \in T_i(t)} D_{t,i} \leq \lambda \sum_{j=1}^{M} n_{ij}(t) \cdot \sigma_j^2 + \frac{\log(3/\delta)}{\lambda}\right) \geq 1 - \frac{\delta}{3}. \quad \left[\mathbb{E}[D_{t,i}^2 \mid \mathcal{F}_{t-1}] = \sigma_j^2\right]$$

Optimizing over $\lambda$ gives us $\lambda^* = \sqrt{\dfrac{\log(3/\delta)}{\sum_{j=1}^{M} n_{ij}(t) \cdot \sigma_j^2}}.$

$$\implies \mathbb{P}\left(\sum_{t \in T_i(t)} D_{t,i} \leq 2\sqrt{\log\left(\frac{3}{\delta}\right) \sum_{j=1}^{M} n_{ij}(t)\sigma_j^2}\right) \geq 1 - \frac{\delta}{3}.$$

A union bound across K arms and T time-steps gives us

$$\implies \mathbb{P}\left(\sum_{t \in T_i(t)} D_{t,i} \leq 2\sqrt{\log\left(\frac{3KT}{\delta}\right) \sum_{j=1}^{M} n_{ij}(t)\sigma_j^2}\right) \geq 1 - \frac{\delta}{3}.$$

Dividing both sides by $n_i(t)$ we get obtain concentration bound with probability $1 - \dfrac{\delta}{3}$

$$\left|\hat{\mu}_t(i) - \mu_i\right| \le \frac{2\sqrt{\log(3KT/\delta)\sum_{j=1}^{M} n_{ij}(t)\sigma_j^2}}{n_i(t)}.$$

To conclude the proof of Lemma 4.6, we must satisfy the Freedman boundary constraint. Specifically we need to make sure $n_i(t)$ is large enough so $\lambda^* < \dfrac{1}{\bar{\eta}}$

$$\lambda^* = \sqrt{\frac{\log(3KT/\delta)}{\sum_{j=1}^{n} n_{ij}(t)\cdot\sigma_j^2}} < \frac{1}{\bar{\eta}}.$$

**Variance Identification (VarI):** If a source $j \in S_{\mathcal{G}}$ satisfies $\sigma_j^2 \ge c^{*2}$, then we can derive an empirical test to identify such sources with high probability. Let $\tau_{\text{VarI}}$ denote the number of queries required for this identification. Recall from the preprocessing phase that, once source $j$ has been queried at least $\tau_{\text{VarI}}$ times, we can compute the LCB on the variance estimate using Equation (4). Specifically

$$\text{LCB}_{\tau_{\text{VarI}}}^{\sigma,\text{pre}}(j) = \hat{\sigma}_j^2(\tau_{\text{VarI}}) - 8\bar{\eta}^2\sqrt{\frac{\log(12M/\delta)}{\tau_{\text{VarI}}}},$$

$$\ge \sigma_j^2 - 16\bar{\eta}^2\sqrt{\frac{\log(12M/\delta)}{\tau_{\text{VarI}}}}, \quad [\text{Using Lemma 4.2}]$$

$$\ge c^{*2} - 16\bar{\eta}^2\sqrt{\frac{\log(12M/\delta)}{\tau_{\text{VarI}}}}, \quad [\text{As } \sigma_j^2 > c^{*2}]$$

$$\ge c^{*2} - \frac{c^{*2}}{2}, \quad [\text{Choosing } \tau_{\text{VarI}} = \frac{1024\,\bar{\eta}^4\log(12M/\delta)}{c^{*4}}]$$

$$\ge \frac{c^{*2}}{2}.$$

Hence if a source has $\text{LCB}_{\tau_{\text{VarI}}}^{\sigma,\text{pre}}(j) \ge \dfrac{c^{*2}}{2}$ after being queried for $\tau_{\text{VarI}}$ iterations, then we can conclude that its true variance $\sigma_j^2$, satisfies $\sigma_j^2 \ge c^{*2}$ with high probability.
Finally, note that from Theorem 4.3 $\tau_P = O(\tau_{\text{VarI}})$, so $\tau_P$ subsumes $\tau_{\text{VarI}}$. Henceforth, we only use $\tau_P$.

**Required Starting Conditions:** Consider the denominator of $\lambda^*$. Let $\tilde{j}$ be a source that satisfies our **VarI** check. Now

$$\sum_{j=1}^{n} n_{ij}(t)\cdot\sigma_j^2 \ge n_{i\tilde{j}}(t)\cdot\sigma_{\tilde{j}}^2 \ge \alpha c^{*2}.$$

where $\alpha$ is the number of times we must query an arm with source $\tilde{j}$. Hence

$$\sqrt{\frac{\log(3KT/\delta)}{\alpha c^*}} < \frac{1}{\bar{\eta}} \implies \alpha > \frac{\bar{\eta}^2\log(3KT/\delta)}{c^{*2}}.$$

which is the required condition for the mean reward concentration bound to hold and concludes the proof. $\qquad\square$

## C.4. Validity of the UCB/LCB Constructions

**Corollary C.6** (Mean Reward UCB). *Consider any $\delta \in (0, 1)$. At any time step $t \in [T]$ and arm $i \in [K]$, with probability at least $1 - \dfrac{\delta}{3}$:*

$$\mu_i \leq \text{UCB}_t^\mu(i).$$

*Proof.* The proof of the above result directly follows from the mean reward concentration defined in Lemma 4.6, our definition of $\text{UCB}_t^\mu(i)$ from Equation (10) and an application of Corollary 4.5. $\square$

**Corollary C.7** (Reward Variance LCB). *Consider any $\delta \in (0, 1)$. At any time step $t \in [T]$ and source $j \in [M]$, with probability at least $1 - \dfrac{\delta}{3}$:*

$$\sigma_j \geq \text{LCB}_t^\sigma(j).$$

*Proof.* The proof of the above result directly follows from the mean reward concentration defined in Lemma 4.4, our definition of $\text{LCB}_t^\mu(i)$ from Equation (8) and an application of Corollary 4.5. $\square$

## D. Regret Analysis:

### D.1. Proof of Theorem 5.1

**Theorem 5.1** (Main Result: Regret Analysis of SOAR). *Consider $\delta \in (0, 1)$. Assume there exists a source $j \in S_{\mathcal{G}} \subseteq [M]$ such that $\sigma_j^2 \geq c^{*2}$ where $c^*, \gamma \in \mathbb{R}_+$ are user-defined parameters of the order of $\sigma_j$. Recall $Q_j(t)$ as defined in Equation (6) where $\nu \in \mathbb{R}_+$ is a user-defined parameter chosen on the scale of fourth central moment $\kappa_j$.*

*Then for any choice of preprocessing budget $\tau_p \geq \frac{1024 \cdot \bar{\eta}^4 \log(12M/\delta)}{c^{*4}}$, initial-exploration parameters $\alpha \geq \frac{\bar{\eta}^2 \log(3KT/\delta)}{c^{*2}}$,*

$\beta \geq 2K + \frac{4\bar{\eta}^4 \log(3MT/\delta)}{\nu} + \frac{16\bar{\eta}^4 \log(3MT/\delta)}{c^{*4}}$, *the regret of SOAR (Algorithm 4) can be bounded by* $\tilde{O}\Bigg( \frac{M\bar{\eta}^4 \bar{\mu}}{c^{*4}} + \frac{K\bar{\eta}^2 \bar{\mu}}{c^{*2}} +$

$KM\bar{\mu} + \frac{M\bar{\eta}^4 \bar{\mu}}{\nu} + \sqrt{K}\sqrt{KMc^{*2} + \sum\limits_{j:\,\sigma_j - \sigma^* > \gamma} \frac{Q_j(T)}{(\gamma + 2\sigma^*)^2} + \sum_{i=2}^{K} \frac{(\sigma^* + \gamma)^2}{\Delta_i}} \Bigg)$.[2]

*with high probability $(1 - \delta)$. SOAR can also be shown to yield an instance-independent (worst-case) regret bound of*

$\tilde{O}\left( \frac{M\bar{\eta}^4 \bar{\mu}}{c^{*4}} + \frac{K\bar{\eta}^2 \bar{\mu}}{c^{*2}} + KM\bar{\mu} + \frac{M\bar{\eta}^4 \bar{\mu}}{\nu} + (\sigma^* + c^*)\sqrt{KT} \right)$

**Proof:** We split the total regret into the *exploration regret* from the preprocessing and initial-exploration phases and the regret of the adaptive UCB–LCB phase over the remaining $T - M\tau_p - K\alpha - \tilde{M}\beta$ rounds, and bound each in turn.

**Exploration Regret:** We begin by stating the regret accumulated by preprocessing and initial exploration:

1. Preprocessing: Preprocessing accumulates regret of order $M\tau_p$, where $\tau_p \geq \dfrac{1024\bar{\eta}^4 \log(12M/\delta)}{c^{*4}}$.

2. Initial exploration: The initial exploration phase accumulates regret of order $K\alpha + \tilde{M}\beta$, where $\tilde{M} := |S_{\mathcal{G}}|$ and

$$\alpha \geq \frac{\bar{\eta}^2 \log(3KT/\delta)}{c^{*2}}, \qquad \beta \geq 2K + \frac{4\bar{\eta}^4 \log(3MT/\delta)}{\nu} + \frac{16\bar{\eta}^4 \log(3MT/\delta)}{c^{*4}}.$$

Hence, the regret incurred from exploration is:

$$\begin{aligned}
\text{Reg}_T^{\text{Exp}} &= M\tau_p \bar{\mu} + K\alpha\bar{\mu} + M\beta\bar{\mu} \\
&= \frac{M \cdot 1024\bar{\eta}^4 \log(12M/\delta)}{c^{*4}} \bar{\mu} + \frac{K\bar{\eta}^2 \log(3KT/\delta)}{c^{*2}} \bar{\mu} + M\left( 2K + \frac{4\bar{\eta}^4 \log(3MT/\delta)}{\nu} + \frac{16\bar{\eta}^4 \log(3MT/\delta)}{c^{*4}} \right) \bar{\mu}.
\end{aligned}$$

The order of $\text{Reg}_T^{\text{Exp}}$ is

$$\tilde{O}(\text{Reg}_T^{\text{Exp}}) = \tilde{O}\left( \frac{M\bar{\eta}^4 \bar{\mu}}{c^{*4}} + \frac{K\bar{\eta}^2 \bar{\mu}}{c^{*2}} + KM\bar{\mu} + \frac{M\bar{\eta}^4 \bar{\mu}}{\nu} \right).$$

We now seek to bound the regret for the remaining $T - M\tau_p - K\alpha - \tilde{M}\beta$ rounds of SOAR . To achieve this we must first bound $n_i(t)$ and $m_j(t)$.

**Bounding $n_i(t)$:** Note that at any time $t$, arm-$i$ can not be selected if $\text{UCB}_t^\mu(i) \leq \text{UCB}_t^\mu(i^*)$. So arm-$i$ only gets selected at time $t$ if:

$$\mu_{i^*} \leq \text{UCB}_t^\mu(i^*) \leq \text{UCB}_t^\mu(i) = \hat{\mu}_i(t) + \frac{2\sqrt{\log(3KT/\delta) \sum_{j=1}^{\tilde{M}} n_{ij}(t) \cdot 2\hat{\sigma}_j^2(t)}}{n_i(t)}, \quad [\text{From Corollary 4.5}]$$

$$\mu_{i^*} \leq \mu_i + \frac{4\sqrt{\log(3KT/\delta) \sum_{j=1}^{M} n_{ij}(t) \cdot (3\sigma_j^2)}}{n_i(t)}, \quad [\text{Using (Lemma 4.6)}]$$

$$\implies n_i(t) \leq \frac{4\sqrt{3 \log(3KT/\delta) \sum_{j=1}^{M} n_{ij}(t) \cdot \sigma_j^2}}{\Delta_i}.$$

---

[2]Throughout the paper, the $\tilde{O}(\cdot)$ notation suppresses polylogarithmic factors in the problem parameters.

**Bounding $m_j(t)$:** We also know that at any time $t$, a non optimal source $j$ gets selected if, $\text{LCB}_t^{\sigma^2}(j) \leq \text{LCB}_t^{\sigma^2}(j^*)$

$$\sigma_j^2 \leq \text{UCB}_t^{\sigma^2}(j),$$
$$\implies \sigma_j^2 \leq \hat{\sigma}_j^2(t) + \text{conf}_t^{\sigma^2}(j),$$
$$\sigma_j^2 - 2 \cdot \text{conf}_t^{\sigma^2}(j) \leq \hat{\sigma}_j^2(t) - 2 \cdot \text{conf}_t^{\sigma^2}(j) + \text{conf}_t^{\sigma^2}(j),$$
$$\sigma_j^2 - 2 \cdot \text{conf}_t^{\sigma^2}(j) \leq \hat{\sigma}_j^2(t) - \text{conf}_t^{\sigma^2}(j) \leq \text{LCB}_t^{\sigma^2}(j),$$
$$\sigma_j^2 - 2 \cdot \text{conf}_t^{\sigma^2}(j) \leq \text{LCB}_t^{\sigma^2}(j) \leq \text{LCB}_t^{\sigma^2}(j^*) \leq \sigma^{*2},$$
$$\sigma_j^2 - \sigma^{*2} \leq 2 \cdot \text{conf}_t^{\sigma^2}(j), \implies \Delta_j^{\sigma^2} \leq 2 \cdot \text{conf}_t^{\sigma^2}(j).$$

Where $\Delta_j^{\sigma^2}$ is defined as $\sigma_j^2 - \sigma^{*2}$

From our variance concentration defined in Lemma 4.4

$$\text{conf}_t(j) = 2\sqrt{\frac{Q_j(t)\log(3MT/\delta)}{m_j(t) - K}}.$$

Hence we have

$$\Delta_j^{\sigma^2} \leq 4\sqrt{\frac{Q_j(t)\log(3MT/\delta)}{m_j(t) - K}},$$
$$m_j(t) - K \leq \frac{16Q_j(t)\log(3MT/\delta)}{(\Delta_j^{\sigma^2})^2},$$
$$m_j(t) \leq K + \frac{16Q_j(t)\log(3MT/\delta)}{(\Delta_j^{\sigma^2})^2}.$$

Therefore, for our algorithm to operate normally i.e, select the optimal source we must query each source at least:

$$m_j(t) > K + \frac{16Q_j(t)\log(3MT/\delta)}{(\Delta_j^{\sigma^2})^2}. \tag{15}$$

**Deriving the Regret for the Remaining Rounds:** The regret in $T - M\tau_p - K\alpha - \tilde{M}\beta$ rounds of Running SOAR:

$$\text{Reg}_T = \sum_{i=2}^{K} n_i(T)\Delta_i \leq \sum_{i=2}^{K} 4\sqrt{3\log(3KT/\delta)\sum_{j=1}^{M} n_{ij}(t) \cdot \sigma_j^2},$$

$$\frac{\text{Reg}_T}{4\sqrt{3\log(3KT/\delta)}} \leq \sum_{i=2}^{K} \sqrt{\sum_{j|\sigma_j - \sigma^* > \gamma} n_{ij}(T)\sigma_j^2 + \sum_{j|\sigma_j - \sigma^* \leq \gamma} n_{ij}(T)\sigma_j^2} \quad \text{[split surviving variances across } \gamma\text{]},$$

$$\leq \sum_{i=2}^{K} \left(\sqrt{\sum_{j|\sigma_j - \sigma^* > \gamma} n_{ij}(T)\sigma_j^2} + \sqrt{\sum_{j|\sigma_j - \sigma^* \leq \gamma} n_{ij}(T)\sigma_j^2}\right) \quad [\sqrt{a+b} \leq \sqrt{a} + \sqrt{b}],$$

$$\leq \sum_{i=2}^{K} \left(\sqrt{\sum_{j|\sigma_j - \sigma^* > \gamma} n_{ij}(T)(\sigma_j - \sigma^* + \sigma^*)^2} + \sqrt{\sum_{j|\sigma_j - \sigma^* \leq \gamma} n_{ij}(T)\sigma_j^2}\right) \quad [\pm\sigma^*],$$

$$\leq \underbrace{\sum_{i=2}^{K} \sqrt{\sum_{j|\sigma_j - \sigma^* > \gamma} n_{ij}(T)(\sigma_j - \sigma^*)^2}}_{A} + \underbrace{\sum_{i=2}^{K} \sqrt{\sum_{j|\sigma_j - \sigma^* > \gamma} n_{ij}(T) \cdot \sigma^{*2}}}_{B}$$

$$+ \underbrace{\sum_{i=2}^{K} \sqrt{\sum_{j|\sigma_j - \sigma^* \leq \gamma} n_{ij}(T)\sigma_j^2}}_{C}.$$

where the last step follows since $[(a + b)^2 \leq 2a^2 + 2b^2]$. We will bound term A, term B and term C separately, as shown below:

**i) Bounding Term A:**

$$
\begin{aligned}
A &= (K-1) \sum_{i=2}^{K} \frac{1}{K-1} \sqrt{\sum_{j|\sigma_j - \sigma^* > \gamma} n_{ij}(T)(\sigma_j - \sigma^*)^2}, \\
&\leq \sqrt{(K-1)} \sqrt{\sum_{i=2}^{K} \sum_{j|\sigma_j - \sigma^* > \gamma} n_{ij}(T)(\sigma_j - \sigma^*)^2} \quad \text{[Using Cauchy-Schwarz inequality (Steele, 2004)]}, \\
&\leq \sqrt{K} \sqrt{\sum_{j|\sigma_j - \sigma^* > \gamma} m_T(j)(\sigma_j - \sigma^*)^2}, \\
&\leq \sqrt{K} \sqrt{\sum_{j|\sigma_j - \sigma^* > \gamma} \left( K + \frac{16 \, Q_j(T) \log(3MT/\delta)}{(\Delta_j^{\sigma^2})^2} \right)(\sigma_j - \sigma^*)^2} \quad \text{[From 15]}, \\
&\leq \sqrt{K} \sqrt{\sum_{j|\sigma_j - \sigma^* > \gamma} K(\sigma_j - \sigma^*)^2 + \sum_{j|\sigma_j - \sigma^* > \gamma} \frac{16 \, Q_j(T) \log(3MT/\delta)}{(\sigma_j - \sigma^*)^2 (\sigma_j + \sigma^*)^2}(\sigma_j - \sigma^*)^2}, \\
&\leq \sqrt{K} \sqrt{\sum_{j|\sigma_j - \sigma^* > \gamma} K \left( \sqrt{\sigma^{*2} + c^{*2}} - \sigma^* \right)^2 + \sum_{j|\sigma_j - \sigma^* > \gamma} \frac{16 \, Q_j(T) \log(3MT/\delta)}{(\sigma_j + \sigma^*)^2}} \quad \left[ \sigma_j < \sqrt{\sigma^{*2} + c^{*2}} \right], \\
&\leq \sqrt{K} \sqrt{\sum_{j|\sigma_j - \sigma^* > \gamma} K c^{*2} + \sum_{j|\sigma_j - \sigma^* > \gamma} \frac{16 \, Q_j(T) \log(3MT/\delta)}{(\sigma_j + \sigma^*)^2}} \quad [\sqrt{a+b} \leq \sqrt{a} + \sqrt{b}], \\
&\leq \sqrt{K} \sqrt{K M c^{*2} + \sum_{j|\sigma_j - \sigma^* > \gamma} \frac{16 \, Q_j(T) \log(3MT/\delta)}{(\gamma + 2\sigma^*)^2}} \quad \text{[Using } \gamma \text{ constraint from the sum]}.
\end{aligned}
$$

The final bound on term A is: $A \leq \sqrt{K} \sqrt{K M c^{*2} + \sum_{j|\sigma_j - \sigma^* > \gamma} \frac{16 \, Q_j(T) \log(3MT/\delta)}{(\gamma + 2\sigma^*)^2}}$

**ii) Bounding Term B:**

$$
\begin{aligned}
B &= \sum_{i=2}^{K} \sqrt{\sum_{j|\sigma_j - \sigma^* \leq \gamma} n_{ij}(T) \cdot \sigma_j^2} \leq \sum_{i=2}^{K} \sqrt{\sum_{j=1}^{M} n_{ij}(T) \cdot \sigma^{*2}}, \\
&\leq \sigma^* \sum_{i=2}^{K} \frac{\sqrt{\sum_{j=1}^{M} n_{ij}(T) 2L \, \sigma^*}}{\sqrt{\Delta_i}} \frac{\sqrt{\Delta_i}}{\sqrt{2L\sigma^*}}, \\
&\leq \sigma^* \sqrt{\sum_{i=2}^{K} \frac{2L\sigma^*}{\Delta_i}} \sqrt{\sum_{i=2}^{K} \sum_{j=1}^{M} \frac{n_{ij}(T)\Delta_i}{2L\sigma^*}}, \\
&\leq \frac{\sigma^*}{2} \left[ \sum_{i=2}^{K} \frac{2L\sigma^*}{\Delta_i} + \sum_{i=2}^{K} \sum_{j=1}^{M} \frac{n_{ij}(T)\Delta_i}{2L\sigma^*} \right] = \sum_{i=2}^{K} \frac{L\sigma^{*2}}{\Delta_i} + \frac{\text{Reg}_T}{4L} \quad \left[ \text{where } L = 4\sqrt{3\log(3KT/\delta)} \right].
\end{aligned}
$$

The final bound on term B is: $B \leq \sum_{i=2}^{K} \frac{L\sigma^{*2}}{2\Delta_i} + \frac{\text{Reg}_T}{4L}$ where $L = 4\sqrt{3\log(3KT/\delta)}$.

**Bounding Term C:**

$$C = \sum_{i=2}^{K} \sqrt{\sum_{j|\sigma_j - \sigma^* \leq \gamma} n_{ij}(T)\, \sigma_j^2} = \sum_{i=2}^{K} \sqrt{\sum_{j|\sigma_j - \sigma^* \leq \gamma} n_{ij}(T)\, (\gamma + \sigma^*)^2}.$$

Proceeding in a similar fashion to how we bounded term B, we get

The final bound on term C is: $C \leq \sum_{i=2}^{K} \dfrac{L(\sigma^* + \gamma)^2}{\Delta_i} + \dfrac{\text{Reg}_T}{4L}$ where $L = 4\sqrt{3\log(3KT/\delta)}$.

**Putting it All Together:** Hence the regret in $T - M\tau_p - K\alpha - \tilde{M}\beta$ rounds of SOAR:

$$\frac{\text{Reg}_T}{2L} \leq 2\sqrt{K}\sqrt{KMc^{*2} + \sum_{j|\sigma_j-\sigma^*>\gamma} \frac{16\, Q_j(T)\log(3MT/\delta)}{(\gamma + 2\sigma^*)^2} + \sum_{i=2}^{K}\frac{L\sigma^{*2}}{\Delta_i} + \sum_{i=2}^{K}\frac{L(\sigma^*+\gamma)^2}{\Delta_i}},$$

$$\frac{\text{Reg}_T}{2L} \leq 2\sqrt{K}\sqrt{KMc^{*2} + \sum_{j|\sigma_j-\sigma^*>\gamma} \frac{16\, Q_j(T)\log(3MT/\delta)}{(\gamma + 2\sigma^*)^2} + 2\sum_{i=2}^{K}\frac{L(\sigma^*+\gamma)^2}{\Delta_i}} \quad [\text{As } \gamma \in \mathbb{R}_+].$$

$$\text{Reg}_T \leq 4L\sqrt{K}\sqrt{KMc^{*2} + \sum_{j|\sigma_j-\sigma^*>\gamma} \frac{16\, Q_j(T)\log(3MT/\delta)}{(\gamma + 2\sigma^*)^2} + 4\sum_{i=2}^{K}\frac{L^2(\sigma^*+\gamma)^2}{\Delta_i}}$$

**Total Regret:** The total regret of SOAR $= \text{Reg}_T^{\text{Exp}} + \text{Reg}_T$

$$\text{Reg}_T^{\text{SOAR}} = M\tau_p\bar{\mu} + K\alpha\bar{\mu} + M\beta\bar{\mu} + 4L\sqrt{K}\sqrt{KMc^{*2} + \sum_{j|\sigma_j-\sigma^*>\gamma} \frac{16\, Q_j(T)\log(3MT/\delta)}{(\gamma + 2\sigma^*)^2} + \sum_{i=2}^{K}\frac{4L^2(\sigma^*+\gamma)^2}{\Delta_i}}.$$

where, recalling the exploration budgets from Section 4, $\tau_p \geq \dfrac{1024\bar{\eta}^4\log(12M/\delta)}{c^{*4}}$, $\alpha \geq \dfrac{\bar{\eta}^2\log(3KT/\delta)}{c^{*2}}$, and $\beta \geq 2K + \dfrac{4\bar{\eta}^4\log(3MT/\delta)}{\nu} + \dfrac{16\bar{\eta}^4\log(3MT/\delta)}{c^{*4}}$, and $L := 4\sqrt{3\log(3KT/\delta)}$.

Collecting the exploration and adaptive-phase contributions, the total regret of SOAR is of order

$$\tilde{O}(\text{Reg}_T^{\text{SOAR}}) = \tilde{O}\left(\frac{M\bar{\eta}^4\bar{\mu}}{c^{*4}} + \frac{K\bar{\eta}^2\bar{\mu}}{c^{*2}} + KM\bar{\mu} + \frac{M\bar{\eta}^4\bar{\mu}}{\nu}\right)$$

$$+ \tilde{O}\left(\sqrt{K}\sqrt{KMc^{*2} + \sum_{j|\sigma_j-\sigma^*>\gamma} \frac{16\, Q_j(T)}{(\gamma + 2\sigma^*)^2} + \sum_{i=2}^{K}\frac{(\sigma^*+\gamma)^2}{\Delta_i}}\right).$$

**Instance Independent (Worst Case) Regret:** This proves the first part of Theorem 5.1, which yields an instance-dependent guarantee of SOAR. Finally, to see the worst case (instance independent) regret analysis of SOAR, note that the regret in the final $T - M\tau_p - K\alpha - \tilde{M}\beta$ rounds of SOAR can be alternatively bounded as:

$$\text{Reg}_T = \sum_{i=2}^{K} n_i(T)\Delta_i \leq \sum_{i=2}^{K} 4\sqrt{3\log(3KT/\delta)\sum_{j=1}^{M} n_{ij}(T)\sigma_j^2},$$

$$\leq \sqrt{\sigma^{*2}+c^{*2}}\sum_{i=2}^{K} 4\sqrt{3\log(3KT/\delta)\sum_{j=1}^{M} n_{ij}(T)}\quad (\sigma_j \leq \sqrt{\sigma^{*2}+c^{*2}}),$$

$$\leq 4\sqrt{\sigma^{*2}+c^{*2}}\sqrt{3K\log(3KT/\delta)}\sqrt{\sum_{i=2}^{K}\sum_{j=1}^{M} n_{ij}(T)}\quad (\text{applying Cauchy's Schwarz}),$$

$$\leq 4(\sigma^*+c^*)\sqrt{3KT\log(3KT/\delta)}\quad (\text{as } \sqrt{a+b} \leq \sqrt{a} + \sqrt{b}).$$

Therefore the order the instance independet regret is

$$\tilde{O}(\text{Reg}_T^{\text{Exp}} + (\sigma^* + c^*)\sqrt{KT}$$
$$\equiv \tilde{O}\left(\frac{M\bar{\eta}^4\bar{\mu}}{c^{*4}} + \frac{K\bar{\eta}^2\bar{\mu}}{c^{*2}} + KM\bar{\mu} + \frac{M\bar{\eta}^4\bar{\mu}}{\nu} + (\sigma^* + c^*)\sqrt{KT}\right).$$

proving the final claim of Theorem 5.1.

## D.2. Regret Proof in Alternate Regime

**Theorem 5.4** (Regret Analysis of SOAR in alternate regime). *Consider $\delta \in (0,1)$. Assume that all sources $j \in S_{\mathcal{G}} \subseteq [M]$ are such that $\sigma_j^2 < c^{*2}$ where $c^* \in \mathbb{R}_+$ is a user-defined parameter of the order of $\sigma_j$.*

*Then for any choice of preprocessing budget $\tau_p \geq \frac{1024 \cdot \bar{\eta}^4 \log(12M/\delta)}{c^{*4}}$, inital exploration parameter $\alpha \geq \bar{\eta}^2 c^{*2} \log(3KT/\delta)$, then the regret of SOAR (Algorithm 4) can be bounded by $\tilde{O}\left(\frac{M\bar{\eta}^4\bar{\mu}}{c^{*4}} + K\bar{\eta}^2 c^{*2}\bar{\mu} + \sum_{i=2}^K \frac{c^{*2}}{\Delta_i}\right)$ with high probability $(1-\delta)$. In this regime, SOAR can also be shown to yield an instance-independent (worst-case) regret bound of*

$$\tilde{O}\left(\frac{M\bar{\eta}^4\bar{\mu}}{c^{*4}} + K\bar{\eta}^2 c^{*2}\bar{\mu} + Kc^{*2}\sqrt{T}\right)$$

*Proof.* To analyze the regret of this regime, we begin by deriving a lemma for the mean reward concentration similar to Lemma 4.6.

**Lemma D.1** (Mean-Reward Concentration: Case 2). *Consider any $\delta \in (0,1)$. If there exists no $j$ such that $\sigma_j^2 \geq c^{*2}$ then for any time step $t \in [T], i \in [K]$, and any realization of $\alpha > \bar{\eta}^2 c^* \log(3KT/\delta)$, with probability at least $1 - \frac{\delta}{3}$:*

$$|\mu_i - \hat{\mu}_t(i)| \leq \frac{2c^*\sqrt{\log(3KT/\delta)}}{\sqrt{n_i(t)}}.$$

*Proof.* Suppose $T_t(i) = \{s \leq t : i_s = i\}$ denotes the set of time steps at which arm $i$ was selected up to time $t$. Consider the sequence of random variables defined by

$$D_{t,i} := X_{t,i} - \mu_i, \quad \text{for all } t \in T_t(i).$$

To establish a concentration bound, the key observation is that $\{D_{t,i}\}_{t \in T_t(i)}$ forms a martingale difference sequence with respect to the filtration $\mathcal{F}_{t-1} = \sigma\left(\{X_s, i_s, j_s\}_{s=1}^{t-1}, i_t = i\right)$, generated by all arm selections and observations prior to round $t$. Precisely note that each $D_{t,i}$ is integrable, adapted to $\mathcal{F}_t$, and satisfies $\mathbf{E}[D_{t,i} \mid \mathcal{F}_{t-1}] = \mathbf{E}[X_{t,i} - \mu_i \mid \mathcal{F}_{t-1}] = \mu_i - \mu_i = 0$. Thus, by construction, the sequence satisfies the martingale difference property and enables us to leverage standard martingale concentration inequalities.

**Theorem D.2** (Freedman's Inequality, (Raban)). *Let $\{(D_k, \mathcal{F}_k)\}$ be a martingale difference sequence such that*

1. $\mathbb{E}[D_k \mid \mathcal{F}_{k-1}] = 0$

2. $D_k \leq b$.

*Then for all $\lambda \in (0, 1/b)$ and $\delta \in (0,1)$,*

$$\mathbb{P}\left(\sum_{k=1}^T D_k \leq \lambda \sum_{k=1}^T \mathbb{E}[D_k^2 \mid \mathcal{F}_{k-1}] + \frac{\log(1/\delta)}{\lambda}\right) \geq 1 - \delta.$$

We have a bound $\bar{\eta}$ on $D_{t,i}$, and the first requirement of the theorem has already been proven. Applying Freedman's, we get

$$\mathbb{P}\left(\sum_{t \in T_i(t)} D_{t,i} \leq \lambda \sum_{t \in T_i(t)} \mathbb{E}[D_{t,i}^2|\mathcal{F}_{t-1}] + \frac{\log(3/\delta)}{\lambda}\right) \geq 1 - \frac{\delta}{3},$$

$$\mathbb{P}\left(\sum_{t \in T_i(t)} D_{t,i} \leq \lambda \sum_{j=1}^{M} n_{ij}(t) \cdot \sigma_j^2 + \frac{\log(3/\delta)}{\lambda}\right) \geq 1 - \frac{\delta}{3}. \quad \left[\mathbb{E}[D_{t,i}^2|\mathcal{F}_{t-1}] = \sigma_j^2\right]$$

Choosing $\lambda^* = \dfrac{\sqrt{\log(3/\delta)}}{\sqrt{c^{*2} \cdot n_i(t)}}$.

$$\mathbb{P}\left(\sum_{t \in T_i(t)} D_{t,i} \leq \frac{\sqrt{\log(3/\delta)} \sum_{j=1}^{M} n_{ij}(t) \cdot \sigma_j^2}{\sqrt{c^{*2} \cdot n_i(t)}} + \frac{\sqrt{c^{*2} \cdot n_i(t)}}{\sqrt{\log(3/\delta)}} \cdot \log(3/\delta)\right) \geq 1 - \frac{\delta}{3},$$

$$\mathbb{P}\left(\sum_{t \in T_i(t)} D_{t,i} \leq \frac{\sqrt{\log(3/\delta)} \sum_{j=1}^{M} n_{ij}(t) \cdot c^*}{\sqrt{n_i(t)}} + c^* \sqrt{n_i(t) \cdot \log(3/\delta)}\right) \geq 1 - \frac{\delta}{3}, \quad [\sigma_j^2 < c^*]$$

$$\mathbb{P}\left(\sum_{t \in T_i(t)} D_{t,i} \leq 2c^* \sqrt{n_i(t) \log(3/\delta)}\right) \geq 1 - \frac{\delta}{3}.$$

Dividing both sides by $n_i(t)$, union bounding over all arms and across all timesteps, we get our mean concentration bound w.p $1 - \dfrac{\delta}{3}$

$$\left|\hat{\mu}_i(t) - \mu_i\right| \leq \frac{2c^* \sqrt{\log(3KT/\delta)}}{\sqrt{n_i(t)}}.$$

Now to satisfy the Freedman constraint we need:

$$\lambda^* = \frac{c^* \sqrt{\log(3KT/\delta)}}{\sqrt{n_i(t)}} < \frac{1}{\bar{\eta}},$$

$$n_t(i) > \bar{\eta}^2 c^{*2} \log(3KT/\delta).$$

That is, each arm $i \in [K]$ is pulled at least $\alpha = \dfrac{\bar{\eta}^2 \log(3KT/\delta)}{c^{*2}}$ times to ensure that the mean concentration bound and the subsequently derived UCB estimate Equation (16) hold with high probability.

$\square$

**UCB Equation:** The best arm can be identified via an Upper Confidence Bound estimate using the derived mean reward concentration Lemma D.1, specifically

$$\text{UCB}_t^\mu(i) := \hat{\mu}_t(i) + \frac{2c^* \sqrt{\log(3KT/\delta)}}{\sqrt{n_t(i)}}. \tag{16}$$

Note that under our regime, we may treat every surviving source as having effective variance $c^*$ and then apply a standard UCB multi-armed bandit analysis using Equation (16). With that in mind, let us returning back to the proof of Theorem 5.4, where the regret can be derived as follows:

**Bounding $n_i(t)$** Note that at any time $t$, arm-$i$ can not be selected if $\text{UCB}_t^\mu(i) \leq \text{UCB}_t^\mu(i^*)$. So arm-$i$ only gets selected at time $t$ if:

$$\mu_{i^*} \leq \text{UCB}_t^\mu(i^*) \leq \text{UCB}_t^\mu(i) = \hat{\mu}_t(i) + \frac{2c^*\sqrt{\log(3KT/\delta)}}{\sqrt{n_i(t)}},$$

$$\mu_{i^*} \leq \mu_t(i) + \frac{4c^*\sqrt{\log(3KT/\delta)}}{\sqrt{n_i(t)}}, \quad [\text{Using Lemma D.1}]$$

$$\implies n_i(t) \leq \frac{16c^{*2}\log(3KT/\delta)}{\Delta_i^2}.$$

**Total Regret :** We begin by deriving the regret incurred from the exploration phase.

$$\text{Reg}_T^{\text{Exp}} = M\tau_p\bar{\mu} + K\alpha\bar{\mu}$$
$$= M\left(\frac{1024\bar{\eta}^4\log(12M/\delta)}{c^{*4}}\right)\bar{\mu} + K\left(\bar{\eta}^2c^{*2}\log(3KT/\delta)\right)\bar{\mu}.$$

The Regret in $T - K\alpha$ rounds of Running SOAR in this regime :

$$\text{Reg}_T = \sum_{i=2}^K n_i(T)\Delta_i \leq \sum_{i=2}^K \frac{16c^{*2}\log(3KT/\delta)}{\Delta_i},$$

$$\leq 16c^{*2}\log(3KT/\delta)\sum_{i=2}^K \frac{1}{\Delta_i},$$

$$\leq \frac{c^{*2}L^2}{6}\sum_{i=2}^K \frac{1}{\Delta_i}.$$

where $L = 4\sqrt{\log(3KT/\delta)}$

The total regret of SOAR in the alternate regime is $(\text{Reg}_T^{\text{SOAR}}) = \text{Reg}_T^{\text{Exp}} + \text{Reg}_T = M\tau_p\bar{\mu} + K\alpha\bar{\mu} + \sum_{i=2}^K \frac{c^{*2}L^2}{6\Delta_i}$

where, recalling the budgets from Section 4, $\tau_p \geq \frac{1024\bar{\eta}^4\log(12M/\delta)}{c^{*4}}$ and $\alpha \geq \bar{\eta}^2c^{*2}\log(3KT/\delta)$.

Collecting these terms, the total regret of SOAR is of order

$$\tilde{O}(\text{Reg}_T^{\text{SOAR}}) = \tilde{O}\left(\frac{M\bar{\eta}^4\bar{\mu}}{c^{*4}} + K\bar{\eta}^2c^{*2}\bar{\mu} + \sum_{i=2}^K \frac{c^{*2}}{\Delta_i}\right).$$

**Worst Case Bound:** This proves the first part of Theorem 5.4, which yields an instance-dependent guarantee of SOAR. Finally, to see the worst case (instance independent) regret analysis of SOAR, note that the regret in $T - M\tau_p - K\alpha - \tilde{M}\beta$ rounds of SOAR can be alternatively bounded as:

$$\text{Reg}_T \leq \frac{c^{*2}L^2(K-1)\sqrt{T}}{6}.$$

Therefore, the order of the instance-independent regret is

$$\tilde{O}(\text{Reg}_T^{\text{Exp}} + Kc^{*2}\sqrt{T}$$
$$\equiv \tilde{O}\left(\frac{M\bar{\eta}^4\bar{\mu}}{c^{*4}} + K\bar{\eta}^2c^{*2}\bar{\mu} + Kc^{*2}\sqrt{T}\right).$$

proving the final claim of Theorem 5.4 $\qquad\qquad\qquad\qquad\qquad\qquad\qquad\qquad\qquad\qquad\qquad\qquad$ $\square$

# E. Supplementary for Adaptive c*

In our paper, $c^* \in \mathbb{R}_+$ is a user-specified parameter that serves as a proxy for the unknown minimum source standard deviation $\sigma^* := \min_{j \in [M]} \sigma_j$. In this section we propose a method to eliminate $c^*$ from algorithm specification: we construct a n estimate $\hat{\sigma}^{*2}$ from the PREPROCESS variance estimates that concentrates around $\sigma^{*2}$, derive the resulting regret bound in terms of $\sigma^*$ directly, and provide a principled floor for the degenerate regime $\sigma^* \to 0$.

## E.1. Role of $c^*$ in the Regret of Theorem 5.1

We first recall precisely how $c^*$ enters the regret bound of Theorem 5.1. The parameter $c^*$ appears through three distinct components of the analysis:

1. PREPROCESS (Theorem 4.3), which contributes $\tilde{O}\left(\dfrac{M\bar{\eta}^4}{c^{*4}}\right)$ to the regret.

2. Mean reward concentration (Lemma 4.6), which contributes $\tilde{O}\left(\dfrac{K\bar{\eta}^2}{c^{*2}}\right)$ to the regret.

3. Variance sandwiching (Corollary 4.5), which contributes $\tilde{O}\left(\dfrac{M\bar{\eta}^4}{c^{*4}}\right)$ to the regret.

If $\sigma^*$ were known, the natural choice would be $c^* = \sigma^*$. The remainder of this section makes this idea rigorous by constructing an adaptive proxy for $\sigma^*$ from the PREPROCESS variance estimates.

## E.2. An Adaptive Proxy for $\sigma^*$

Recall the preprocessing LCB on the source variance defined in Equation (4):

$$\mathrm{LCB}_{\tau_p}^{\sigma^2,\mathrm{pre}}(j) = \max\left\{\hat{\sigma}_{j,\mathrm{pre}}^2(\tau_p) - 8\bar{\eta}^2\sqrt{\frac{\log(12M/\delta)}{\tau_p}}, \, 0\right\}.$$

We define the adaptive proxy and corresponding source index as

$$\hat{\sigma}^{*2} := \min_{j \in [M]} \mathrm{LCB}_{\tau_p}^{\sigma^2,\mathrm{pre}}(j), \qquad \hat{j}^* := \arg\min_{j \in [M]} \mathrm{LCB}_{\tau_p}^{\sigma^2,\mathrm{pre}}(j).$$

That is, $\hat{\sigma}^{*2}$ is the smallest LCB across all sources at the end of preprocessing, and $\hat{j}^*$ is the source attaining this minimum.

The following lemma shows that, after a sufficient preprocessing budget, the adaptive proxy $\hat{\sigma}^{*2}$ sandwiches $\sigma^{*2}$ up to a factor of two.

**Lemma E.1** (Concentration of the Adaptive Proxy). *Fix $\delta \in (0,1)$. Suppose PREPROCESS is run with budget*

$$\tau_p \geq \frac{1024\,\bar{\eta}^4\,\log(12M/\delta)}{\sigma^{*4}}.$$

*Then, with probability at least $1 - \delta/3$,*

$$\frac{\sigma^{*2}}{2} \leq \hat{\sigma}^{*2} \leq \sigma^{*2}.$$

*Proof.* We prove the upper and lower inequalities in separately.

**Upper Bound: $\hat{\sigma}^{*2} \leq \sigma^{*2}$.** By Lemma 4.2, for every source $j \in [M]$ we have $\mathrm{LCB}_{\tau_p}^{\sigma^2,\mathrm{pre}}(j) \leq \sigma_j^2$ with probability at least $1 - \delta/3$. In particular, this holds for the optimal source $j^*$, so $\mathrm{LCB}_{\tau_p}^{\sigma^2,\mathrm{pre}}(j^*) \leq \sigma_{j^*}^2 = \sigma^{*2}$. Taking the minimum over $j \in [M]$ on the left-hand side preserves the inequality:

$$\hat{\sigma}^{*2} = \min_{j \in [M]} \mathrm{LCB}_{\tau_p}^{\sigma^2,\mathrm{pre}}(j) \leq \mathrm{LCB}_{\tau_p}^{\sigma^2,\mathrm{pre}}(j^*) \leq \sigma^{*2}.$$

**Lower Bound:** $\hat{\sigma}^{*2} \geq \sigma^{*2}/2$. Let $\hat{j}^*$ denote the source achieving the minimum in (E.2). By definition,

$$\hat{\sigma}^{*2} = \text{LCB}_{\tau_p}^{\sigma^2,\text{pre}}(\hat{j}^*) = \hat{\sigma}_{\hat{j}^*,\text{pre}}^2(\tau_p) - 8\bar{\eta}^2\sqrt{\frac{\log(12M/\delta)}{\tau_p}}. \tag{17}$$

We now lower-bound the empirical variance $\hat{\sigma}_{\hat{j}^*,\text{pre}}^2(\tau_p)$. By Lemma 4.2, with probability at least $1 - \delta/3$,

$$\hat{\sigma}_{\hat{j}^*,\text{pre}}^2(\tau_p) \geq \sigma_{\hat{j}^*}^2 - 8\bar{\eta}\,\sigma_{\hat{j}^*}\sqrt{\frac{\log(12M/\delta)}{\tau_p}} \geq \sigma^{*2} - 8\bar{\eta}^2\sqrt{\frac{\log(12M/\delta)}{\tau_p}},$$

where the last inequality uses $\sigma_{\hat{j}^*} \geq \sigma^*$ and $\sigma_{\hat{j}^*} \leq \bar{\eta}$ (Assumption 2.1). Substituting into (17) gives

$$\hat{\sigma}^{*2} \geq \sigma^{*2} - 16\bar{\eta}^2\sqrt{\frac{\log(12M/\delta)}{\tau_p}}.$$

By the choice of $\tau_p$ stated in the theorem,

$$16\bar{\eta}^2\sqrt{\frac{\log(12M/\delta)}{\tau_p}} \leq 16\bar{\eta}^2 \cdot \sqrt{\frac{\sigma^{*4}}{1024\,\bar{\eta}^4}} = \frac{\sigma^{*2}}{2}.$$

Which in turn yields $\hat{\sigma}^{*2} \geq \sigma^{*2} - \sigma^{*2}/2 = \sigma^{*2}/2$.

A union bound over the two events used shows that both upper and lower bound hold simultaneously with probability at least $1 - \delta/3$. $\qquad\square$

*Remark* E.2. The budget $\tau_p = \Omega(\bar{\eta}^4 \log(M/\delta)/\sigma^{*4})$ required by Lemma E.1 is exactly of the same order as the budget already prescribed by Theorem 4.3, with $c^*$ replaced by $\sigma^*$. Hence the concentration in (E.1) is achieved without requiring prior knowledge of $\sigma^*$: one simply runs PREPROCESS for the budget it already prescribes, and computes $\hat{\sigma}^{*2}$ from the resulting variance estimates.

## E.3. Regret Bound with the Adaptive Proxy

We are now in a position to state the regret bound of SOAR when the fixed parameter $c^*$ is replaced by the adaptive proxy $\hat{\sigma}^{*2}$ defined in (E.2).

**Theorem E.3** (Regret of SOAR with Adaptive $c^*$)**.** *Consider $\delta \in (0,1)$ and assume there exists a source $j \in S_\mathcal{G} \subseteq [M]$ with $\sigma_j^2 \geq \sigma^{*2}$. Suppose SOAR is run with $c^*$ replaced by the adaptive proxy $\hat{\sigma}^{*2}$ defined in (E.2) throughout. Then, with probability at least $1 - \delta$, the regret of SOAR is bounded by*

$$\tilde{O}\left(\frac{M\bar{\eta}^4\bar{\mu}}{\sigma^{*4}} + \frac{K\bar{\eta}^2\bar{\mu}}{\sigma^{*2}} + KM\bar{\mu} + \frac{M\bar{\eta}^4\bar{\mu}}{\nu} + \sigma^*\sqrt{KT}\right).$$

*Proof Sketch.* By Lemma E.1, with probability at least $1 - \delta/3$, we have $\sigma^{*2}/2 \leq \hat{\sigma}^{*2} \leq \sigma^{*2}$. Substituting this two-sided bound in place of $c^{*2}$ in the proof of Theorem 5.1:

- The PREPROCESS term $\tilde{O}(M\bar{\eta}^4/c^{*4})$ becomes $\tilde{O}(M\bar{\eta}^4/\sigma^{*4})$.

- The mean-reward concentration term $\tilde{O}(K\bar{\eta}^2/c^{*2})$ becomes $\tilde{O}(K\bar{\eta}^2/\sigma^{*2})$.

- The variance-sandwiching term $\tilde{O}(M\bar{\eta}^4/c^{*4})$ becomes $\tilde{O}(M\bar{\eta}^4/\sigma^{*4})$.

- The leading $(\sigma^* + c^*)\sqrt{KT}$ in the instance-independent bound becomes $\sigma^*\sqrt{KT}$, since $\hat{\sigma}^* \leq \sigma^*$ is absorbed into the leading term.

Combining the four bullet points with the exploration and sub-optimality terms from Theorem 5.1 gives the stated bound. $\quad\square$

**E.4. Safeguarding Against $\sigma^* \to 0$**

The adaptive proxy $\hat{\sigma}^{*2}$ alone does not handle the degenerate regime $\sigma^* \to 0$: as $\sigma^*$ shrinks, the preprocessing and exploration terms in Lemma E.1 blow up. To remedy this, we combine the adaptive proxy with a small floor:

$$\hat{\sigma}^{*2} := \max\left\{ \min_{j \in [M]} \text{LCB}_{\tau_p}^{\sigma^2,\text{pre}}(j), \ c^{*2} \right\}. \tag{18}$$

We now derive a principled value for the floor $c^*$ that does not require prior knowledge of $\sigma^*$.

**Deriving the Optimal Floor $c^* = O(T^{-1/10})$.** Assume $\bar{\mu} = O(1)$. We focus on the regime $\sigma^* < 1$ which is precisely the case in which a floor is needed (when $\sigma^* \geq 1$, Theorem E.3 already gives a satisfactory bound). In this regime, $\frac{K\bar{\eta}^2}{c^{*2}}$ is dominated by $\frac{(M+K)\bar{\eta}^4}{c^{*4}}$ for small $c^*$.

Substituting (18) into the worst case bound of Theorem 5.1, the instance-independent regret simplifies to

$$\tilde{O}\left( \min\left\{ \frac{(M+K)\bar{\eta}^4}{\sigma^{*4}}, \frac{(M+K)\bar{\eta}^4}{c^{*4}} \right\} + \max\left\{ \min_{j \in [M]} \text{LCB}_{\tau_p}^{\sigma^2,\text{pre}}(j), \ c^{*2} \right\} \sqrt{KT} + KM\bar{\mu} + \frac{M\bar{\eta}^4}{\nu} \right).$$

Since $\sigma^* < 1$, we have $c^* > \sigma^*$ in this regime, and we can further upper bound the expression above by

$$\tilde{O}\left( \frac{(M+K)\bar{\eta}^4}{c^{*4}} + c^*\sqrt{KT} + KM\bar{\mu} + \frac{M\bar{\eta}^4}{\nu} \right). \tag{19}$$

Optimizing $f(c^*) := \frac{(M+K)\bar{\eta}^4}{c^{*4}} + c^*\sqrt{KT}$ over $c^*$:

$$f'(c^*) = -\frac{4(M+K)\bar{\eta}^4}{c^{*5}} + \sqrt{KT} = 0 \implies c^*_{\text{opt}} = \left( \frac{4(M+K)\bar{\eta}^4}{\sqrt{KT}} \right)^{1/5}.$$

Hence $c^*_{\text{opt}} = O(T^{-1/10})$. Substituting back yields

$$\frac{(M+K)\bar{\eta}^4}{c^*_{\text{opt}}{}^4} = c^*_{\text{opt}}\sqrt{KT} = O\left( (M+K)^{1/5} \bar{\eta}^{4/5} (KT)^{2/5} \right).$$

Now substituting this in Equation (19) we get the final regret expression:

$$\tilde{O}\left( (M+K)^{1/5} \bar{\eta}^{4/5} (KT)^{2/5} + KM\bar{\mu} + \frac{M\bar{\eta}^4\bar{\mu}}{\nu} \right). \tag{20}$$

*Remark* E.4 (On the $(KT)^{2/5}$ rate). At first glance, the term of order $\tilde{O}\left( (M+K)^{1/5} \bar{\eta}^{4/5} (KT)^{2/5} \right)$ in Theorem E.3 appears to beat the classical $\Omega(\sqrt{KT})$ minimax lower bound for the $K$-armed bandit problem. We emphasize that this is *not* a violation of any known lower bound, for two reasons.

First, this branch is only active in the regime $\sigma^* \to 0$, since the bound in (20) is a minimum and the $\sigma^*$-dependent branch already dominates when $\sigma^*$ is bounded away from zero. The classical $\Omega(\sqrt{KT})$ lower bound, by contrast, is established under the implicit assumption that the reward noise is bounded away from zero (typically a constant variance $\sigma^2 = \Omega(1)$) (Lattimore & Szepesvári, 2020); it does not apply in the vanishing-noise regime.

Second, consider the case of a single source and a set of arms with variance zero. This reduces to the standard MAB, but per the optimal regret rate the regret is zero. Similarly here we are not really violating any bound; this is simply the case where the true $\sigma^*$ is very, very small.

*Remark* E.5 (Refined choice of $c^*$). For clarity, we upper-bounded $\frac{\bar{\eta}^2}{c^{*2}}$ by $\frac{\bar{\eta}^4}{c^{*4}}$ in the derivation of $c^*_{\text{opt}}$. A finer analysis can be obtained by retaining both terms separately and performing a case analysis on whether $\frac{K}{M} \lessgtr \frac{\bar{\eta}^2}{c^{*2}}$.

## F. Experimental Setup and Results

In this section, we describe the experimental setup and present our baseline comparison results along with the results obtained by varying the number of arms $K$ and the number of sources $M$.

## F.1. Variation in the Number of Arms and Sources

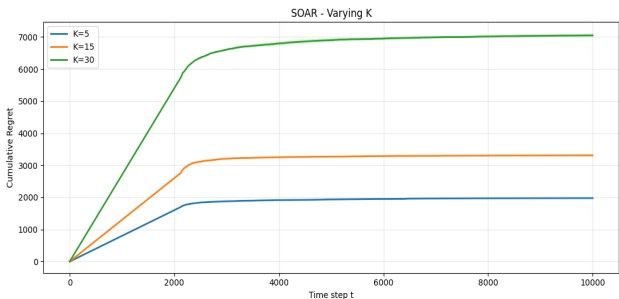

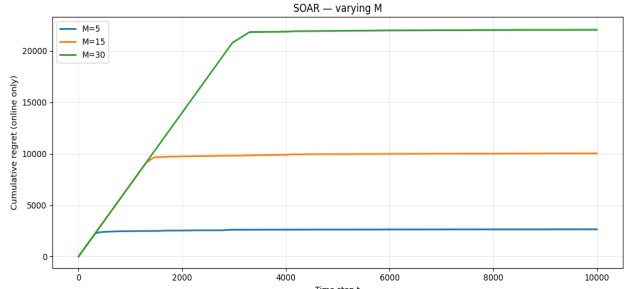

*Figure 4.* Regret of SOAR with varying number of arms $K \in \{5, 15, 30\}$

*Figure 5.* Regret of SOAR with varying number of sources $M \in \{5, 15, 30\}$

We evaluate SOAR on synthetic multi-source bandit tasks under two complementary setups. Arm rewards are modeled as Gaussian random variables with fixed means and source-dependent variances. Arm means are either drawn uniformly from $[1, 10]$ (rounded to one decimal place) or fixed as $\mu = [1, 5, 8, 6, 4]$ as described below. Source variances are specified directly or sampled uniformly from $[1, 3]$ (rounded to one decimal place). We can choose $\nu$ as per Remark 5.2 and $c^* = 2$. All runs are executed for a time horizon of $T = 10,000$. In both setups, we observe an initial linear growth in regret, which stems from the preprocessing phase combined with the initial exploration rounds.

**Varying $K$.** We fix the number of sources to $M = 3$ with variances $\sigma = [5, 1, 4]$, and vary the number of arms $K \in \{5, 15, 30\}$. For each configuration, arm means are drawn uniformly from $[1, 10]$ (rounded to one decimal place). This setup isolates how SOAR scales with the number of arms while holding source variability constant.

**Varying $M$.** We fix the arms to $\mu = [1, 5, 8, 6, 4]$ and vary the number of sources $M \in \{5, 15, 30\}$. The source variances are sampled independently from $[1, 3]$ (rounded to one decimal place). This setup isolates how SOAR scales with the number of sources while holding the arm structure fixed. The initial growth in regret clearly reflects the expected scaling with $M$, consistent with our theoretical analysis.

## F.2. Baseline Comparison Results

For both baselines, we set arm means in the range $[0, 1]$ and source variances in the range $[1, 5]$. Both scenarios are evaluated over $T = 20,000$ rounds.

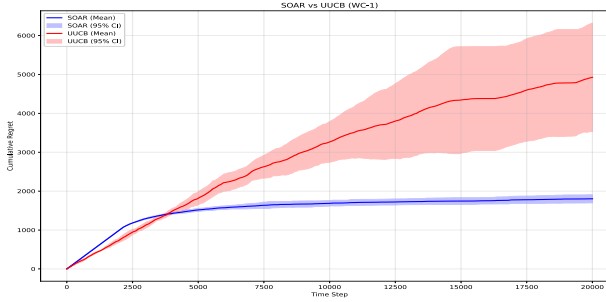

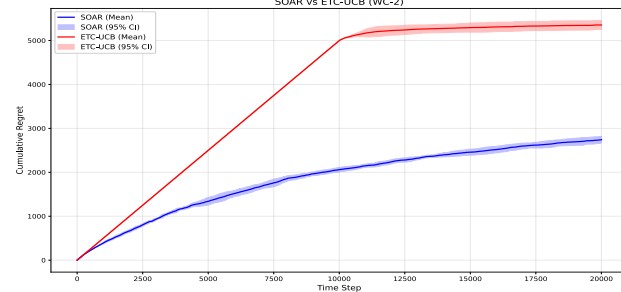

*Figure 6.* SOAR vs Baseline-1:UUCB WC-1

*Figure 7.* SOAR vs Baseline-2: ETC-UCB WC-2

**Baseline-1: UUCB.** For the Uniform-UCB baseline, we construct instances with multiple high-variance sources alongside a low-variance source (WC-1: $K = 5$, $M = 3$), creating a stark variance disparity. In this setting, SOAR initially incurs a higher cost due to its exploration phase, but then stabilizes and achieves substantially lower regret by adaptively prioritizing the low-variance source. In contrast, the uniform baseline continues to suffer from repeatedly sampling the high-variance sources throughout the horizon.

**Baseline 2: ETC-UCB.** For the Explore-Then-Commit UCB baseline, we consider sources with gradually increasing variances, introducing only incremental differences across sources (WC-2: $K = 10$, $M = 8$). Here, SOAR effectively handles the fine-grained variance differences by relying on continuous confidence bounds for adaptive source selection, whereas the two-phase baseline incurs significant regret due to its rigid elimination phase.

### F.3. MovieLens Panel

We evaluate source adaptivity on a real-world ratings dataset using a fixed panel constructed from the MovieLens 32M dataset (Harper & Konstan, 2015).

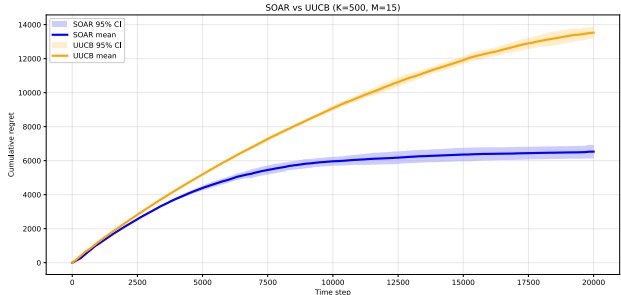
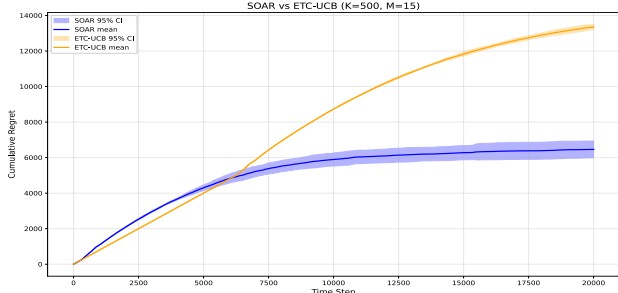

*Figure 8.* SOAR vs. Baseline-1:UUCB on MovieLens.  *Figure 9.* SOAR vs. Baseline-2:ETC-UCB on MovieLens.

We select $M = 15$ reviewers who have each rated the same set of $K = 500$ movies, yielding a total of 7,500 ratings. Each movie is treated as an arm, with its unknown mean reward $\mu_i \in [0, 5]$ corresponding to the average rating within the panel. Each reviewer is treated as a data source, and the empirical variance of their ratings over the selected movies is used as the source variance. Hence $\bar{\eta}^2 = 25$. We choose $c^* = 1, \nu = 30$. At each round, the learner selects a movie–reviewer pair and observes the corresponding rating. Regret is computed with respect to the movie with the highest empirical mean rating in the panel. We compare SOAR against the two baselines from Section 3: UUCB, which samples reviewers uniformly at random and applies UCB-V to the arms, and ETC-UCB, which first identifies a low-variance reviewer via uniform exploration and then runs UCB on the arms using only that reviewer. All algorithms are run for a fixed horizon ($T = 20,000$), and results are averaged over multiple independent runs with shaded regions indicating 95% confidence intervals.

Figure 8 and Figure 9 report cumulative regret as a function of time for SOAR and the two baselines. The figures show that SOAR consistently incurs lower cumulative regret than both baselines by rapidly concentrating sampling on low-variance reviewers while identifying highly rated movies. These results indicate that SOAR remains effective on real-world rating data, and not only on controlled synthetic benchmarks.

