# OpenReview forum: "Data-Source Adaptive Online Learning under Heteroscedastic Noise"
_ICML.cc/2026/Conference — ICML 2026 regular_

### Official Review · Reviewer_TAJz · 2026-03-02

**Soundness:** 3
**Presentation:** 4
**Significance:** 3
**Originality:** 4
**Overall Recommendation:** 4
**Confidence:** 4

**Summary:**

This paper studies the multi-source multi-armed bandit (MAB) setting. At each round, the algorithm selects both a source $j$ and an arm $i$. The chosen arm determines the mean of the reward, while the chosen source determines the reward distribution.

In standard stochastic MAB with variance $\sigma^2$, the regret lower bound scales as $\sigma^2 \sum_i \frac{\log T}{\Delta_i}$. The objective in this setting is to achieve an upper bound of $(\sigma^\*)^2 \sum_i \frac{\log T}{\Delta_i}$, where $\sigma^\*$ is the variance of the best source.

The paper first develops confidence bounds for both the mean and the variance. Estimating the variance is technically challenging because the variance depends on the unknown mean (i.e., $(r_t - \mu)^2$), and $\mu$ must itself be estimated. As a result, constructing valid confidence intervals for the variance requires more careful analysis than standard concentration bounds.

Using these confidence bounds, the proposed algorithm proceeds in three stages:

Preprocessing: Each source is sampled $\tau_p$ times (with random arms) to eliminate suboptimal sources.

Initial exploration: Each arm is sampled $\alpha$ times (using one of the active sources), and each active source is sampled $\beta$ times (with random arms).

Main phase: A UCB-style strategy is applied, selecting the arm with the highest upper confidence bound and the source with the lowest lower confidence bound on variance.

The paper also introduces two baseline algorithms:
(1) a uniform-source strategy, and
(2) a commit strategy that explores to identify an approximately optimal source and then commits to it.

The authors provide lower bounds showing these baselines are suboptimal and present empirical results on a real-world dataset demonstrating that their proposed method achieves higher rewards.

**Compliance With Llm Reviewing Policy:**

Affirmed.

**Final Justification:**

The paper investigates an interesting and largely unexplored question of multi-source MAB. The ideas appear original, and the overall direction is promising.

However, I have one major concern. The algorithm requires a parameter $c^\*$, and the paper does not explain how to estimate it. This is particularly problematic because the optimal choice of $c^\*$—needed to achieve the desired $(\sigma^\*)^2 \sum \log T / \Delta_i$ bound—is $\sigma^\*$ itself. In other words, the algorithm effectively assumes knowledge one of the important quantities it aims to adapt to.

This issue was not addressed in the rebuttal. Moreover, multiple reviewers raised this concern in their responses to the rebuttal, yet the authors did not provide a follow-up clarification. If the authors had clarified how they intend to deal with this (e.g., by assuming knowledge of $\sigma^*$ or proposing an adaptive approach), I would have considered it more carefully. As it stands, the lack of response makes this concern more significant.

Therefore, I decrease my score to 3.

*EDIT*: I find the late reply satisfactory and reincrease my score. I expect those ideas to be incorporated in the final version.

**Key Questions For Authors:**

1. The parameter $\gamma$ does not appear to influence the algorithm itself and seems to be used only in the analysis. Moreover, the bound appears to hold for any $\gamma$. Why is $\gamma$ treated as an input parameter rather than simply optimizing over it in the analysis (e.g., selecting the best $\gamma$ post hoc)?

2. Is there an intuitive reason for the preprocessing subroutine, or is it mainly required for technical purposes in the analysis?

3. How would the algorithm adapt to a setting where the source quality (variance) depends on the arm, i.e., where different sources have different variances for different arms?

**Limitations:**

yes

**Strengths And Weaknesses:**

**Soundness:** The technical development appears solid. One concern is the dependence on the parameter $c^\*$. While the introduction emphasizes a regret bound depending on $\sigma^\*$, the main theorem involves $(\sigma^* + c^\*)$. The discussion of how to choose or estimate $c^\*$ is minimal, with the suggestion that it should be “some small constant.” Since tuning such parameters is central to balancing exploration and exploitation, the lack of a principled method for selecting or adapting $c^\*$ weakens the completeness of the result. It would strengthen the paper to clarify whether $c^\*$ can be estimated adaptively (e.g., via a doubling trick) or whether the guarantees degrade gracefully under misspecification.

**Presentation:** Overall, the presentation is good. I appreciated the structure of first developing the confidence bounds and then building the algorithm on top of them. This separation makes the logic of the approach clear. There are, however, a few minor clarity issues:

1. Lemma 3.1 uses the term “sampling budget,” but it is unclear whether this refers to the total number of samples across all sources or per source. The notation $\tau_p$ initially suggests a global budget, while the algorithm later treats it as per-source, which caused confusion.

2. In Assumption 2.1, “all sourcet” appears to be a typo, and the support is stated to start from $\bar{\eta}$ rather than $-\bar{\eta}$.

3. In line 1305, term $B$ is misstated (though corrected immediately afterwards).

4. In line 1323, the second equality should be an inequality.

5. The text states “for any choice of $\tau_p > \dots$,” but $\tau_p$ does not appear in the final bound. This is confusing, since clearly extreme choices (e.g., $\tau_p = T$) would yield linear regret. Indeed, in the appendix (line 1351), $\tau_p$ appears to be fixed more precisely. This should be clarified in the main statement.

**Originality:**
The problem formulation is natural and, to the best of my knowledge, this is the first paper to address the multi-source MAB setting where the source controls variance and the goal is to achieve regret scaling with $(\sigma^\*)^2$. The development of variance confidence bounds in the presence of an unknown mean is technically nontrivial and constitutes a meaningful contribution.

**Significance:** The question studied is well-motivated: in many practical settings, data sources differ in reliability (variance), and exploiting low-variance sources while maintaining exploration is important. Achieving regret that scales with $(\sigma^\*)^2$ rather than a worst-case variance is a meaningful improvement. While the reliance on the parameter $c^\*$ slightly weakens the practical completeness of the result, the overall contribution is significant and advances the understanding of variance-aware bandits.

---

> ### Author Rebuttal · Authors · 2026-03-30
>
> We thank the reviewer for their careful reading and constructive feedback. We address each point below.
>
> ## Weaknesses
> >W1 (Adpative estimation of $c^{\*}$): The discussion of how to choose or estimate $c^{\*}$ is, $\dots$ misspecification.
>
> We thank the reviewer for this question. Choosing $c^{\*}$ too small increases the preprocessing and exploration budget $\propto 1/c^{\*4}$, resulting in higher upfront cost but leaving the dominant regret term $\tilde O\left({\sigma^{\*}}^2 \sum_{i \neq i^{\*}} \frac{1}{\Delta_i}\right)$ unaffected. On the other hand if $c^{\*}$ exceeds all source standard deviations, the algorithm enters the alternate regime of Theorem 5.4, where the dominant term becomes $\tilde O\left(c^{\*2} \sum_{i \neq i^*} \frac{1}{\Delta_i}\right)$, scaling with $c^{\*2}$ rather than $\sigma^{*2}$. Thus the algorithm is robust to underestimation of $c^{\*}$ (at the cost of preprocessing overhead), but overestimation can inflate the leading regret term, motivating a conservatively small choice in practice. Crucially, misspecification of $c^{\*}$ does not cause catastrophic failure.
>
> Regarding adaptivity, we recall the instance-independent regret expression $\tilde O\Bigg( \frac{ M\bar \eta^4 \bar \mu}{{c^{\*}}^4} + {\frac{K\bar \eta^2 \bar \mu}{{c^{\*}}^2}} + {KM \bar \mu + \frac{M\bar \eta^4 \bar \mu}{\nu} + \frac{M \bar \eta^4 \bar \mu}{{c^{\*}}^4}} + (\sigma^* + c^{\*}) \sqrt{KT}\Bigg)$. Taking $\bar \mu = 1$, all quantities apart from $\sigma^{\*}$ are known, so one could optimize this expression using an estimate $\hat \sigma_* = \min_{j\in [M]} \text{LCB}^{\sigma}_t (j) + \epsilon$ for very small $\epsilon > 0$. However, our concentration inequalities and UCB/LCB constructions were derived under a fixed $c^{\*}$ and need not hold under an adaptive scheme, introducing analytical complications that may preclude straightforward adaptation.
>
> We view principled adaptive selection of $c^{\*}$ as an interesting direction for future work.
>
> >W2: Presentation issues
>
> We sincerely thank the reviewer for the careful reading. We clarify that $\tau_p$ is a per-source budget; each source is queried $\tau_p$ times (Algorithm 1, line 4), for a total preprocessing cost of $M \cdot \tau_p$. We will make this explicit and fix the corresponding theorem statement in revision. We also acknowledge the typos in Assumption 2.1 ($[\bar \eta, \bar \eta] \to [-\bar \eta, \bar \eta]$) and elsewhere, and will correct them.
>
> ## Key Questions for Authors
>  >Q1 (Clarification on the role of $\gamma$): The parameter $\gamma \dots \gamma$ post hoc)?
>
> The reviewer is correct that $\gamma$ does not influence the algorithm's execution. As discussed in Remark 5.2 (point 2), $\gamma$ handles the degenerate regime where $\sigma^* \to 0$ and, since the bound holds for any $\gamma > 0$, can be optimized post hoc. We agree that listing $\gamma$ as an input to Algorithm 2 is misleading and will move its introduction to the regret analysis in the revision.
>
> >Q2: Is there an intuitive reason for the preprocessing subroutine, or is it mainly required for technical purposes in the analysis?
>
> PREPROCESS serves both an intuitive and a technical purpose.
> Intuitively, it acts as a coarse filter that removes sources whose noise is too large to aid learning, directly addressing the failure mode of Baseline-1 (Uniform Source MAB) specifically Worst-Case 1, where high-variance sources dominate performance. Eliminating these sources upfront prevents the adaptive phase from wasting pulls on high-variance sources which was the very scenario that caused Baseline 1 to yield regret scaling with $\sigma_{\text{max}}$ rather than $\sigma^{\*}$.
>
> Technically, PREPROCESS ensures all surviving sources satisfy $\sigma_j^2 \leq {\sigma^{\*}}^2 + {c^{\*}}^2$. This is critical in two parts of the regret proof. First, in the instance-dependent bound (Term A, lines 1283–1293): without pruning, $K(\sigma_j - \sigma^{\*})^2$ could grow with the worst source's variance, but PREPROCESS and some algebra allows us to bound this term by $K{c^{\*}}^2$. Second, in the instance-independent bound (lines 1377–1387): bounding $\sigma_j \leq \sigma^{\*} + c^{\*}$ for all surviving sources enables the $\tilde O \left( \left(\sigma^{\*}+c^{\*}\right) \sqrt{KT} \right)$ worst-case rate.
>
> >Q3: How would the algorithm adapt to a setting where the source quality (variance) depends on the arm, i.e., where different sources have different variances for different arms?
>
> We thank the reviewer for this interesting suggestion. Our paper specifically studies the setting where source variances are arm-independent, which is the natural model for applications such as clinical trials across hospitals with differing measurement reliability or recommender systems with user populations of varying feedback quality. The arm-dependent variance setting is a fundamentally different problem requiring new algorithmic and analytical tools, and we view it as a compelling direction for future work.

---

> > ### Author Rebuttal · Reviewer_TAJz · 2026-04-01
> >
> > The rebuttal clarifies that underestimating $c^*$ mainly leads to a longer preprocessing phase. However, I am not fully convinced this resolves the concern. The preprocessing phase contributes non-negligibly to the regret, so extending it can meaningfully affect performance.
> >
> > More importantly, I still find it unclear why the paper does not suggest a principled way to choose or estimate $c^*$, or alternatively present a regret bound that removes this dependence (as mentioned in the rebuttal as being possible). Relying on an unspecified “small constant” weakens the completeness of the result. In general, allowing algorithms to assume such parameters without guidance risks trivializing the problem - for example, one could introduce oracle knowledge of problem-dependent quantities and obtain improved bounds without addressing the core estimation challenge.

---

> > > ### Author Response · Authors · 2026-04-08
> > >
> > > ## Re. Concern about Adaptive $c^\*$
> > >
> > > We thank the reviewer for their careful reading and excellent suggestions, which meaningfully improve our paper.
> > >
> > > ---
> > >
> > > 1. **Quantifying the role of $c^{\*}$ in regret:** We first clarify how $c^{\*}$ enters the regret bound. It does so through three components:
> > >    - PREPROCESS (Theorem 4.1), contributing $\tilde O\left(\frac{M\bar{\eta}^4}{c^{\*4}}\right)$ to the regret.
> > >    - Mean reward concentration (Lemma 3.5), contributing $\tilde O\left(\frac{K\bar{\eta}^2}{c^{\*2}}\right)$ to the regret.
> > >    - Variance sandwiching (Corollary 3.4), contributing $\tilde O\left(\frac{M\bar{\eta}^4}{c^{\*4}}\right)$ to the final regret.
> > >
> > >    If $\sigma^\*$ were known, the natural choice would be $c^{\*} = \sigma^*$.
> > >
> > > 2. **An adaptive proxy for $\sigma^\*$:** In the paper, $c^{\*}$ serves as a fixed proxy for the unknown $\sigma^\*$. We welcome the reviewer's suggestion of making this adaptive. Define the adaptive proxy: $\hat{\sigma}^{*2} := \min_{j \in [M]} \text{LCB}^{\sigma}_t(j)$
> > >
> > > 3. **The proxy concentrates around $\sigma^*$:** By Lemma 3.3, after $\tilde O\left(\frac{\bar{\eta}^4}{\nu}\right)$ queries per source, $\hat{\sigma}^2_j(t)$ concentrates around $\sigma^2_j$. Corollary 3.4 then guarantees $\hat{\sigma}^{*2} \geq \frac{\sigma^{*2}}{2}$ with high probability once $\tilde O\left(\frac{\bar{\eta}^4}{\sigma^{*4}}\right)$ total source queries have been made.
> > >    - The algorithm accumulates sufficient source queries during preprocessing and initial exploration, so no prior knowledge of $\sigma^\*$ is needed for this concentration.
> > >
> > >    - Similarly, Lemma 3.5 holds after $\tilde O\left(\frac{K\bar{\eta}^2}{\sigma^{*2}}\right)$ arm queries, and PREPROCESS completes after $\tilde O\left(\frac{M\bar{\eta}^4}{\sigma^{*4}}\right)$ rounds both now in terms of $\sigma^\*$.
> > >
> > > 4. **Resulting regret with the adaptive proxy:** Since $\hat{\sigma}^{*2}$ concentrates around $\sigma^{*2}$, replacing the fixed $c^{\*}$ with $\hat{\sigma}^{*2}$ yields: $\tilde O \left(\frac{M\bar{\eta}^4\bar{\mu}}{\sigma^{*4}} + \frac{K\bar{\eta}^2\bar{\mu}}{\sigma^{*2}} + KM\bar{\mu} + \frac{M\bar{\eta}^4\bar{\mu}}{\nu} + \sigma^\* \sqrt{KT}\right)$ eliminating dependence on a user-specified $c^{\*}$ .
> > >
> > > 5. **Safeguarding against $\sigma^\* \to 0$.** In our paper we used a fixed $c^{\*}$ to handle the degenerate case $\sigma^\* \to 0$. With the adaptive scheme, a more principled choice is: $\hat{\sigma}^{*2} := \max\left[\min_{j \in [M]} \text{LCB}^{\sigma}_t(j) ,  c^{\*2}\right] \text{--(E1)}$
> > >
> > > One choice is $c^{\*} = O(T^{-1/10})$, is justified below.
> > >
> > > **Deriving the floor $c^{\*} = O(T^{-1/10})$:** Assuming $\bar{\mu} = O(1)$ and noting that for the regime $\sigma^* < 1$ (i.e., $\sigma^* \to 0$, which is precisely the case requiring a floor), the term $\frac{K\bar{\eta}^2}{c^{\*2}}$ is dominated by $\frac{(M+K)\bar{\eta}^4}{c^{\*4}}$ for small $c^{\*}$.  In this case, the instance-independent regret simplifies to: $\tilde O \left(\min\left[\frac{(M+K)\bar{\eta}^4}{\sigma^{*4}}, \frac{(M+K)\bar{\eta}^4}{c^{\*4}}\right]  +  (\sigma^\* + c^{\*})\sqrt{KT} + KM\bar{\mu} + \frac{M\bar{\eta}^4}{\nu} \right).$
> > >
> > > Noting that $\sigma^* < 1$ (as $\sigma^* \to 0$), in this regime $c^{\*} > \sigma^*$ and we can further upper bound the above expression as $\tilde O \left({ \frac{(M+K)\bar{\eta}^4}{c^{\*4}}}  +   c^{\*}\sqrt{KT} + KM\bar{\mu} + \frac{M\bar{\eta}^4}{\nu} \right).$
> > >
> > > Optimizing $f(c^{\*}) = \frac{(M+K)\bar{\eta}^4}{c^{\*4}} + c^{\*}\sqrt{KT}$ over $c^{\*}$:
> > > $$f'(c^{\*}) = -\frac{4(M+K)\bar{\eta}^4}{c^{\*5}} + \sqrt{KT} = 0 \implies c^{\*}_{\mathrm{opt}} = \left(\frac{4(M+K)\bar{\eta}^4}{\sqrt{KT}}\right)^{1/5}.$$
> > >
> > > Hence $c^{\*}_{\mathrm{opt}} = O(T^{-1/10})$ and the final regret expression becomes:
> > >
> > > $$\tilde{O}\left(\min\left[\frac{M\bar{\eta}^4\bar{\mu}}{\sigma^{*4}} + \frac{K\bar{\eta}^2\bar{\mu}}{\sigma^{*2}}, (M+K)^{1/5} (\bar{\eta})^{4/5} (KT)^{2/5} \right] + \sigma^\* \sqrt{KT} + KM\bar{\mu} + \frac{M\bar{\eta}^4\bar{\mu}}{\nu} \right),$$
> > >
> > > When $\sigma^* \geq 1$, the bound from point 4 already suffices.
> > >
> > > Note that the novel choice of $\hat{\sigma}^{*2}$ in E1 keeps preprocessing terms finite even as $\sigma^\* \to 0$, while retaining the desired $O(\sigma^\* \sqrt{KT})$ rate. The optimal $c^{\*}_{\mathrm{opt}} = O(T^{-1/10})$ derived above eliminates the need to specify $c^{\*}$ apriori.
> > >
> > > To clarify on a minor point, a finer choice of $c^{\*}$ can be obtained by retaining both terms separately rather than upper bounding $\frac{\bar{\eta}^2}{c^{\*2}}$ by $\frac{\bar{\eta}^4}{c^{\*4}}$, via a case analysis on whether $\frac{K}{M} \lessgtr \frac{\bar{\eta}^2}{c^{\*2}}$. We can include this refined analysis in the final version if needed.
> > >
> > > ---
> > >
> > > We hope to have satisfactorily answered your final concerns. We remain committed to further clarifications, given an opportunity. We sincerely urge the reviewer to kindly reconsider the score in light of the clarifications, and the regret-bound free of $c^\*$ dependence.

---

### Official Review · Reviewer_FH6S · 2026-03-07

**Soundness:** 3
**Presentation:** 1
**Significance:** 2
**Originality:** 3
**Overall Recommendation:** 4
**Confidence:** 2

**Summary:**

This paper considers a stochastic $K$-armed bandit problem with $M$ independent data sources. Each arm $i \in [K]$ is associated with an unknown mean reward $\mu_i$, and each source $j \in [M]$ is associated with an unknown noise variance $\sigma_j^2$. At each round, the learner selects an arm–source pair $(i, j)$ and observes a noisy reward whose variance depends only on the source.

The learner has no prior knowledge of either the arm means or the source variances. The objective is to minimize cumulative regret relative to the optimal arm, while adaptively selecting sources in order to mitigate the effect of heteroscedastic noise on learning efficiency.

The authors leverage an optimistic–pessimistic algorithm to obtain their regret bound, combining upper confidence bounds (UCBs) on arm rewards with lower confidence bounds (LCBs) on source variances.

Their final regret bound is $O({\sigma*}^2\sum_{i=2}^K\Delta_i^{-1})$ where $\sigma_*^2$ is the optimal variance and $\Delta_i$ is the suboptimality gap of arm $i$.

The algorithm is fairly standard: they perform mean estimation for both the arm expectations and the source variances, and then apply the corresponding concentration inequalities to derive the confidence bonuses. The resulting method is an optimistic–pessimistic bandit algorithm with an initial exploration warm-up phase.

**Compliance With Llm Reviewing Policy:**

Affirmed.

**Final Justification:**

I think the studied setup is interesting and can lead to more advanced research questions. I also think the algorithmic approach presented is clean and intuitive.
However, writing should get improved and explicit lower bound should be presented for completely characterise the learned setup.
Thus, after rebuttal, I’ll maintain my original quite positive evaluation.

**Key Questions For Authors:**

1. Is your upper bound optimal? Do you have a matching lower bound that proves this?

2. What exactly is $c^*$? Is it known to the algorithm in advance?

3. What are WC1 and WC2? Can you explicitly state a regret lower bound corresponding to each of the selected algorithms?

4. In the experiments, why were those baselines chosen? Why are they reasonable? Are there no existing algorithms that are more directly comparable?

**Limitations:**

yes

**Strengths And Weaknesses:**

**Strengths**

1. Interesting setup that could attract interest from the community, especially if extended beyond the stated assumptions and possibly to the adversarial setting.

2. Clean algorithmic approach.

3. As far as I understand, the analysis appears sound (I did not check the appendix proofs in detail).

4. The paper includes experiments: synthetic experiments on Gaussian arms and additional empirical evaluations.


**Weakness**

1. Abstract need to be re-write – it is unclear due to too-many unexplained notation.

2. Strong assumptions throughout the way – e.g., known $ \bar{\eta} $, $c*$.

3. Theorem 5.1 statementet should be simplified as this is hard to understand.

4. The paper is not fluent to the reader; it feels like a set of paragraph fixed together, many definitions with a separated intuitions paragraph, unreadable algorithm for a really unnecessary reason (simply keep all estimator computaions out of it, and keep the algorithmic scheme as clear and clean as possible).

5. There are many unclear intuitions in the paper that comes to somehow convince the approach is optimal; however, since any regret bound is not presented explicitly, those explantion just confuse. E.G., wc1, wc2 and more throughput the paper.

---

> ### Author Rebuttal · Authors · 2026-03-30
>
> ## Weaknesses
> >W1, W3, W4: Presentation Issues
>
> We sincerely thank the reviewer for the detailed presentation feedback. We agree that the overall presentation of the abstract, algorithm, theorems, and the paper can be improved and commit to making the changes (rewriting, removing estimator computations and simplification) in the final submission.
>
> >W2: Strong assumptions throughout the way – e.g., known $\bar \eta$, $c^{\*}$
>
>  We appreciate the opportunity to clarify $c^{\*}$. $c^{\*}$ is not an assumption but rather a user-defined parameter. We further discuss this in the response to "Key Questions to Authors" Q2.
>
> >W3: There are many unclear intuitions in the paper that comes to somehow convince the approach is optimal; however, since any regret bound is not presented explicitly, those explantion just confuse. E.G., wc1, wc2 and more throughput the paper.
>
> We discuss this in our responses to ""Key Questions to Authors" Q3 and Q4.
>
> ## Key Questions for Authors
> >Q1: Is your upper bound optimal? Do you have a matching lower bound that proves this?
>
> We thank the author for this question. We note that the standard K-armed MAB problem has a lower bound of $\tilde \Omega\left(\sum_{i\neq i^{\*}} \frac{1}{\Delta_i} \right)$ [1]. With a single source of variance $\sigma^2$ this bound becomes $\tilde \Omega\left(\sum_{i\neq i^{\*}} \frac{\sigma^2}{\Delta_i} \right)$. Now in the multi-source setting, any algorithm with access to the best source (with variance ${\sigma^{\*}}^2$) cannot do better than $\tilde \Omega\left(\sum_{i\neq i^{\*}} \frac{{\sigma^{\*}}^2}{\Delta_i} \right)$. SOAR's dominant regret term of $\tilde O\left( \sum_{i=2}^K \frac{{\sigma^{\*}}^2}{\Delta_i} \right)$ (Remark 5.2, point 1) matches this, establishing near-optimality.
>
> >Q2: What exactly is $c^*$? Is it known to the algorithm in advance?
>
> We appreciate the opportunity to clarify. $c^{\*}$ is a user-defined parameter. As discussed in Remark 3.2, $c^{\*}$ serves as a variance floor to prevent pathological explosion of terms like $\frac{1}{\sigma_j^4}$ when source variances approach zero. As discussed in Remark 5.2 (point 5), any fixed choice of $c^* \sim O(1)$ yields oracle-optimal leading-order regret.
>
> >Q3: What are WC1 and WC2? Can you explicitly state a regret lower bound corresponding to each of the selected algorithms?
>
> We thank the reviewer for this question. WC1 and WC2 are worst-case instances introduced in Section 4 (under the header "Some Baselines and Limitations'') and analyzed in detail in Appendix B with explicit regret upper bounds. WC1 is an instance where all sources except the optimal one have similarly high variance, causing Baseline-1 (Uniform Source MAB) to suffer regret scaling with $\sigma_{\max}$ rather than $\sigma^{\*}$. WC2 is an instance where all source variances are nearly identical, causing Baseline-2 (Two-Phase MAB) to waste excessive effort distinguishing indistinguishable sources. The corresponding regret bounds for both baselines and SOAR are compared in Remark 5.3.
>
> Regarding lower bounds: for Baseline 1, the classical instance-dependent lower bound [1] applied with effective variance $\tilde \sigma^2 = \frac{\sum_{j=1}^M \sigma_j^2}{M}$ directly gives $\tilde \Omega \left(\sum_{i\neq i^{\*}}\frac{{\tilde\sigma}^2}{\Delta_i}\right)$, which under WC1 becomes $\tilde \Omega \left(\sum_{i\neq i^{\*}}\frac{{\sigma_{\text{max}}}^2}{\Delta_i}\right)$, confirming the upper bound is tight. For Baseline 2, the first phase is a best-arm identification problem over sources, whose known sample-complexity lower bounds [2] yield the $\frac{1}{\left(\Delta_j^{\sigma^2}\right)^2}$ dependence found in Baseline 2's bound in Remark 5.3. The second phase of baseline 2 is a standard MAB on the identified source, whose regret is tight by the classical lower bound [1]. This confirms the upper bound is tight.
>
> >Q4: In the experiments, why were those baselines chosen? Why are they reasonable? Are there no existing algorithms that are more directly comparable?
>
> As discussed in our related work (Section 1) and Appendix B, no existing algorithm addresses joint source-arm selection under unknown heterogeneous variances. Existing variance-adaptive methods, such as UCB-V [3], only handle arm selection and have no mechanism for source selection, and pairing them with any naive source strategy recovers Baseline 1. Our baselines, thus, represent the most natural strategies for this setting, and we will add a comparison table of related work to make this distinction clearer.
>
> ## References:
>
> [1] Lattimore, T. and Szepesvári, C. Bandit algorithms. Cambridge University Press, 2020
>
> [2] Audibert, J.-Y., Bubeck, S., and Munos, R. "Best Arm Identification in Multi-Armed Bandits." COLT, 2010
>
> [3] Audibert, J. Y., Munos, R., and Szepesvári, C. Exploration–exploitation tradeoff using variance estimates in multi-armed bandits. Theoretical Computer Science, 410(19):1876–1902, 2009

---

> > ### Author Rebuttal · Reviewer_FH6S · 2026-04-02
> >
> > I thank the authors for their response, which clarified my questions, and I decided to keep my score;
> > I do think that more explicit lower bound should be presented and writing should get much improved.

---

> > > ### Author Response · Authors · 2026-04-08
> > >
> > > We thank the reviewer for taking the time to review our rebuttal and detailed feedback regarding our work. We hope our responses have adequately addressed your concerns, and we sincerely appreciate your consideration in maintaining the positive score. Your comments and suggestions have been very helpful in strengthening the paper, and we will incorporate the clarifications and revisions discussed in this rebuttal into the final version.

---

### Official Review · Reviewer_H6av · 2026-03-10

**Soundness:** 3
**Presentation:** 3
**Significance:** 3
**Originality:** 3
**Overall Recommendation:** 5
**Confidence:** 4

**Summary:**

The authors study online learning with heterogeneous feedback in stochastic multi-armed bandits. They introduce SOAR (Source-Optimistic Adaptive Regret Minimization), which jointly selects arms and data sources when observations come from sources with unknown, differing noise variances.

Each arm has an unknown reward mean, and each source adds noise with an unknown variance. SOAR first eliminates clearly high-variance sources, then uses an adaptive UCB–LCB strategy to guide arm–source selection. It combined upper confidence bounds on arm rewards with lower confidence bounds on source variances.

Theoretical analysis shows near-oracle instance-dependent regret (up to logarithmic factors) relative to a bandit algorithm that knows the best low-variance source. Experiments on synthetic benchmarks and a MovieLens dataset show SOAR achieves lower regret than baselines such as uniform source selection and a two-phase source-then-arm approach.

**Compliance With Llm Reviewing Policy:**

Affirmed.

**Final Justification:**

I justifications for the hyperparameters are all reasonable.

**Key Questions For Authors:**

What are some other, specific problems that could be modeled by multiple heterogeneous sources?

Given the large number of hyperparameters, could this algorithm by practically utilized?

Is source code available for reproduction?

**Limitations:**

Real-world problems (specifically recommender-system problems) can be very large-scale. Would SOAR scale to 10's of thousands of items and millions of reviewers? If not, would that be an interesting avenue for future work?

**Strengths And Weaknesses:**

The paper is theoretically sound. Practical application is somewhat complicated by the presence of many hyperparameters.

Plot text is small and hard to read. Figure captions do not sufficiently explain the plots. Labeling is inconsistent (e.g., “uniform” vs. “Baseline-1”). There is a line "all sourcest, support(Djt) ⊆[¯ η,¯η].".  I'm not sure what this is supposed to say.

This is an interesting problem, and it is clearly-motivated. Empirical comparisons to stronger algorithms (in particular, algorithms that account for heteroscedasticity) and on more problems would enhance the reader's understanding of the algorithm's place in field.

---

> ### Author Rebuttal · Authors · 2026-03-30
>
> We thank the reviewer for their careful reading and constructive feedback. We address each point below.
> ## Weaknesses
>  > W1: Plot text is small and hard to read. Figure captions do not sufficiently explain the plots. Labeling is inconsistent (e.g., “uniform” vs. “Baseline-1”). There is a line "all sourcest, support(Djt) $\subseteq [\bar\eta, \bar \eta]$". I'm not sure what this is supposed to say.
>
> We sincerely thank the reviewer for pointing out these issues. We will enlarge figure text, add detailed captions, and standardize labeling throughout. Regarding the line pointed out, i.e., Assumption 2.1, this line contains two overlapping typos and should read: "For all sources $j$, support$(D_j) \subseteq [-\bar\eta, \bar\eta]$. We will fix this in the revision.
>
> > W2: Empirical comparisons to stronger algorithms (in particular, algorithms that account for heteroscedasticity) and on more problems would enhance the reader's understanding of the algorithm's place in field.
>
> As addressed in our related work section, no existing bandit algorithm (to the best of our knowledge) directly addresses multi-source settings with unknown heteroscedastic noise. Existing variance-adaptive methods (e.g., UCB-V [1]) adapt to per-arm variance within a single source but do not reason about selecting among multiple sources. Our baselines represent the most natural strategies for this setting, and we will add a comparison table of related work to make this distinction clearer.
>
> [1] Audibert, J. Y., Munos, R., and Szepesvári, C. Exploration–exploitation tradeoff using variance estimates in multi-armed bandits. Theoretical Computer Science, 410(19):1876–1902, 2009.
>
> ## Key Questions For Authors:
> >Q1: What are some other, specific problems that could be modeled by multiple heterogeneous sources?
>
> Beyond the examples of clinical trials given in the paper, our framework naturally applies to crowdsourcing platforms, where tasks to evaluate (e.g., which product description is best) serve as arms and crowd workers with varying reliability serve as sources. Another problem is LLM evaluation, where arms are different prompt templates or strategies you're evaluating, and sources are different LLM judges or human evaluators scoring them. Each evaluator has different noise/consistency in their ratings, and we want to identify the best prompt while learning which evaluator is the most reliable.
>
> >Q2: Given the large number of hyperparameters, could this algorithm by practically utilized?
>
> While SOAR involves several parameters, only $\nu$, $\delta$, and ${c^\*}$ need to be specified by the user.  The remaining quantities $(\alpha, \beta, \tau_p)$ can be derived from this knowledge (Equations 11, 12, and Theorem 4.1). $\gamma$ does not influence the algorithm's execution and is introduced purely in the regret analysis (Section 5) to handle the degenerate regime where $\sigma^{\*}  \to 0$, as discussed in Remark 5.2 (point 2). Crucially, none requires knowledge of the unknown problem quantities such as $\Delta^\mu$ or $\Delta^{\sigma^2}$. Hence, to the best of our knowledge, we believe that hyperparameters are not a barrier to practical utilization.
>
> >Q3: Is source code available for reproduction?
>
> We hope to make the code available in the final submission.
>
> ## Limitations
> >L1: Real-world problems (specifically recommender-system problems) can be very large-scale. Would SOAR scale to 10's of thousands of items and millions of reviewers? If not, would that be an interesting avenue for future work?
>
> The costs for the preprocessing subroutine (PREPROCESS), along with the initial exploration cost to ensure the core concentration inequalities hold, scale linearly with $M$ and $K$.  The instance-independent (worst-case) regret scales as $(\sigma^{\*} + c^{\*})\sqrt{KT}$, growing with $\sqrt{K}$, which is standard for UCB-type algorithms [2].
>
> However, we note that PREPROCESS and the initial exploration are parallelizable since arms and sources can be queried independently ($M$ sources queried with a single arm, $K$ arms queries with a single source), and for certain applications, pre-filtering using historical data or other contexts could significantly reduce $M$ and $K$ before running SOAR.
>
> For example, in the movie recommendation setting, one could pre-filter reviewers based on activity level or historical rating consistency to retain only a manageable subset of sources, and similarly restrict the movie pool to active or recently released titles, before applying SOAR on the reduced problem.
>
> We agree that scaling to millions of sources and tens of thousands of items is an interesting avenue for future work, and we thank the reviewer for this suggestion.
>
> [2] Lattimore, T. and Szepesvári, C. Bandit algorithms. Cambridge University Press, 2020

---

> > ### Author Rebuttal · Reviewer_H6av · 2026-04-02
> >
> > >  only $\nu$, $\delta$, and ${c^*}$ need to be specified
> >
> > This is still quite a lot, and the difficulty of tuning them to a problem will depend, in part (ideally mostly), on the sensitivity of the method to them. Ablation studies on several example problems would help mitigate concerns.

---

> > > ### Author Response · Authors · 2026-04-08
> > >
> > > ## Reg. concern about parameters.
> > > We thank the reviewer for taking the time to review our rebuttal and for their insightful feedback.
> > > We first clarify that $\nu$, $\delta$, and $ c^\*$ are not hyperparameters in the conventional sense, but rather algorithm parameters with theoretically prescribed settings (Remark 5.2). They do not require tuning or cross-validation. The parameter $\delta$ is a standard confidence level, while the parameters $\nu$ and $c^\*$ have principled settings described in Remark 5.2. To the extent that the reviewer's request for an "ablation'' refers to removing these parameters entirely, we note that this is not meaningful here: each parameter plays a distinct and necessary role in the algorithm's design (e.g., $c^\*$ controls variance separation, $\nu$ stabilizes fourth-moment estimation, $\delta$ sets the confidence level). We would be happy to be corrected if we have misunderstood the reviewer's intent; regardless, we would be happy to include empirical sensitivity plots showing regret as a function of $(\nu, \delta, c^\*)$ in the final submission (ICML rules do not permit paper updates during the discussion period).
> > >
> > > That said, the precise impact of each parameter choice is quantified in our regret bound (Theorem 5.1). Remark 5.2 (points 3 and 5) provides explicit guidance on how to set $\nu$ and $c^*$ based on these tradeoffs.
> > >
> > > Moreover, as discussed in our response to Reviewer TAJz and bYSU (Rebuttal Acknowledgment - Re. Concern about Adaptive $c^\*$), $c^*$ can be set adaptively, thereby eliminating the need for the user to specify it. We conjecture that a similar adaptive approach can be extended to $\nu$ as well, leaving only the standard confidence parameter $\delta$.
> > >
> > > We hope to have satisfactorily answered your final concerns. We remain committed to further clarifications, given an opportunity. We sincerely urge the reviewer to kindly reconsider the score in light of the clarifications provided above.

---

### Official Review · Reviewer_bYSU · 2026-03-13

**Soundness:** 2
**Presentation:** 2
**Significance:** 3
**Originality:** 3
**Overall Recommendation:** 4
**Confidence:** 3

**Summary:**

This paper studies stochastic multi-armed bandits where each arm can be sampled from different sources, and the reward variance differs across sources and is unknown to the learner. The paper proposes an algorithm that achieves both instance-dependent and worst-case regret bounds comparable to the setting where the optimal source (with the lowest variance) is known in advance. Experiments are conducted to validate the effectiveness of the proposed method.

**Compliance With Llm Reviewing Policy:**

Affirmed.

**Final Justification:**

My concerns regarding the presentation and the bounded-noise assumption have been addressed, so I have increased my score. However, it still seems challenging to choose $c_*$ in a way that adapts to different variance regimes.

**Key Questions For Authors:**

- How should $c_*$ be specified so that it can automatically adapt to different regimes of $\sigma_j$?

- How does the parameter $\gamma$ affect the implementation of the algorithm?

**Limitations:**

yes

**Strengths And Weaknesses:**

### Strengths

The strengths of this paper are as follows:

- The problem formulation is novel and interesting. Although the paper makes several simplifications such as assuming that the mean of each arm is the same and that, within each resource, the variance is identical across all arms. It still provides a useful baseline model for the multi-source MAB setting.

- The proposed algorithm achieves a nearly optimal regret bound without requiring prior knowledge of the optimal environment with the smallest covariance.

### Weaknesses

The weaknesses of the paper are as follows:

-  The proposed method requires setting the parameter $c_*$, which should scale with the unknown covariance $\sigma_j$. Specifically, the algorithm requires choosing $\alpha$ according to the relationship between $c_*$ and $\sigma_j$, but $\sigma_j$ is unknown in practice. This makes the parameter setting somewhat unclear.

- The proposed method requires the reward to be bounded. It is unclear how the framework can be extended to the case of sub-Gaussian noise.

- The proof of Lemma 3.1 is somewhat confusing. In Eq. (2), $\hat{\sigma}^2_{j,pre}(t)$ is defined using $\hat{\mu}\_{i_s}$, but $\hat{\mu}\_{i_s}$ itself is not defined. I presume that $\hat{\mu}\_{i_s}$ denotes the mean estimator corresponding to sample $X_s$. However, in the proof of Lemma 3.1 the subscript $i_s$ is omitted. If one considers the corresponding estimator $\hat{\mu}\_{i_k}$ for each sample $X_k$, it is unclear whether Term II can still be bounded using the current arguments. Although this does not appear to be a critical issue, since in the preprocessing stage one could simply sample from a fixed arm, it would be helpful if the authors clarified the notation.

-  It is unclear how the parameter $\gamma$ affects the implementation of the algorithm. I could only identify $\gamma$ appearing in Algorithm 1 (line 1), but it is not clear how it influences subsequent steps.

- There are several typos and presentation issues in the paper:

  - Line 28: should be $\tilde{O}(\cdot)$
  - Assumption 1: $[\bar{\eta}, \bar{\eta}] \rightarrow [-\bar{\eta}, \bar{\eta}]$
  - The text in the figures is too small and appears compressed.


Overall, I find that the paper proposes an interesting problem and presents a promising algorithm. However, several aspects of the paper remain unclear, particularly regarding the algorithmic definition, parameter settings, and parts of the analysis. I would consider increasing my score if these concerns are properly addressed.

---

> ### Author Rebuttal · Authors · 2026-03-30
>
> We thank the reviewer for their careful reading and constructive feedback. We address each point below.
> ## Weaknesses
> >W1: The proposed method requires setting the parameter $c^{\*}$ , which should scale with the unknown covariance $\sigma_j$. Specifically, the algorithm requires choosing according to the relationship between $c^{\*}$ and $\sigma_j$ , but is unknown in practice. This makes the parameter setting somewhat unclear.
>
>  We thank the reviewer for raising this point. We would first like to clarify that we discuss variances across sources, not covariances.
> As discussed in Remark 3.2 $c^{\*}$ serves as a variance floor to prevent pathological explosion of terms like $\frac{1}{\sigma_j^4}$ when source variances approach zero. As discussed in Remark 5.2 (point 5), any fixed choice of $c^{\*} \sim O(1)$ yields oracle-optimal leading-order regret; the cost of setting $c^{\*}$ small appears only in the preprocessing and exploration terms.
> That said, we acknowledge the reviewer's practical concern: the $\frac{1}{{c^{\*}}^4}$ dependence in the preprocessing and exploration budget means that an excessively small $c^{\*}$ incurs a high (though finite) upfront cost. We also note that if $c^{\*}$ is set too large, the algorithm transitions to the alternate regime of Theorem 5.4 rather than failing. We consider the case of an adaptive $c^*$ in Q1 of this response.
>
> >W2: The proposed method requires the reward to be bounded. It is unclear how the framework can be extended to the case of sub-Gaussian noise.
>
> The boundedness assumption is used to apply Freedman's concentration inequality [1] in our key lemmas. Extending to sub-Gaussian noise is feasible from a concentration standpoint, as one can substitute sub-Gaussian MDS bounds (e.g., [2], Theorem 2.3) or variance-concentration results for sub-Gaussian variables (e.g., [3], Theorem S1; [4], Theorem 2.10).
>
> [1] Raban, D. Statistics 210b lecture 6 notes: Theorem 1.1 (Freedman’s inequality). Online lecture notes. Accessed via Pillowmath repository.
>
> [2] Wainwright, M. Basic tail and concentration bounds, 2015.
>
> [3] Fontaine et al., Online A-Optimal Design and Active Linear Regression, ICML 2021.
>
> [4] Boucheron et al., Concentration Inequalities, 2013
>
>
> >W3 (Lemma 3.1 clarification): The proof of Lemma 3.1 is somewhat confusing $\dots$ if the authors clarified the notation.
>
> We note that the subscript convention $i_s$ for the arm selected at time s is established in the Problem Setting (Section 2), and the empirical mean estimator $\mu_{i}(t)$ is defined in Equation (2) of Section 3. The notation $\hat \mu_{i_s}$ in the variance estimator refers to this same quantity evaluated for arm $i_s$. We acknowledge, however, that the omission of the $i_s$ subscript in the proof of Lemma 3.1 is a source of confusion, and we should have explicitly noted that since PREPROCESS fixes a single arm $i_0$ (Algorithm 1, line 2). We will correct this in the revised manuscript.
>
> >W4: It is unclear how the parameter $\gamma$ affects the implementation of the algorithm. I could only identify appearing in Algorithm 1 (line 1), but it is not clear how it influences subsequent steps
>
>  We thank the reviewer for this observation. We would like to clarify that $\gamma$ does not appear in Algorithm 1 (PREPROCESS), and we believe the reviewer may be referring to Algorithm 2 (SOAR), where $\gamma$ is listed as an input parameter on line 1.
> The parameter $\gamma$ does not influence the algorithm's execution and is introduced purely in the regret analysis (Section 5) to handle the degenerate regime where $\sigma^* \to 0$, as discussed in Remark 5.2 (point 2). In typical settings where $\sigma^*$ is bounded away from zero, $\gamma$ can be set to any small constant without affecting the dominant regret term. We acknowledge that listing $\gamma$ as an input to Algorithm 2 is misleading and will remove it from the algorithm description in the revised manuscript, instead introducing it at the point of the regret analysis.
>
> >W5: Presentation Issues
>
> We sincerely thank the reviewer for pointing out these issues, we will fix the typos and enlarge the figures in revision.
>
> ## Key Questions to Authors
> >Q1: How should $c^{\*}$ be specified so that it can automatically adapt to different regimes of $\sigma_j$?
>
> In W1 we discuss the reasoning behind choosing $c^{\*}$ as a small constant. We now briefly discover the challenges with an adaptive $c^{\*}$.
> One natural idea for adaptive selection would be to optimize the regret expression using an LCB-based estimate of $\sigma^{\*}$; however, our current concentration inequalities and corollaries were derived under a fixed $c^{\*}$ and need not hold under an adaptive scheme, raising nontrivial analytical complications. We view principled adaptive selection of $c^{\*}$ as an interesting direction for future work.
>
> >Q2: How does the parameter $\gamma$ affect the implementation of the algorithm?
>
>  We would like to refer the reviewer to the answer for W4.

---

> > ### Author Rebuttal · Reviewer_bYSU · 2026-04-03
> >
> > Thank you for the response. My concerns regarding the presentation and the bounded-noise assumption have been addressed. However, it still seems challenging to choose $c_*$ in a way that adapts to different variance regimes.

---

> > > ### Author Response · Authors · 2026-04-08
> > >
> > > ## Re. Concern about Adaptive $c^\*$
> > >
> > > We thank the reviewer for taking the time to review our rebuttal and for their insightful feedback.
> > >
> > > ---
> > >
> > > 1. **Quantifying the role of $c^{\*}$ in regret:** We first clarify precisely how $c^{\*}$ enters the regret bound. It does so through three components:
> > >    - PREPROCESS (Theorem 4.1), contributing $\tilde O\left(\frac{M\bar{\eta}^4}{c^{\*4}}\right)$ to the regret.
> > >    - Mean reward concentration (Lemma 3.5), contributing $\tilde O\left(\frac{K\bar{\eta}^2}{c^{\*2}}\right)$ to the regret.
> > >    - Variance sandwiching (Corollary 3.4), contributing $\tilde O\left(\frac{M\bar{\eta}^4}{c^{\*4}}\right)$ to the final regret.
> > >
> > >    If $\sigma^\*$ were known, the natural choice would be $c^{\*} = \sigma^*$.
> > >
> > > 2. **An adaptive proxy for $\sigma^\*$:** In the paper, $c^{\*}$ serves as a fixed proxy for the unknown $\sigma^\*$. We welcome the reviewer's suggestion of making this adaptive. Define the adaptive proxy: $\hat{\sigma}^{*2} := \min_{j \in [M]} \text{LCB}^{\sigma}_t(j)$
> > >
> > > 3. **The proxy concentrates around $\sigma^*$:** By Lemma 3.3, after $\tilde O\left(\frac{\bar{\eta}^4}{\nu}\right)$ queries per source, $\hat{\sigma}^2_j(t)$ concentrates around $\sigma^2_j$. Corollary 3.4 then guarantees $\hat{\sigma}^{*2} \geq \frac{\sigma^{*2}}{2}$ with high probability once $\tilde O\left(\frac{\bar{\eta}^4}{\sigma^{*4}}\right)$ total source queries have been made.
> > >    - The algorithm accumulates sufficient source queries during preprocessing and initial exploration, so no prior knowledge of $\sigma^\*$ is needed for this concentration.
> > >
> > >    - Similarly, Lemma 3.5 holds after $\tilde O\left(\frac{K\bar{\eta}^2}{\sigma^{*2}}\right)$ arm queries, and PREPROCESS completes after $\tilde O\left(\frac{M\bar{\eta}^4}{\sigma^{*4}}\right)$ rounds both now in terms of $\sigma^\*$.
> > >
> > > 4. **Resulting regret with the adaptive proxy:** Since $\hat{\sigma}^{*2}$ concentrates around $\sigma^{*2}$, replacing the fixed $c^{\*}$ with $\hat{\sigma}^{*2}$ yields: $\tilde O \left(\frac{M\bar{\eta}^4\bar{\mu}}{\sigma^{*4}} + \frac{K\bar{\eta}^2\bar{\mu}}{\sigma^{*2}} + KM\bar{\mu} + \frac{M\bar{\eta}^4\bar{\mu}}{\nu} + \sigma^\* \sqrt{KT}\right)$ eliminating dependence on a user-specified $c^{\*}$ .
> > >
> > > 5. **Safeguarding against $\sigma^\* \to 0$.** In our paper we used a fixed $c^{\*}$ to handle the degenerate case $\sigma^\* \to 0$. With the adaptive scheme, a more principled choice is: $\hat{\sigma}^{*2} := \max\left[\min_{j \in [M]} \text{LCB}^{\sigma}_t(j) ,  c^{\*2}\right] \text{--(E1)}$
> > >
> > > One choice is $c^{\*} = O(T^{-1/10})$, is justified below.
> > >
> > > **Deriving the floor $c^{\*} = O(T^{-1/10})$:** Assuming $\bar{\mu} = O(1)$ and noting that for the regime $\sigma^* < 1$ (i.e., $\sigma^* \to 0$, which is precisely the case requiring a floor), the term $\frac{K\bar{\eta}^2}{c^{\*2}}$ is dominated by $\frac{(M+K)\bar{\eta}^4}{c^{\*4}}$ for small $c^{\*}$.  In this case, the instance-independent regret simplifies to: $\tilde O \left(\min\left[\frac{(M+K)\bar{\eta}^4}{\sigma^{*4}}, \frac{(M+K)\bar{\eta}^4}{c^{\*4}}\right]  +  (\sigma^\* + c^{\*})\sqrt{KT} + KM\bar{\mu} + \frac{M\bar{\eta}^4}{\nu} \right).$
> > >
> > > Noting that $\sigma^* < 1$ (as $\sigma^* \to 0$), in this regime $c^{\*} > \sigma^*$ and we can further upper bound the above expression as $\tilde O \left({ \frac{(M+K)\bar{\eta}^4}{c^{\*4}}}  +   c^{\*}\sqrt{KT} + KM\bar{\mu} + \frac{M\bar{\eta}^4}{\nu} \right).$
> > >
> > > Optimizing $f(c^{\*}) = \frac{(M+K)\bar{\eta}^4}{c^{\*4}} + c^{\*}\sqrt{KT}$ over $c^{\*}$:
> > > $$f'(c^{\*}) = -\frac{4(M+K)\bar{\eta}^4}{c^{\*5}} + \sqrt{KT} = 0 \implies c^{\*}_{\mathrm{opt}} = \left(\frac{4(M+K)\bar{\eta}^4}{\sqrt{KT}}\right)^{1/5}.$$
> > >
> > > Hence $c^{\*}_{\mathrm{opt}} = O(T^{-1/10})$ and the final regret expression becomes:
> > >
> > > $$\tilde{O}\left(\min\left[\frac{M\bar{\eta}^4\bar{\mu}}{\sigma^{*4}} + \frac{K\bar{\eta}^2\bar{\mu}}{\sigma^{*2}}, (M+K)^{1/5} (\bar{\eta})^{4/5} (KT)^{2/5} \right] + \sigma^\* \sqrt{KT} + KM\bar{\mu} + \frac{M\bar{\eta}^4\bar{\mu}}{\nu} \right),$$
> > >
> > > When $\sigma^* \geq 1$, the bound from point 4 already suffices.
> > >
> > > Note that the novel choice of $\hat{\sigma}^{*2}$ in E1 keeps preprocessing terms finite even as $\sigma^\* \to 0$, while retaining the desired $O(\sigma^\* \sqrt{KT})$ rate. The optimal $c^{\*}_{\mathrm{opt}} = O(T^{-1/10})$ derived above eliminates the need to specify $c^{\*}$ apriori.
> > >
> > > To clarify on a minor point, a finer choice of $c^{\*}$ can be obtained by retaining both terms separately rather than upper bounding $\frac{\bar{\eta}^2}{c^{\*2}}$ by $\frac{\bar{\eta}^4}{c^{\*4}}$, via a case analysis on whether $\frac{K}{M} \lessgtr \frac{\bar{\eta}^2}{c^{\*2}}$. We can include this refined analysis in the final version if needed.
> > >
> > > ---
> > >
> > > We hope to have satisfactorily answered your final concerns. We remain committed to further clarifications, given an opportunity. We sincerely urge the reviewer to kindly reconsider the score in light of the clarifications, and the regret-bound free of $c^\*$ dependence.

---

### Decision · Program_Chairs · 2026-04-30

**Decision:**

Accept (regular)

**Comment:**

This paper studies the finite armed MAB problem with heterogeneous data sources. It is assumed that all data sources share the same means, but have datasource-dependent variance levels.
Hence an optimal algorithm needs to simultaneously learn the MAB and also identify the optimal datasource to achieve optimal regret.

They propose an algorithm where the dominating term only depends on the optimal noise level.

The reviewers are all in favour of acceptance and agree that their concerns have been sufficiently addressed in the rebuttal. There is an agreement that the problem setting is of interest and the technical solution sufficiently novel.